

# AMOC-emulator M-AMOC1.0 for uncertainty assessment of future projections

Pepijn Bakker[1,*] and Andreas Schmittner[1]

[1]College of Earth, Ocean and Atmospheric Sciences, Oregon State University, USA
[*]Now at MARUM – Center for Marine Environmental Sciences, University of Bremen, Bremen, Germany

*Correspondence to:* Pepijn Bakker (pbakker@marum.de)

**Abstract.** State-of-the-science global climate models show that global warming is likely to weaken the Atlantic Meridional Overturning Circulation (AMOC). While such models are arguably the best tools to perform AMOC projections, they do not allow a comprehensive uncertainty assessment because of limited computational resources. Here we present an AMOC-emulator, a box model with a number of free parameters that can be tuned to mimic the sensitivity of the AMOC to climate change of a specific global climate model. The AMOC-emulator (M-AMOC1.0) is applied to simulations of global warming and melting of the Greenland Ice Sheet, performed with an intermediate complexity model. Predictive power of the AMOC-emulator is shown by comparison with a number of additional warming and Greenland Ice Sheet melt scenario that have not been used in the tuning of the AMOC-emulator, highlighting the potential of the AMOC-emulator to assess the uncertainty in AMOC projections.

## 1 Introduction

The Atlantic Meridional Overturning Circulation (AMOC) is an important part of the climate system due to its effects on transports of heat, carbon, nutrients and other tracers. Projections consistently show reductions of the AMOC due to future global warming and changes in freshwater fluxes (*IPCC Climate Change*, 2013), with the possibility of an irreversible transition to a shutdown state (Stommel, 1961; Stouffer et al., 2006), a prime example of a tipping point in the climate system (Lenton et al., 2008). However, previous AMOC projections did not consider the role of enhanced melting of the Greenland Ice Sheet (GIS) in a warming world (*IPCC Climate Change*, 2013). In an ongoing model inter-comparison project (AMOCMIP), a number of state-of-the-science ocean-atmosphere general circulation models (GCMs) are forced with projected changes in greenhouse-gas concentrations and realistic GIS melt to improve existing AMOC projections (http://people.oregonstate.edu/~schmita2 /Projects/AMOC/). Large ensemble simulations are necessary for probabilistic projections, policy relevant risk assessment of future emission scenarios and to assess the uncertainties of AMOC projections due to climate sensitivity, polar amplification, GIS melt and model dependent sensitivity of the AMOC to such climatic changes. The high degree of complexity and spatial resolution of GCMs make them too computationally expensive to perform such an analysis and thus a model is needed that is much cheaper to run, but nonetheless captures important characteristics of the GCM's AMOC response to forcing changes.





To this end we build on previous efforts to develop an AMOC-emulator: a simple box model that uses physical relationships to represent the most important mechanisms and feedbacks that govern the AMOC's response to changes in surface temperatures and freshwater input. The AMOC-emulator presented here is an adjusted version of the multi-box ocean model of Zickfeld et al. (2004), coupled to a 1D atmospheric energy balance model to include the first-order feedback from the at-

mosphere on AMOC changes, the so-called Bjerknes feedback (Rahmstorf and Willebrand, 1981). It describes the AMOC strength as a linear function of the density contrast between the North Atlantic and the South Atlantic, as first introduced by Stommel (1961). The AMOC-emulator can reproduce the behavior of a specific GCM by tuning a number of free parameters such that the difference between the AMOC's response in the GCM and the AMOC emulator to a given set of changes in boundary conditions is minimized. The combined physics-based AMOC model and the statistical framework presented in

this manuscript is dubbed M-AMOC1.0. Previous studies have used simple models to emulate the AMOC behavior in higher complexity climate models, either based on a physical model (Zickfeld et al., 2004; Schleussner et al., 2011) or on a statistical model (Challenor et al., 2006; Schleussner et al., 2014). The approach described here is designed specifically to allow a limited number of GCM climate projections, limited in the forcing scenarios that have been sampled or simulation length, from multiple climate models to be combined into a Bayesian framework of century time-scale probabilistic AMOC projections.

Nonetheless, we acknowledge the uncertainty underlying the AMOC-emulator, especially considering that many fundamental aspects of the mechanisms driving and sustaining the AMOC are still under debate (e.g. Lozier, 2010), as is the assumption that the meridional Atlantic density contrast between the North Atlantic and the South Atlantic is the first order driver of the AMOC (de Boer et al., 2010; Butler et al., 2016).

In the following the governing equations of the AMOC-emulator are given and a description of how parameters in the

AMOC-emulator can be tuned towards AMOC projections of an individual higher order climate model using so-called Simulated Annealing (SA). Afterwards, the methodology will be applied to a series of climate projections with the model of intermediate complexity UVic forced by different scenarios for greenhouse-gas concentration changes and melting of the GIS. In the final discussion section, we will investigate the predictive power of the AMOC-emulator by comparing independent UVic AMOC projections with the outcomes of the UVic-based AMOC-emulator.

## 2  Methods

### 2.1  AMOC-emulator

The AMOC-emulator presented here is an extension of the box model described by Zickfeld et al. (2004), that is in turn an extension of the classical Stommel model (Stommel, 1961). A 1-D atmospheric energy balance model (Sellers, 1969) is

added to include the feedback between atmospheric meridional heat transport and AMOC induced changes in the meridional temperature gradient, the so-called Bjerknes feedback. The oceanic part of the AMOC-emulator consists of four well mixed boxes that represent the low-latitude upper Atlantic ocean, North Atlantic, South Atlantic and the deep Atlantic Ocean (Fig. 1).





The meridional ocean volume transport of the AMOC ($\mathrm{m^3 s^{-1}}$) is proportional to the density gradient between the North and South Atlantic:

$$m = k\frac{\rho_3 - \rho_1}{\rho_0} \tag{1}$$

where $k$ is a hydraulic constant, a tunable parameter (Tab. 2) that links the meridional density difference to the AMOC strength and $\rho_0$ is a reference density for sea water. Densities are derived from temperature and salinity following the non-linear equation of state of seawater of Millero and Poisson (1981). Temperature and salinity in the different boxes are a function of advection by the AMOC and exchange with the overlying atmosphere. Atmosphere-to-ocean heat exchange $\boldsymbol{H_{a2o}}$ relates to the temperature difference between the ocean ($\boldsymbol{T_o}$) and the atmospheric surface temperature ($\boldsymbol{T_a}$) through a thermal coupling constant $\Gamma$, a parameter used in our tuning procedure (Tab. 2):

$$\boldsymbol{H_{a2o}} = \Gamma(\boldsymbol{T_a} - \boldsymbol{T_o}) \tag{2}$$

Note that throughout the text we will use bold font to indicate arrays. Meridional freshwater fluxes between the South Atlantic and the low-latitude Atlantic ($F_1$) and between the low-latitude Atlantic and the North Atlantic ($F_2$) are prescribed and they represent both atmospheric water vapor transport as well as wind-driven oceanic freshwater transport. Note that both $F_1$ and $F_2$ are defined positive for northward freshwater transport, therefore, poleward atmospheric water vapour transport is a positive contribution to $F_2$ and a negative contribution to $F_1$. The sign of $F_1$ and $F_2$ depends on the sum of both atmospheric water vapour and wind-driven oceanic freshwater transport. Combining equations (1) and (2), changes in ocean temperature ($\boldsymbol{T_o}$) and salinity ($\boldsymbol{S}$) as a function of time ($t$) are given by the following set of ordinary differential equations for the four ocean boxes (see Fig. 1 for the definitions and extents of the different boxes):

$$\frac{\delta T_{o1}}{\delta t} = \frac{m}{V_1}(T_{o4} - T_{o1}) + \lambda_1(T_{a1} - T_{o1}) \tag{3}$$

$$\frac{\delta T_{o2}}{\delta t} = \frac{m}{V_2}(T_{o1} - T_{o2}) + \lambda_2(T_{a2} - T_{o2}) \tag{4}$$

$$\frac{\delta T_{o3}}{\delta t} = \frac{m}{V_3}(T_{o2} - T_{o3}) + \lambda_3(T_{a3} - T_{o3}) \tag{5}$$

$$\frac{\delta T_{o4}}{\delta t} = \frac{m}{V_4}(T_{o3} - T_{o4}) \tag{6}$$

$$\frac{\delta S_1}{\delta t} = \frac{m}{V_1}(S_4 - S_1) + \frac{S_0 F_1}{V_1} \tag{7}$$

$$\frac{\delta S_2}{\delta t} = \frac{m}{V_2}(S_1 - S_2) + \frac{S_0(F_1 - F_2)}{V_2} \tag{8}$$

$$\frac{\delta S_3}{\delta t} = \frac{m}{V_3}(S_2 - S_3) + \frac{S_0(F_2 + F_{GIS})}{V_3} \tag{9}$$

$$\frac{\delta S_4}{\delta t} = \frac{m}{V_4}(S_3 - S_4) \tag{10}$$





Here the box volumes are given by $V$, $\lambda$ are individual coupling constants and $F_{gis}$ is the imposed melt rate of the GIS. The individual coupling constants in the different boxes are related to the thermal coupling constant $\Gamma$, the box thicknesses $z$, $\rho_0$ and the specific heat capacity of seawater $c_o$:

$$\lambda = \frac{\Gamma}{c_o \rho_o z} \tag{11}$$

Changes in atmospheric water vapour component of the meridional freshwater fluxes $F$ (i.e. $F_1$ and $F_2$) are parameterized as a function of global temperature anomalies $\Delta T_{glob}$ following Zickfeld et al. (2004):

$$F = F_0 + h \Delta T_{glob} \tag{12}$$

Here $F_0$ are the combined wind-driven oceanic and atmospheric meridional freshwater fluxes for the reference state and h the regional hydrological sensitivities. In this description we make use of the proportionality between global temperature changes and meridional atmospheric water vapour transport found in complex climate models (Manabe and Stouffer, 1981; Ganopolski et al., 2001). Freshwater fluxes $F_1$, $F_2$ and coefficients $h_1$ and $h_2$ are included in the tuning procedure (Tab. 2). We assume that wind-driven oceanic meridional freshwater transport is static. The freshwater term in the salinity calculation of the North Atlantic box (Eq. 9) allows to include a freshwater forcing originating from the GIS ($F_{GIS}$). Note that this freshwater term is not compensated for elsewhere and therefore leads to global mean salinity changes.

The 1D energy balance model describes the atmospheric surface temperatures $T_a$ as a function of the top-of-the-atmosphere incoming shortwave (first term on rhs of Eq. 13), outgoing longwave radiation (second term on rhs), the meridional divergence of atmospheric heat fluxes (third term on rhs) and the atmosphere-to-ocean heat exchange (fourth term on rhs; see also Eq. 2) through:

$$C \frac{\delta T_a}{\delta t} = S(1 - \alpha_p) - \epsilon \sigma T_a{}^4 - \nabla . H_a - H_{a2o} \tag{13}$$

where $\sigma$ is the Stefan Boltzmann constant and $\alpha_p$ the planetary albedo. For the latter we assume a simple sinusoidal profile (Tab. 1). The factor $\epsilon$ in the longwave radiation term of Eq. 13 is traditionally regarded as the atmospheric emissivity. However, as we will see later on, in the context of this AMOC-emulator it is more appropriate to view it as a 'total atmosphere effect', that is a temporal and spatial varying parameter that effectively combines atmospheric emissivity, the greenhouse effect and on top of that all other processes included in a GCM that cause regional temperatures to differ from global temperature changes. In the next section it is described how the 'total atmosphere effect' is used to include GCM specific future changes in regional temperatures into the AMOC-emulator. The third term on the rhs of Eq. 13 describes meridional atmospheric heat fluxes, represented here as a diffusive process:

$$H_a = -C K \frac{\delta T_a}{\delta y} \tag{14}$$





with $y$ the meridional distance and $K$ the meridional atmospheric eddy diffusivity. The latter includes a simple meridional dependency following a sinusoidal profile based on observations of the meridional heat flux and temperatures (Tab. 1). By including meridional atmospheric heat fluxes, we allow for a negative feedback between changes in North Atlantic sea-surface temperatures and atmospheric meridional heat transport. As the AMOC weakens, the North Atlantic cools while the low-latitude Atlantic warms, increasing the meridional temperature contrast that in turn leads to an increase in meridional atmospheric heat transport towards the North Atlantic.

We performed the integration of the set of equations using a simple Euler forward sceme and an asynchronous coupling of the energy balance model and the AMOC box model, with timesteps of 7 days and 28 days respectively.

The AMOC-emulator includes a number of constants and parameters. In Tab. 1 all constants and prescribed parameter values are given while the tunable parameters are discussed in Sect. 2.2. We acknowledge that some of the prescribed parameter values are uncertain, however, the behavior of the AMOC-emulators is dominated by the tuning parameters.

## 2.2 AMOC-emulator parameter tuning

A number of parameters determine the AMOC-emulator's sensitivity to changes in regional freshwater fluxes and temperatures, and its response timescales. These parameters allow tuning of the AMOC-emulator to reproduce AMOC evolutions simulated by a specific GCM. We choose seven parameters: $k$, $\Gamma$, $F_1$, $F_2$, $h_1$, $h_2$ and $V_4$. Note that the volume of the deep Atlantic box ($V_4$) is viewed as a tuning parameter because it represents the unknown and model-dependent portion of the deep ocean that is involved in transporting North Atlantic Deep Water southwards. This selection of parameters is somewhat subjective, but it proved a good balance between, on the one hand, sufficient degrees of freedom to tune the AMOC emulator's behavior towards that of a specific GCM and, on the other hand, the efficiency to find an optimal parameter fit.

The parameters are tuned by minimizing a cost function $C$ that describes the misfit between the AMOC evolution simulated by a specific GCM and the AMOC-emulator:

$$C = \sum_{s=1}^{s=end} \sum_{t=1}^{t=end} (m_{emu} - m_{gcm})^2 \tag{15}$$

with $m_{emu}$ the AMOC time-series simulated by the AMOC-emulator for forcing scenario $s$ and $m_{gcm}$ the AMOC time-series simulated by the GCM for that same forcing scenario. This allows us to include a number of different climate simulations that are driven by different forcings into the tuning of a single AMOC-emulator.

The tuning procedure follows a SA algorithm (Lombardi, 2015), a probabilistic technique to approximate the global optimum of a given function (Fig. 2). In short, it generates a candidate parameter set by applying a random perturbation to the parameter set of the previous iteration, and subsequently accepts or declines this new parameter set based on a stochastic mechanism. An example of how the parameters evolve through the SA algorith is given in Figure 3. In detail the method works as follows:



1. Start by picking random initial values $p_i$ for the different parameters for the first iteration from a uniform distribution $U$:

$$p_i = U(p_{start}(1-z), p_{start}(1+z)) \tag{16}$$

with $p_{start}$ the start values of the tuning parameters and $z$ a multiplication constant. For $p_{start}$ we use the values found by Zickfeld et al. (2004) and $z = 2$; implying that the initial values are randomly selected from the start values $\pm 200\%$. Step 1 is repeated until all parameter values are within the bounds given in Tab. 2.

2. For this set of initial parameters the AMOC-emulator is used to simulate all scenarios for which GCM output is available and the initial cost function $C_i$ (Eq. 15) is calculated.

3. In the first step of the iterative part of the SA algorithm, a candidate parameter set $p_c$ is obtained analogues to step 1:

$$p_c = U(p_{it-1} - \psi_{it}p_{it-1}, p_{it-1} + \psi_{it}p_{it-1}) \tag{17}$$

Note that for the first iteration the initial values are used instead of the values from the previous iteration step. In contrast to step 1, the multiplication factor $\psi_{it}$ for iteration $it$, the so-called 'SA-temperature', is chosen here to be a function of the magnitude of the cost function. This allows for a more accurate tuning as the global optimum is approached:

$$\psi_{it} = min(\psi_{max}, max(\psi_{min}, a10^b \log_{10}(C_{it})^c)) \tag{18}$$

This formulation enables the algorithm to test candidate parameter values that are far away from the current parameter values during the first stage of the tuning process ($\psi_{max}$=0.2 gives maximum 20% change in parameter values), while as the global optimum is approach, only local candidate parameters are tested that are close to the current parameter values ($\psi_{min}$=0.01 gives minimum 1% change in parameter values). For constant $a$, $b$ and $c$ we used 2.9, -4 and 3.36 respectively. The efficiency of the tuning procedure is fairly sensitive to the choices of parameters $\psi_{min}$, $\psi_{max}$, $a$, $b$ and $c$. However, the resulting GCM-based AMOC-emulator parameter sets are not. Step 3 is repeated until all values are within the bounds given in Tab. 2.

4. The AMOC-emulator then uses the candidate parameter sets $p_c$ to simulate the AMOC evolution and calculate a candidate cost function $C_c$.

5. Whether or not the candidate parameter set $p_c$ is kept is determined by a stochastic mechanism that enables the algorithm to 'escape' from a local minimum in the cost function. Figure 3 provides an example of how the cost function increases on several iteration steps in order to escape local minima. The stochastic mechanism works as follows:



if $C_c < C_{it-1}$   then $\boldsymbol{p_{it} = p_c}$

if $C_c => C_{it-1}$ then:

      if $U(0,1) < A$ then $\boldsymbol{p_{it} = p_c}$ else $\boldsymbol{p_{it} = p_{it-1}}$

with $U(0,1)$ a random draw from a uniform distribution between 0 and 1. The acceptance criterion $A$ is determined by the SA-temperature $\psi_{it}$ and the SA-temperature maximum $\psi_{max}$:

$$A = \psi_{it} A_i \psi_{max}^{-1}.$$

For the initial acceptance criterion value $A_i$ we use 0.6.

6. Steps 3-5 are repeated until a stopping criterion is satisfied. We stop the procedure when significant improvements are no longer made. More specifically when the linear trend in $C$ over the last $n$ iterations falls below a critical slope $s_c$. We used $n$=50 and $s_c$=0.05.

Because of non-linearities in the AMOC-emulator, there is more than one parameter set that provides a good fit between the AMOC evolution in the AMOC-emulator and a specific GCM. Therefore, we repeat the SA algorithm until a total of 100 reasonable fits are found and select from this the 10 parameter sets with the smallest corresponding cost function. A good maximum cost function for a reasonable fit was found to be $5*10^4 (\mathrm{m^3 s^{-1}})^2$.

## 2.3 AMOC-emulator forcings

The AMOC-emulator is designed to be forced with regional temperature changes and GIS melt provided by a specific GCM simulation. Regional temperature changes are prescribed in the AMOC-emulator through the 'total atmosphere effect' (see also Sec. 2.1). Using Eq. 13, assuming steady state and no atmosphere-ocean heat exchange we can use the individual GCM temperature time series ($\boldsymbol{T_{gcm}}$) from a specific region $r$, and for the historical period or an RCP scenario, to solve for the time evolution of $\epsilon$:

$$\epsilon = \frac{\boldsymbol{S}(1 - \boldsymbol{\alpha_p}) - \triangledown\left(-C\boldsymbol{K}\frac{\delta \boldsymbol{T_{gcm}}}{\delta y}\right)}{\sigma \boldsymbol{T_{gcm}}^4} \tag{19}$$

Through this method we ensure that the regional surface temperature forcings of the AMOC-emulator for all atmospheric boxes closely resembles that of the GCM. We acknowledge that simulated regional surface temperatures of the AMOC-emulator forced by prescribed changes in 'total atmosphere effect', will diverge somewhat from the GCM time series when subsequently atmosphere-ocean heat exchange is included, allowing the AMOC to redistribute heat from the South Atlantic ($T_{a1}$) to the North Atlantic ($T_{a3}$), but the effect on regional temperatures is small, with a maximum impact of ~0.1K in the North Atlantic in the UVic-based temperatures used in this study (Fig. 4). We will use regional surface temperatures averaged





over all longitudes in a specific latitude band as specified by the latitudinal extents of the different boxes in the atmospheric component of the AMOC-emulator. Note that the temperature forcing files need to be interpolated onto the temporal resolution used in the atmospheric component of the AMOC-emulator.

GIS melt is included as an annual mean sum of all GIS melt that is introduced into the GCM ($F_{gis}$ in Eq. 9), the methodology
that we used to project future GIS melt is provided in the next section. Note that the GIS melt forcing needs to be interpolated onto the temporal resolution used in the ocean component of the AMOC-emulator.

### 2.4 UVic simulations

To investigate the effectiveness of the tuning procedure and the predictive power of the resulting AMOC-emulator we conducted a number of climate projections and sensitivity experiments with the University of Victoria (UVic) Earth System Climate Model
version 2.9 (Weaver et al., 2001). This model of intermediate complexity includes a three-dimensional dynamical ocean with 19 vertical levels at $3.6°$x$1.8°$ horizontal resolution governed by the primitive equations, coupled to a two-dimensional single-level atmosphere, with moisture and heat balances and fluxes between the two mediums, and a dynamical sea ice model.

Five experiments are used to tune the AMOC-emulator, one following historical GHG changes (1850-2006; referred to as Historical) and four following RCP4.5 and RCP8.5 (Representative Concentration Pathways; Meinshausen et al., 2011) and
including or excluding enhanced melting of the GIS (period 2006-2300; experiments are referred to as RCP4.5, RCP4.5-GIS, RCP8.5 and RCP8.5-GIS respectively; see Tab. 3). In these simulations only changes in GHGs and GIS melt are considered, all other boundary conditions are fixed at pre-industrial levels.

The UVic-based regional annual mean atmospheric temperature forcings are shown in Fig. 4 and available in the supplement. The UVic-based surface temperature evolution exibits multi-decadal to centennial oscillations that result from global climate
variability originating from the Southern Ocean, oscillations that the AMOC-emulator translates into AMOC variations as will become clear in Sect. 3.1.

The GIS melt scenarios used in the UVic-based AMOC-emulator simulations are based on the methodology of Lenaerts et al. (2015). Using a high-resolution regional atmospheric climate model for Greenland (RACMO2), forced by a future projection simulated with the HadGEM2-ES GCM following RCP4.5, they describe a strong correlation between local runoff and 500hPa
summer (JJA) temperature changes over different regions of Greenland. We have used this relationships in combination with CMIP5 mean 500hPa summer atmospheric temperature changes to construct GIS melt projections for RCP4.5 and RCP8.5. A seasonal cycle was prescribed in these GIS melt scenarios following Lenaerts et al. (2015). The resulting GIS melt scenarios show a strong increase in GIS melt with annual mean values of ~0.02Sv and ~0.08Sv ($1\mathrm{Sv} = 10^6\mathrm{m}^3\mathrm{s}^{-1}$) in the year 2300 for RCP4.5 and RCP8.5 respectively with summer maxima almost an order of magnitude larger than the annual mean values
(Fig. 5). Note that the runoff projections of Lenaerts et al. (2015) include simulated changes in evaporation, precipitation, snow melt and ice melt. Therefore Greenland runoff calculated within UVic is fixed at the simulated 1970-2001 average values to avoid double counting meltwater flux changes. Moreover, this methodology neglects future changes in GIS solid ice discharge. This choice is motivated by the large uncertainty in the observed and projected sensitivity of GIS solid ice discharge to climate



warming, in the magnitude and even the sign of the future GIS ice discharge contribution to sea-level rise (Nick et al., 2013; Lenaerts et al., 2015; Vizcaino et al., 2015). The GIS melt forcing used in this manuscript is available in the supplement.

## 3 Results & Discussion

### 3.1 UVic-based AMOC-emulator

Five UVic simulations are used in the tuning procedure of the AMOC emulator (Tab. 3). An example of how the cost function, SA-temperature and the tuning parameters of the emulator evolve during the SA tuning process is given in Figure 3 and the parameter ranges of the ten best UVic-based AMOC-emulators values are listed in Tab. 2. In this study we use the maximum meridional overturning streamfunction at 26°N below 500m depth as a measure of the AMOC strength, but note that the AMOC-emulator allows one to chose another measure of the AMOC strength.

Scatter plots between the individual tuning parameters (Fig. 6) reveal some interesting aspects of the AMOC emulator. There is a strong relation between the hydraulic constant $k$ and the thermal coupling constant $\Gamma$, with a larger hydraulic constant leading to a smaller thermal coupling constant. Furthermore, in accordance with Equation 12 there is a quasi-linear relation between $F_1$ and $h_1$ and also between $F_2$ and $h_2$. Finally, it shows that certain parameters are better constrained (i.e. of larger importance), show more clustering of the values with a low cost function (blue colors in Figure 6), for instance $k$, $\Gamma$, $F_1$, $V_4$

and $h_1$, while the opposite is true for $F_2$ and $h_2$. The smaller importance of $F_2$ and $h_2$ also follows from their absence in the steady-state solutions of equations 3–10, as already noted by Zickfeld et al. (2004).

  The ten best UVic-based AMOC-emulators are able to reproduce the overall characteristics of the AMOC in UVic in terms of its sensitivity to changes in heat and freshwater forcing (Fig. 7). Both UVic and the UVic-based AMOC-emulators show a decline of the AMOC of ~1Sv over the historical period and a further weakening of ~5Sv and ~9Sv in RCP4.5 and RCP8.5,

respectively. The impact of GIS melt appears small for both RCP4.5-GISmelt and RCP8.5-GISmelt (~1Sv). The ten different AMOC emulators are also fairly consistent with each other. However, there are also differences between the AMOC evolution simulated by UVic and with the UVic-based AMOC-emulators. Most notably is that the UVic-based AMOC-emulator misses the partial AMOC recovery of ~1Sv after 2150 in RCP4.5 and RCP4.5-GISmelt (resulting in root-mean-square-errors, RSME, of $0.88\pm0.08(\mathrm{m^3s^{-1}})^2$ and $0.78\pm0.07(\mathrm{m^3s^{-1}})^2$ respectively; $\mu\pm\sigma$), appears slightly too sensitive to the GIS melt forcing

in simulation RCP8.5-GIS (resulting in the largest RSME of the four simulations of $0.9\pm0.13(\mathrm{m^3s^{-1}})^2$) and simulates a too direct response of the AMOC strength to multi-decadal surface temperature oscillations.

  It is, however, to be expected that a box-model does not completely capture the behavior of the AMOC as simulated with a higher order climate model. It is also worth noting that the fit for an individual simulation could be improved, for instance the AMOC-emulator does allow for a partial AMOC recovery as UVic shows for RCP4.5, but such an AMOC-emulator is not

found through the SA tuning methodology in this example, because it would degrade the fit for the other scenarios and thus lead to an overall higher cost function.

  The presented AMOC-emulator is an adjusted version of the model by Zickfeld et al. (2004) and includes a 1D atmospheric energy balance model to represent the Bjerknes feedback. To investigate the importance of this first-order atmospheric



meridional heat transport feedback on the AMOC strength evolution, experiments have been performed with and without this feedback (Fig. 9), by adjusting the value of $K$ in Equation 14. The impact of including atmospheric meridional heat transport is a small, but non-negligible ~1Sv strengthening of the control state of the AMOC (not shown) and, more importantly, a slightly lower sensitivity to changes in radiative forcing and GIS melt (Fig. 9). This confirms our understanding of atmospheric meridional heat transport acting as a negative feedback to AMOC changes. The simulations with the atmospheric feedback included have on average a stronger AMOC by 8.1±1.9% ($\mu \pm \sigma$; calculated over all 10 best fits and over all five forcing scenarios).

## 3.2 Predictive power of the UVic-based AMOC-emulator

The value of constructing an AMOC-emulator is in its capability to perform a large number of simulations that allows quantification of uncertainties for a large range of forcings. This raises the question of the fidelity of the AMOC-emulator and errors induced by its use. Here we will assess the predictive capabilities and errors of the AMOC-emulator by comparing it with results from four additional transient UVic climate change simulations, none of which have been used in the AMOC-emulator tuning procedure.

The four independent transient climate change simulations follow idealized scenarios with changes in either the GHG or GIS melt forcing. In two of the simulations RCP8.5-GIS is adjusted by multiplying the GHG forcing changes with respect to the 2006 value by a constant factor of 0.5 (RCP8.5x0.5-GIS) and 1.5 (RCP8.5x1.5-GIS). In the third independent simulation the GIS melt forcing of the original RCP8.5-GIS scenario is multiplied by 1.5 (RCP8.5-GISx1.5). The fourth simulation uses the RCP4.5 GHG forcing but combines it with the RCP8.5-GISx1.5 GIS melt forcing (RCP4.5-GISRCP8.5x1.5; see Tab. 3). Some of these idealized scenarios are unlikely to occur in reality, for instance RCP4.5-GISRCP8.5x1.5, which implies only limited global warming combined with a very large melt rate of the GIS, but these scenarios allow us to assess the bounds within which the AMOC-emulator has predictive power.

The results from these experiments are shown as anomalies relative to the original scenario, meaning RCP8.5-GIS for RCP8.5x0.5-GIS, RCP8.5x1.5-GIS and RCP8.5-GISx1.5, and RCP4.5-GIS for RCP4.5-GISRCP8.5x1.5. We find that for a 50% changes in the GHG forcing the UVic-based AMOC-emulators are well capable of predicting the AMOC evolution of UVic (upper panels Fig. 8; RMSE of $0.73\pm0.11(\mathrm{m^3s^{-1}})^2$ and $0.82\pm0.14(\mathrm{m^3s^{-1}})^2$ in RCP8.5x0.5-GIS and RCP8.5x1.5-GIS respectively, compared to $0.9\pm0.12(\mathrm{m^3s^{-1}})^2$ in RCP8.5-GIS). For a large increase in GIS melt under a low GHG scenario (RCP4.5-GISRCP-8.5x1.5) the RMSE ($0.92\pm0.11(\mathrm{m^3s^{-1}})^2$) increases somewhat compared to the original RCP4.5-GIS RMSE ($0.78\pm0.07(\mathrm{m^3s^{-1}})^2$), but it is for the most extreme GIS melt scenario RCP8.5-GISx1.5 that the RMSE increases from $0.90\pm0.13(\mathrm{m^3s^{-1}})^2$ to $1.6\pm0.22(\mathrm{m^3s^{-1}})^2$. The latter shows that the UVic-based AMOC-emulators tend to overestimate the impact of GIS melt on the AMOC strength under high-end GHG scenarios.

Overall, the predictive power of the AMOC-emulator is good for reasonable forcing scenarios when one considers the simplicity of the model. For forcing scenarios that are increasingly far away from the forcings that are used in tuning the AMOC-emulator, the predictive power decreases. Note also that an AMOC-emulator that is tuned to specific GIS melt experiments is likely not applicable to experiments in which meltwater is applied to a different geographical region or with a different seasonal cycle. This is not to say that the presented methodology and tuning procedure can not equally be applied to



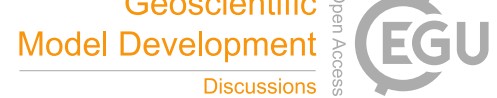

other sources of meltwater input.

## 4   Summary

We have presented an AMOC-emulator that can be used to assess the uncertainties in AMOC projections produced by GCMs
under future changes in surface temperatures and freshwater input. In this study we focused on enhanced GIS melt as a source
of additional freshwater. Following a SA algorithm, the AMOC-emulator can be tuned using a limited number of GCM pro-
jections of future climate change. We test our methodology by tuning the AMOC-emulator towards the AMOC evolution
simulated with the UVic model for the historical period and four future projections up to the year 2300 including increases in
GHG concentrations and GIS melt. The results show that the UVic-based AMOC-emulator captures well the overall character-
istics of the multi-centennial response of the AMOC to changes in regional temperatures and GIS melt. This is confirmed by
testing the predictive power of the AMOC emulator for a number of independent UVic simulations that are not used in the tun-
ing procedure. The AMOC-emulator is a potentially valuable tool to study the uncertainty in GCM-based AMOC projections.

### Code availability

The MATLAB model code of M-AMOC1.0 and UVic-based example forcing files are available in the supplement.

*Acknowledgements.*   This work was supported by a grant from the National Oceanographic and Atmospheric Administration (award number
NA15OAR4310239).



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





**Table 1.** Prescribed parameter values of the AMOC-emulator. The volumes of the oceanic boxes 1-3 and the depths of all oceanic boxes are following Zickfeld et al. (2004). $\bar{\psi}_i$ denotes the average latitude of box $i$.

| Symbol | Value | Unit | Description |
|---|---|---|---|
| $\sigma$ | $5.67\ 10^{-8}$ | $Wm^{-2}K^{-4}$ | Stefan Bolzmann constant |
| $\rho_0$ | 1025 | $kgm^{-3}$ | Density of seawater |
| $\rho_{a0}$ | 1.005 | $kgm^{-3}$ | Density of air |
| $c_o$ | 4000 | $Jkg^{-1}K^{-1}$ | Specific heat of seawater |
| $c_a$ | 1000 | $Jkg^{-1}K^{-1}$ | Specific heat of air |
| $S_0$ | 35 | psu | Reference salinity |
| $V_1$ | $1.1\ 10^{17}$ | $m^3$ | Volume of Southern box |
| $V_2$ | $0.68\ 10^{17}$ | $m^3$ | Volume of low-latitude box |
| $V_3$ | $0.4\ 10^{17}$ | $m^3$ | Volume of Northern box |
| $z_1$ | 3000 | m | Depth of Southern box |
| $z_2$ | 1000 | m | Depth of low-latitude box |
| $z_3$ | 3000 | m | Depth of Northern box |
| $h$ | 8400 | m | Height of atmosphere |
| $C$ | $\rho_{a0}c_a h = 8.4 10^{-8}$ | $Jm^{-2}K^{-1}$ | Atmospheric heat capacity |
| $S_i$ | $S_i = 295 + 125\cos 2\bar{\psi}_i$ | $Wm^{-2}$ | Latitude dependend incoming SW |
| $\alpha_{ip}$ | $\alpha_{ip} = 0.6 - 0.4\cos\bar{\psi}_i$ | - | Latitude dependend planetary albedo |
| $K_i$ | $K_i = 2.10^5 \cos\bar{\psi}_i$ | $m^2 s^{-1}$ | Latitude dependend meridional atmospheric eddy diffusivity |

**Table 2.** AMOC-emulator parameter ranges based on the 10 best fits and the bounds that are applied in the fitting procedure.

| | Hydraulic constant $k$ | Thermal coupling constant ($\Gamma$) | Southern to low-latitude freshwater flux ($F_1$) | Low-latitude to Northern freshwater flux ($F_2$) | Southern to low-latitude hydrological sensitivity ($h_1$) | Low-latitude to Northern hydrological sensitivity ($h_2$) | Volume of deep Atlantic box ($V_4$) |
|---|---|---|---|---|---|---|---|
| Unit | $10^{18}m^3 s^{-1}$ | $10^8 Jm^{-2}s^{-1}K^{-1}$ | Sv | Sv | $SvK^{-1}$ | $SvK^{-1}$ | $10^{14}m^3$ |
| Range | 1.12 to 2.07 | 6.89 to 16.61 | 0.0006 to 0.0154 | -0.23 to 0.02 | 0.001 to 0.009 | -0.022 to 0.002 | 8.3 to 55.0 |
| Bounds | $k>0$ | $\Gamma > 0$ | - | - | - | - | $V_4 > 0$ |





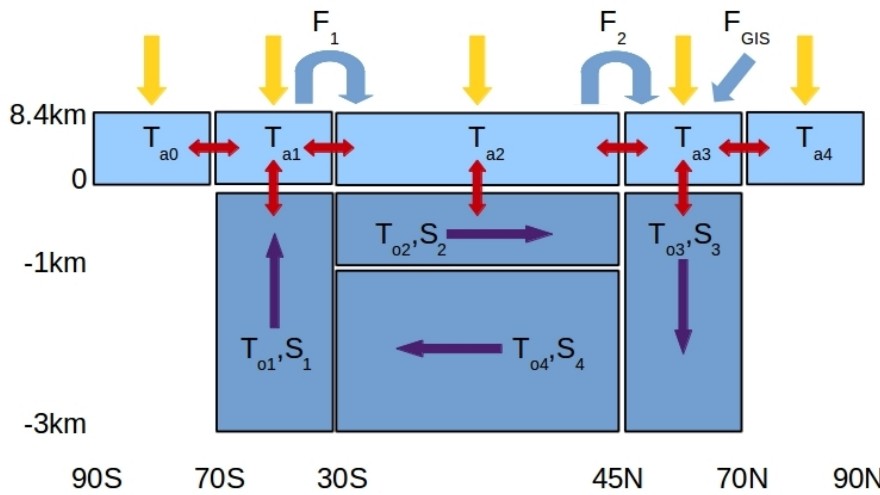

**Figure 1.** Outline of the AMOC-emulator. The five atmospheric boxes in light blue and the four oceanic boxes in dark blue and their respective spatial dimensions. The arrows indicate the direction of advection of heat and salt by the AMOC (purple), heat exchange by the atmosphere (red), freshwater exchange ($F_1$ and $F_2$; light blue), Greenland Ice Sheet melt ($F_{GIS}$; light blue) and radiative forcing (yellow). The capital letters denote the state variable(s) of the different boxes and the subscript the box number and atmosphere or ocean in case of temperatures.

**Table 3.** Experimental design of UVic simulations. The GHG scenarios refer to the scenarios described by Meinshausen et al. (2011). The last column describes if the simulation is used to tune the UVic-based AMOC-emulator or if it is used as an independent check on the predictive power of the AMOC-emulator.

| Experiment name | GHG | GIS melt | Initial conditions | Tuning |
|---|---|---|---|---|
| Historical | Historical | no | 2ky PI | yes |
| RCP4.5 | RCP4.5 | no | 2006 Historical | yes |
| RCP4.5-GIS | RCP4.5 | RCP4.5 | 2006 Historical | yes |
| RCP8.5 | RCP8.5 | no | 2006 Historical | yes |
| RCP8.5-GIS | RCP8.5 | RCP8.5 | 2006 Historical | yes |
| RCP8.5x0.5-GIS | RCP8.5x0.5 | RCP8.5 | 2006 Historical | no |
| RCP8.5x1.5-GIS | RCP8.5x1.5 | RCP8.5 | 2006 Historical | no |
| RCP8.5-GISx1.5 | RCP8.5 | RCP8.5x1.5 | 2006 Historical | no |
| RCP4.5-GISRCP8.5x1.5 | RCP4.5 | RCP8.5x1.5 | 2006 Historical | no |





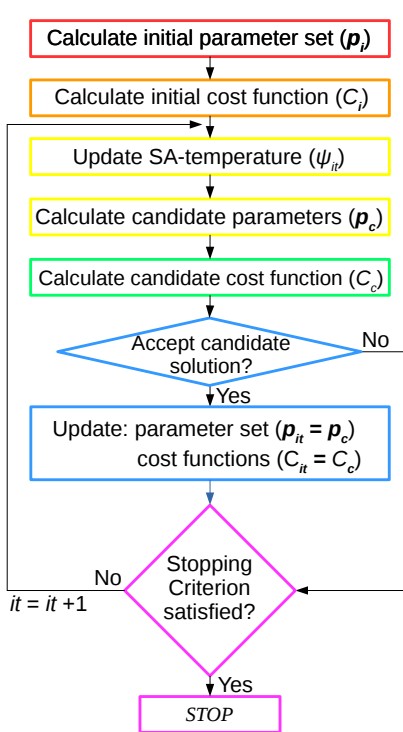

**Figure 2.** Flow chart for Simulated Annealing algorithm. Bold symbols indicate arrays of *n*=7 parameters and different colors reference to the different steps in Section 2.2.





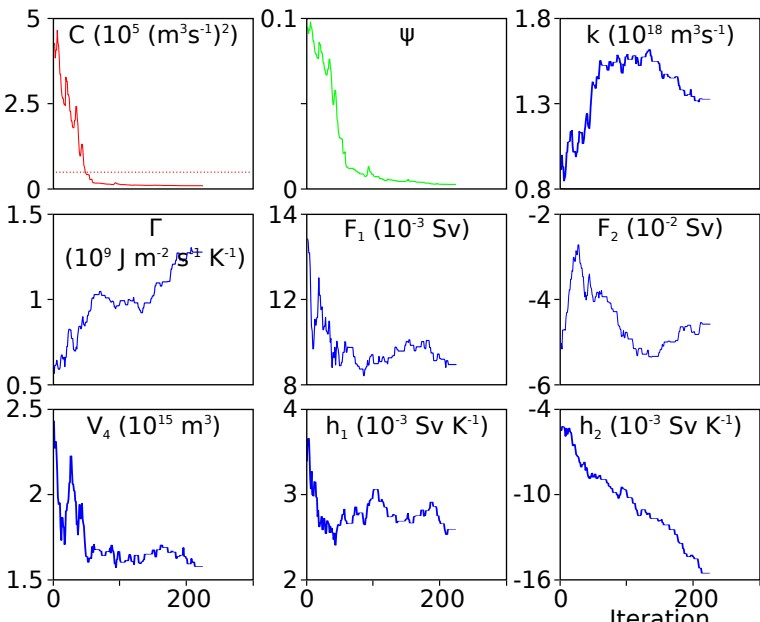

**Figure 3.** Example of the changes as a function of the iteration step in SA-algorithm cost function (C), SA-temperature ($\psi$) and the seven tuning parameters of the AMOC-emulator: hydraulic constant $k$, thermal coupling constant $\Gamma$, southern to low-latitude freshwater flux $F_1$, low-latitude to northern freshwater flux $F_2$, volume of deep Atlantic box $V_4$, southern to low-latitude hydrological sensitivity $h_1$ and low-latitude to northern hydrological sensitivity $h_2$. The horizontal dotted line in the top left panel shows the imposed maximum cost function for a reasonable fit.

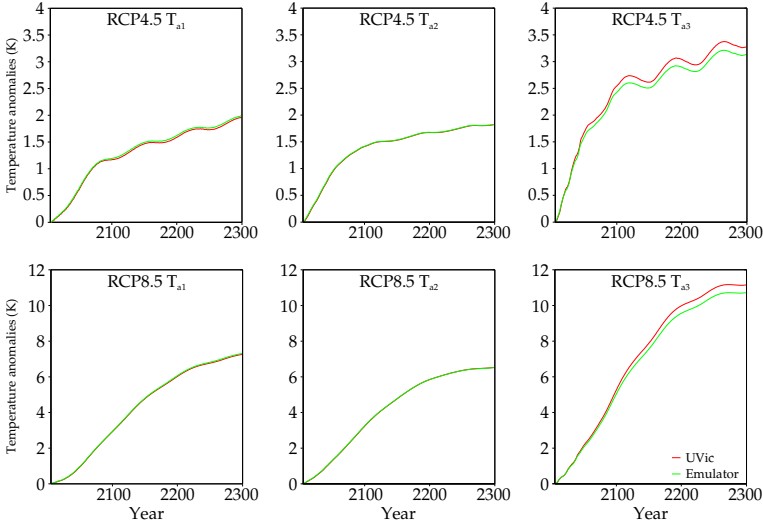

**Figure 4.** UVic (red) regional atmospheric temperature anomalies (relatvie to 2006; K) and the regional temperature forcings used in the AMOC-emulator by tuning the 'total atmosphere effect' (green). Results are given for the historical period, RCP4.5-nomelt and RCP8.5-nomelt and for the boxes 1-3.





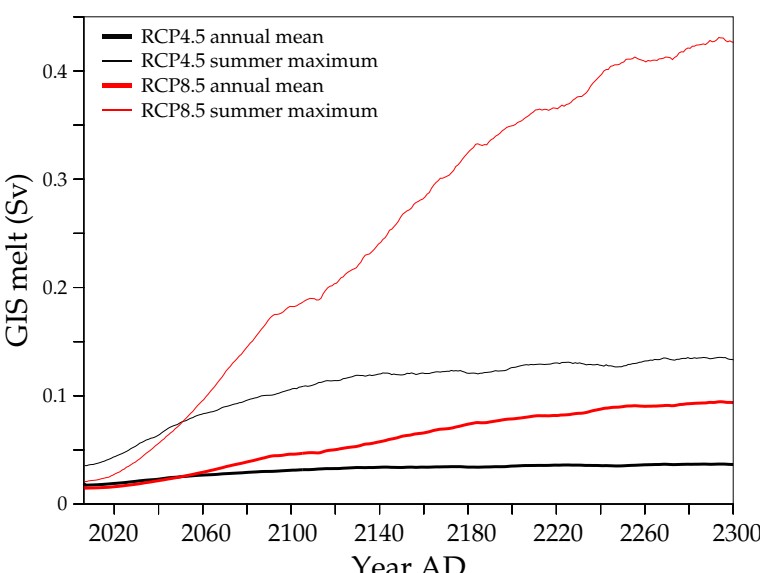

**Figure 5.** Imposed GIS melt forcing (Sv) for RCP4.5 (black) and RCP8.5 (red) for the period 2006-2300 based on the methodology of Lenaerts et al. (2015). Included are the annual mean values (thick lines) and summer maximum (July) GIS runoff (thin lines).





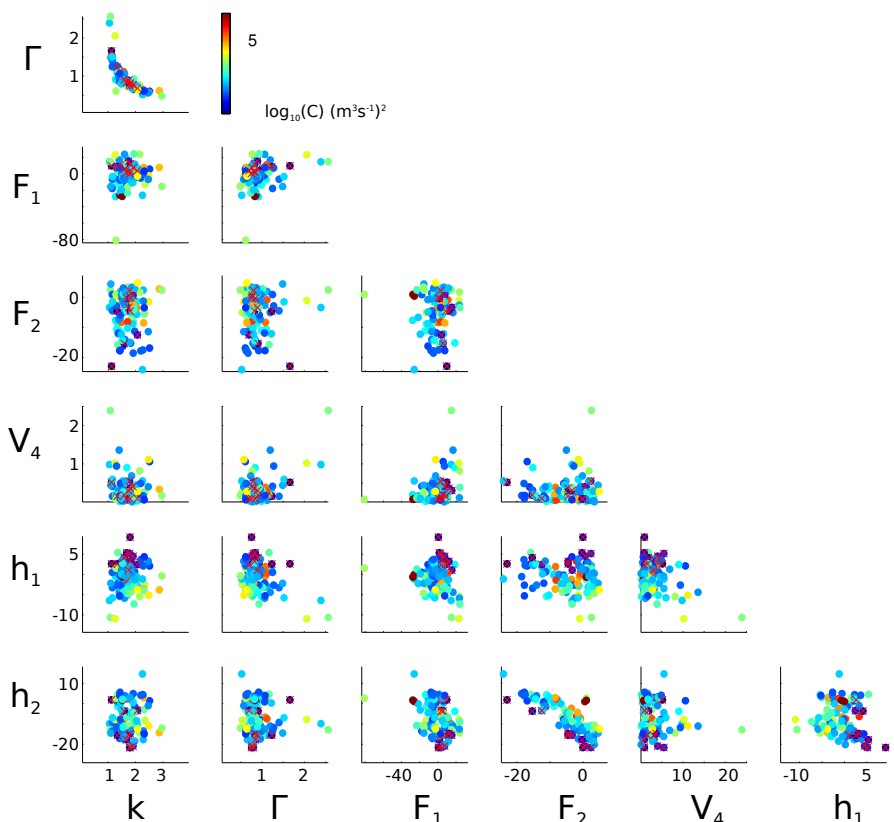

**Figure 6.** Scatter plot of the seven individual tuning parameters for all 100 reasonable fits. Colors indicate cost function value and red crosses the 10 best fits. For the units of the different parameters see Figure 3.



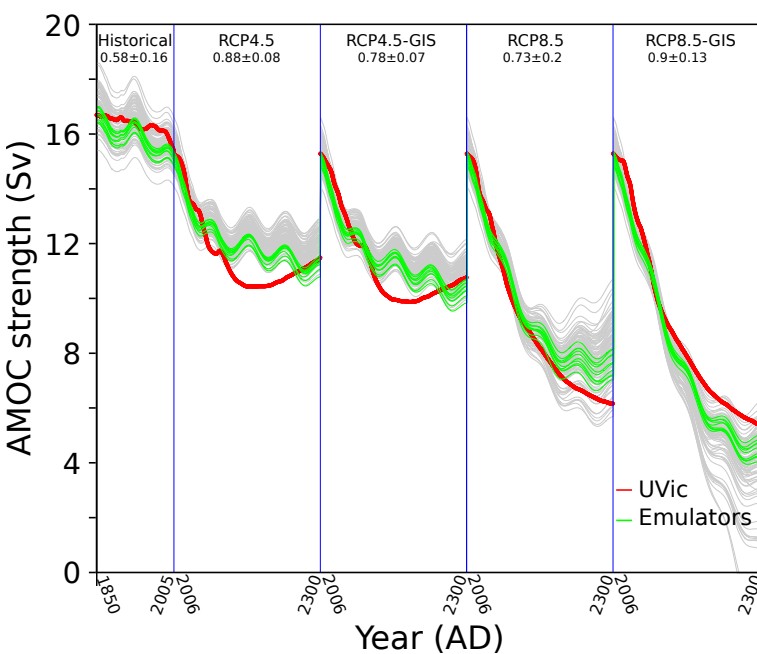

**Figure 7.** Evolution of the AMOC strength (Sv; maximum meridional overturning streamfunction at 26°N below 500m depth) for the historical period, RCP4.5-nomelt, RCP4.5-GISmelt, RCP8.5-nomelt and RCP8.5-GISmelt (left to right) from UVic (red) and the 10 best AMOC-emulator fits to the UVic results (green). For reference the remaining 'reasonable' fits are shown in grey. As an indication of the goodness-of-fit between UVic and the UVic-based AMOC-emulator, RSMEs $((m^3 s^{-1})^2)$ are given in the top of the panels for all 10 AMOC-emulator fits combined ($\mu \pm \sigma$).





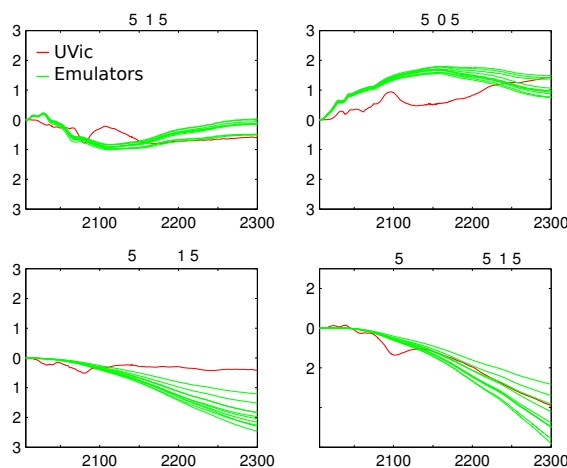

**Figure 8.** Comparison of simulated AMOC strength anomalies (Sv) by UVic (red) and the 10 best UVic-based AMOC-emulators (green) for the four forcing scenarios that have not been used in the tuning procedure: top middle RCP8.5x1.5-GISmelt, top right RCP8.5x0.5-GISmelt, bottom left RCP8.5-GISmeltx1.5 and bottom middle RCP4.5-GISmeltRCP8.5x1.5. Anomalies are calculate with respect to the original experiments, that is RCP8.5-GISmelt for the top two panels and the lower left, and RCP4.5-GISmelt for the results in the lower rigth panel.



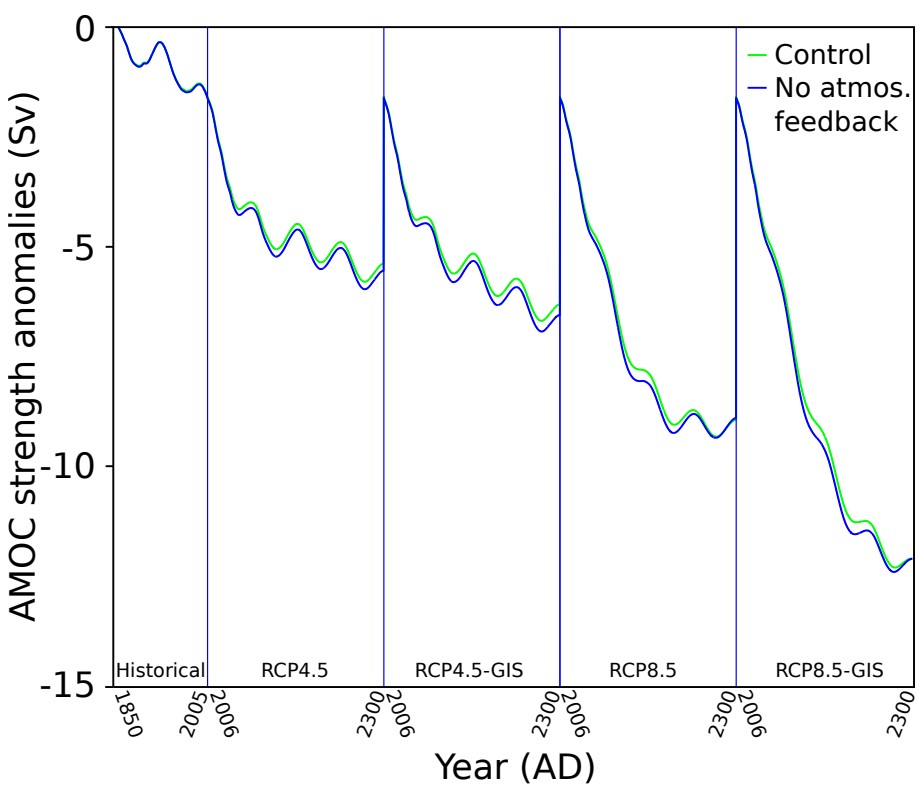

**Figure 9.** Comparison of simulated AMOC strength anomalies (Sv) by the AMOC-emulator with (green) or without (blue) inclusion of meridional atmospheric heat transport. Results are given for the average over the 10 best UVic-based AMOC-emulators and for the historical period, RCP4.5-nomelt, RCP4.5-GISmelt, RCP8.5-nomelt and RCP8.5-GISmelt (left to right).