# Peer review of "AMOC-emulator M-AMOC1.0 for uncertainty assessment of future projections"

_Geoscientific Model Development, 2016_

## Short Comment (SC1) · 9 Jun 2016

This paper develops an AMOC emulator to mimic the results of a small ensemble of runs from a given climate model. After discussing the method and showing how the emulator can represent the main aspects of key variables in the ensemble, the authors show how their simple tool can then be used to examine additional aspects of the parameter space that might be hard to cover with a more computationally expensive climate model. They also evaluate their approach using additional runs used for the training ensemble.

This is a well written and clear paper that is easy to follow. The need for such a tool is well justified. Figures are clear. Evaluation for the case presented is strong. Thus,

I think this paper is appropriate for the literature, and this journal. I would recommend acceptance with only a small number of minor revisions.

page 2, Line 18: Would the approach work in a GCM whose AMOC is controlled by the strength of southern hemisphere winds? This is a potential major caveat, that might be worth discussing more at the end of the paper when the authors expand on applications.

page 4, line 12: Is it reasonable to assume the wind-driven oceanic meridional freshwater transport is static. I might expect it to evolve with an evolving climate, especially as high latitude freshwater sources (e.g. GIS runoff) changes.

Page 5, line 30: algorithm misspelled.

page 6, line 4: Why fixed limits of plus/minus 200%? Why not something based on each variable's variability/range?

page 6, line 8 - analogous misspelled.

Summary: Way too short. I'd like to see more discussion on how such an emulator could be improved, its limitations, questions it might be best for, etc.

Table 1: For last 3 entries, dependent misspelled

Figure 1: Why is the atmospheric part squished so much in the vertical compared to the oceanic component?

Figure 4 caption: relative misspelled.

Figure 8 caption: right misspelled.

---

## Referee Comment (RC1) · Anonymous Referee #1 · 14 Jun 2016

The authors introduce a new box-model based emulator of the Atlantic meridional over-turning circulation, M-AMOC1.0, to the community. I agree entirely with the motivation for doing so. Uncertainty in future climate projections and the responses of GCMs to them due to parameter uncertainty are key challenges in the field of climate science. The paper is well written, and easy to follow. Figures and tables are sufficient and clear (some typesetting issues aside).

I am rather underwhelmed by the performance of M-AMOC1.0 as presented here. It does not capture the response of the GCM sufficiently well that I would be confident in the results. As it currently stands neither the uncertainty in the emulator itself or its ability to assess uncertainty in AMOC projections are explored rigorously.

However, I find the idea interesting and I am sympathetic to the motivation. I would like to see the emulator published, but only subject to major revisions.

Major comments:-

I don't understand the reasoning for why you generate 100 reasonable fits and then select only the 10 best fits (P7 L15). Firstly, this emulator/box-model should be cheap to run, so why choose such small numbers? Surely ensembles of order 10,000 or 100,000 are more reasonable. Secondly, the choice of 10 best fits seems to narrow the ranges of several parameters (e.g. V4, F1, h1, $\Gamma$). By doing this you rule out large regions of parameter space that give perfectly reasonable fits, and could behave differently under different forcing scenarios. If the primary aim is to assess uncertainty in AMOC projections I would expect to see a rigorous analysis of the uncertainty. By discarding large areas of parameter space uncertainty will certainly be underestimated.

In the four scenarios not seen by the emulator (Fig. 8) the behavior of UVic is clearly not captured in two cases (lower left and upper right). There are no confidence intervals plotted (or computed as far as I can tell), but I believe the GCM would lie well outside 2 standard errors in those two cases. Therefore, the GCM would still need to be run for any untested scenario. I would not trust the emulator in its current form.

On multidecadal timescales the emulator is plagued by sensitivity to surface temperature oscillations. These seem to have arisen from the addition of the atmospheric boxes to the ocean box model published by Zickfeld et al., 2004. Can the authors confirm that this is the case, and if so can they control this sensitivity, e.g. by introducing a damping/mixing term?

If the authors have good reason to retain this behavior they need to test the sensitivity to the phase of the variability. For all of the scenarios, the chosen start date (2006) appears to be shortly after a peak in the strong multidecadal variability, so the AMOC is preconditioned to decline at this time. Under all scenarios the AMOC in the 'best' emulators appear to decline faster than the UVic model. Consequently, the SA tuning

and the cost function used may be adversely affected by this multidecadal variability.

On centennial timescales the emulator (as currently presented) does not capture crucial features of the AMOC response to the forcing (Fig. 7). In particular I would draw attention to the RCP4.5 scenarios, in which the GCM exhibits a strong reduction followed by a steady recovery. The emulator fails to identify either the timing or amplitude of the AMOC minimum and it fails to identify the recovery phase. In addition it appears to show signs of a recovery phase under RCP8.5 when UVic shows none. The authors state (P9 L28) that the fit can be improved, but that this would entail a higher overall cost function for the SA tuning method. Is this indicative of a poor choice of cost function? Does it mean that the box model should be tuned separately for each scenario?

A far more substantial summary is required. For example, the emulator's limitations need to be clearly stated (and whether/how the authors think these can be addressed). For what purposes are the emulator suitable in its current form, and for what purposes might it be useful subject to further work? With the current analysis, I disagree with the statement that "...the UVic-based AMOC-emulator captures well the overall characteristics of the multi-centennial response of the AMOC...".

Minor comments:-

Page 3 Line 12: Prescribed FW fluxes: F1 and F2 are tuned parameters. I would have expected these to vary as a function of the forcing/climate. What is the justification for fixing them?

Page 5 Line 10: What you also fail to consider are nonlinearities between these parameters. Co-varying the parameters in Tables 1 and 2 could yield very different behaviours.

Page 5 Line 30: algorith > algorithm

Page 6 Line 4: I find the arbitrary choice of +/- 200% rather strange. What is the

justification for this?

Page 6 Line 8: analogues > analogous

Check typesetting in Tables (e.g. Table 1 column 2)

Table 1: (typo) dependend > dependent

Check typesetting on Figure 8: it appears corrupted.

Figure 4 caption: (typo) relatvie > relative

Figure 8 caption: (typo) calculate > calculated

Figure 8 caption: (typo) rigth > right

[Figure]

---

## Referee Comment (RC2) · Anonymous Referee #2 · 17 Jun 2016

I think the use of emulators has great merits in particular for studying dynamics of large scale systems with substantial uncertainties such as the AMOC. In particular, as CMIP5 simulations miss out on potential important factors influencing future AMOC behaviour such as meltwater influx from Greenland. In this manuscript, Bakker & Schmittner present such an emulator based on previous work e.g. by Zickfeld et al. (2004).

Based on the results presented, however, I have no faith in the emulator to reproduce the AMOC behaviour in more complex models. If this cannot be improved substantially, I regret that cannot recommend a publication of this manuscript. Furthermore, the scientific novelty and relevance for the scientific community is not sufficiently demonstrated. The emulator approach is largely based on Zickfeld et al. (2004) and the importance of own additions (Bjerknes feedback) is not clearly shown. The emulator is then only applied to a single model somewhat contradicting the basic idea of emulator approaches of capturing a range of complex model output. If I'd be looking for an AMOC emulator, I'd like it to reproduce broadly the behaviour of state-of-the-art models.

In addition, the manuscript falls short to address the state of the literature relating to AMOC dynamics, conceptual models, and known short-comings of the Stommel-type model used.

I comment on this in further detail below.

**1 General Comments**

**1.1 Introduction**

The introduction does not sufficiently reflect the state of the literature on AMOC and in particularly not on conceptual models such as the Stommel-Model to study it.

Our understanding of AMOC dynamics has advanced considerably over the last years thanks to ongoing observations e.g. in the Rapid array (see Srokosz & Byrden, Science 2015). In addition a recent studies has suggest that the AMOC might already be in decline (Rahmstorf et al. 2015). While of course not directly relevant for the emulator itself, such observational findings need to be discussed in an approach that emulate AMOC behaviour over the next centuries. This should also include a discussion of atmospheric imprints on the AMOC e.g. such as atmospheric blocking events. It should also allow to assess the performance of GCMs in relation to the observational record.

Much more important though is the discussion in relation to the emulator approach

taken. Stommel type models have been used since quite some time and might be able to capture key dynamics of the AMOC (e.g. bistability). However, they at the same time have faced a lot of criticism and alternative models describing AMOC behaviour exist. This is in particular related to the relevance of Southern Ocean upwelling reflected in a conceptual model by Gnanadesikan (1999) related to changes in the pycnocline depth. A dynamic that is completely missing in the Stommel approach.

This has been explored further in conceptual models and attempts exist to unify pycnocline and freshwater-feedback dynamics. In this context, the authors should consider the work of Sijp et al. (2012) that they may find helpful.

Another question directly relating to the physical plausibility of the Stommel model relates to the relationship of circulation strength and meridional density gradient in a geostrophic ocean. The authors should consider work by Gregoy & Tailleux (2010) that present a kinetic energy approach essential providing a physical explanation for the (empirically supported) meridional density gradient outlining the relevance of the Western Boundary Current in modelling AMOC dynamics.

These comments should not be seen as undermining the Stommel model approach taken here, but they need to be addressed. In short, the authors should show motivate their approach in the light of the most recent literature.

**1.2 The Emulator model**

Here, the work dominantly builds on a previous model by Zickfeld et al. (2004) plus a representation of the Bjerknes feedback. It does however not become sufficiently clear, why this addition will represent a substantial advancement. The authors show the differences in Fig. 9 and describe that this will represent a negative feedback on the AMOC dynamics. But it's not clear, if Figure 9 shows two sets calibrated individually (with and without atmospheric feedback) or just from the optimal parameter set with

this feedback switched on and off. Therefore, I cannot judge if the conclusion drawn by the authors on the importance of the effect are due to their specific parameter set or not.

It would add merit, if the authors could show that the model including the Bjerknes effect will in the end outperform the no-atmospheric feedback model in the fitting procedure. This would also justify, why there model is actually better than the one presented in Zickfeld (2004).

Furthermore, the model includes 5 atmospheric boxes. Why are 5 boxes needed and not 3 to resolve the meridional heat transport? I think that can be easily motivated and maybe I missed it. Maybe it's worth considering to restructure the approach by moving subsection 2.3 further up to discuss the setup of the atmospheric forcings.

In this context, the authors should also reflect on the limitations of the model to re-produce transient AMOC changes that relate to the assumption of well-mixed density within the boxes. This might be in particularly relevant in relation to the Greenland freshwater input. Clearly, this represents an over-simplification and may substantially limit the capabilities of this approach to emulate transient behaviour (I'll further com-ment on this below).

1.3   The tuning to complex model output

In the manuscript, the model is tuned to an EMIC model UVIC. I think that's generally no problem, but somehow contradicts the initial claims by the authors that this emu-lator could now be used to run larger ensembles. What is it exactly that the emulator provides that cannot be done with an EMIC?

In general terms, the strength of an emulator is it's capability to include projections from a range of different models. We have AMOC projections for several CMIP5 models, why is it not applied to those? In addition, there are the AMOC sensitivity studies by

Gregory et al. (2005) and Stouffer et al. (2006) that would provide enough runs to calibrate the model. Why isn't it applied to those runs?

In addition, the authors mention the AMOCMIP project. Can the emulator be applied to the AMOCMIP output? I checked the project homepage and understood that the AMOCMIP will explictly resolve different Greenland basins separately. Is that correct? If so, and following recent findings that it actually matters a lot for North Atlantic dynamics where the freshwater is actually applied, will this emulator be the best tool to reproduce these dynamics? Or should it maybe consist of a subpolar (Labrador Sea) and North Atlantic box? And/or should conceptual models of convection in marginal seas e.g. by Spall (2004) and Straneo (2009) be integrated?

**1.4 Results**

I've to admit I'm not impressed by the capabilities of the emulator in reproducing the model outcome. As apparent from Fig. 7, the emulator is systematically underestimating AMOC reduction for RCP4.5 and RCP8.5 no melt, while then over-estimating it for RCP8.5 plus GIS (maybe due to non-linearities kicking in here and timescale issues discussed above?). The authors discussion of this simply stating that "It is, however, to be expected that a box-model does not completely capture the behavior of the AMOC as simulated with a higher order climate model" is clearly insufficient. In particular, as there have been much simpler AMOC emulators around that actually perform much better (also and in particularly an AMOC recovery, e.g. Schleussner et al. 2014).

The apparent oscillations in the emulator arising from a "too direct response" of the emulator towards multi-decadal surface temperature oscillations also merits more discussion.

It is even worse for the predictions in Fig. 8. First of all, the figure is not well-labelled (no y-axis labeling, panels not clearly distinguishable, and what is given by the numbers

5,1,5?) and that there is no such thing as a top-middle panel for only two boxes.

For none of the panels, the model actually captures key features. It fails to capture the bumps in the top-left and bottom right, and for the two other panels, it gets it wrong completely. I cannot agree to the author's conclusions that "Overall, the predictive power of the AMOC-emulator is good for reasonable forcing scenarios when one considers the simplicity of the model."

**1.5  Summary**

Generally, I miss a section that reflects on the limitations and short-comings of the approach taken, given in particular the apparent limitations in reproducing the EMIC results. Furthermore, an outlook of where this can be applied and what it specific strengths are compared to other approaches should be included.

---

## Referee Comment (RC3) · Anonymous Referee #3 · 21 Jun 2016

**A review of "AMOC-emulator M-AMOC1.0 for uncertainty assessment of future projections"**

I'm a physical oceanographer, not an expert in climate research, and not very familiar with the climate literature, especially in this line of statistical studies. I cannot, therefore, judge whether the authors adequately quote past studies or whether this study is novel. I just trust the authors in that respect.

[Figure]

**1 Summary**

This study develops a simple coupled atmosphere-ocean model ("AMOC-emulator") to predict the strength of AMOC. Some of the model parameters are tuned to fit GCM results. The emulator has some skill in predicting AMOC strength in GCM runs that were not used to tune the parameters.

I agree with the authors that such a simple model can be useful in exploring many climate scenarios, but I think there are two serious problems in this manuscript: 1) the model is dubious; 2) the tuning of parameters is careless (I think) and the validation of the result of the tuning isn't adequate.

In the following, I explain these two points as well as offer some general comments and numerous minor comments, some of which support my arguments on these major points and others, I hope, may be helpful in improving the manuscript.

**2 The model formulation**

**2.1 On using a conceptual model**

I don't understand why this kind of ad-hoc box model is preferred. One would resort to Stommel's box model if one knows almost nothing about AMOC. Stommel's model was just *conceptual*, whose sole purpose is to get very rough ideas on how AMOC might work, and is not designed for the kind of *quantitative* modeling that the present authors pursue.

Stommel's box model was presented in 1961, and we know a lot more today. I would think that a simple *dynamical* model like Gnanadesikan's (1999, Science vol. 283, pp. 2077– ) is much better because even uncertain parameters are based on (that is,

constrained by) clearly-identified dynamics. In contrast, some of the present authors' "parameterizations" aren't adequately defendable; they are based on hand-waving arguments. For example, how can one defend the parameterization that the AMOC strength is proportional to the interspheric surface-density difference? We know a lot better than that.

Perhaps even better, the models of Schloesser et al. (2014, Prog. Oceanogr. vol. 120, pp. 154– ) and McCreary et al. (2016, Prog. Oceanogr. vol. 123, pp. 46– ) provide constraints among integral quantities, such as AMOC strength, thermocline depth, and meridional density difference, and hence can be utilized as a "box model". Those constrains are derived as solutions to dynamical equations rather than assumed on the basis of hand-waving arguments.

In short, I don't see any advantage today in utilizing an old conceptual model for quantitative prediction.

**2.2 AMOC proportional to interhemispheric density difference?**

The authors says that "the assumption that the meridional Atlantic density contrast between the North Atlantic and the South Atlantic is the first order driver of the AMOC" is debatable, but I think that's off the mark. The current wisdom is that the Southern-Ocean winds (and perhaps vertical diffusivity) are the first-order driver. They cite Butler et al. (2016) as the *other side* of the "debate" but Butler et al. do not argue that the surface meridional density gradient "drives" the AMOC. The just use density integrated twice in the vertical as a "diagnostic" of the AMOC.

It is clear from ocean GCM studies that the meridional density gradient is not the first-order driver of AMOC. When the sea-surface density is restored toward a prescribed profile in an ocean-only GCM and windstress is changed in the Southern Ocean, the AMOC strength changes roughly linearly to the windstress. See Toggweiler et

al. (1995, Dee-Sea Research vol. 42, pp.477– ) and the series of studies that follow. This is evidence enough that the interhemispheric density difference does not drive AMOC.

Of course, this evidence is based on ocean-only models, and it is possible that the interhemispheric density difference is correlated with the AMOC strength through atmospheric feedbacks, but to use a one-to-one correspondence like (1) needs justification based on atmosphere-ocean coupled dynamics.

By the way, I found that Butler et al. (2016) still use the traditional hand-waving parameterization $\Delta p_x/L_x \propto \Delta p_y/L_y$. See Schloesser et al. (2012, Prog. Oceanogr. vol. 101, pp. 33– ) for a better parameterization based a lot more on dynamics.

**3  Tuning and validation**

**3.1  GCMs for tuning**

Why aren't multiple coupled GCMs used to tune the parameters? Do the authors recommend that the AMOC-emulator be tuned differently for each model?

The emulator is based on equations that represent physical processes in the real world. Then, if at all possible, the parameters should be tuned on the basis of reality. Granted that there is not enough data for the deep ocean.  Then the second best thing is the publicly-available collections of coupled GCM runs.

I think that studies have indicated that a multi-model ensemble is usually better than a single model to mimic reality. So, the tuned parameters would be more likely better if they are based on multiple models.

**3.2 Variables for tuning**

The variables (salinity, ocean temperature, etc.) of the emulator should be compared with those from the GCM. It is possible that the state of the emulator is very different from that of the GCM even when the AMOC strength $m$ agrees.

On a more basic note, have the authors made sure that all the variables of the emulator take reasonable values? I don't think it would be okay if, say, salinity takes a value of $-100$ psu even if the value of $m$ is reasonable!

If the atmosphere and ocean states aren't realistic, how can we trust the emulator?

One approach to cope with this problem would be to include other variables than AMOC strength in the cost function. Another approach would be to compare various variables between the runs of the tuned model and those of the GCMs (part of validation). I think both are necessary.

**3.3 Models for validation**

Moreover, I think the tuned emulator should be validated against another set of *different* models. Otherwise the validation isn't robust.

I also wonder if an ensemble of runs are necessary for the GCM (UVic) for tuning and validation. For example, there is only one run for each case in Figure 8, but doesn't the AMOC strength differ from realization to realization? I don't know how chaotic the GCM is (because it uses a low-degree-of-freedom atmospheric model), but isn't the reality more or less chaotic?

**4 Forcing**

I may be missing something, but it's not clear to me what forces the emulator. The solar flux $S$ seems constant in time (Table 1), but then how is the increase in green-house gas represented?

If I understand it correctly, $\epsilon$ forces the model (equation 19) toward one particular GCM solution, but wouldn't it damp the emulator's variability? especially when the emulator is to simulate a state that is very different from the GCM state used for $\epsilon$? Doesn't this amount to building the solution into the simulation?

**5 Minor points: math notations**

**5.1 Arrays**

The authors define boldface math symbols to mean "arrays", but I recommend avoiding this unconventional convention. For example, a multiplication of two "arrays" can mean several different things in conventional mathematics. Equation (13) includes the multiplication of the arrays $S$ and $\alpha_p$, which is meant to represent $(S_1\alpha_{p1}, S_2\alpha_{p2}, \ldots)$, which is hardly conventional. $K$ and $T_a$ have the same problem in (14). Also, equation (11) includes $1/z$, by which the authors mean $(1/z_1, 1/z_2, \ldots)$, but which is not widely used in math.

All these problems are usually solved by using indices: for example, (14) can be written as

$$H_{aj} = -CK_j\frac{\delta T_{aj}}{\delta y}, \quad j = 0, \ldots, 4.$$

14'(1)If too many subscripts become nuisance, we could use both super- and subscripts: $H_j^{a2o}$, for example.

**5.2 Subscripts**

I recommend using an upright font for multi-character math symbols such as "start" and "gcm"; or avoiding them. In particular, the subscript "$it$" looks as if it represented two subscripts $i$ and $t$. I recommend using a single-character subscript, such as "$\psi_{i-1}$" or if you insist on multi-character subscript, you may want "$\psi_{\mathrm{it}-1}$" using an upright font.

**6  Point by point comments**

Some of the following comments support my arguments above, some raise other concerns, and others point out minor, mostly editorial, problems. I wrote many of them as I read the manuscript for the first time, and as a result, they include some redundancy. I leave them as they are, because they often reflect difficulties or problems the reader may encounter as she reads the text.

**6.0.1   p. 1, l. 19:**

"due to climate sensitivity, polar amplification, GIS melt and model dependent sensitivity of the AMOC . . . "—I'm confused. Doesn't "climate sensitivity" include all the remaining items in the list? Why is it listed in parallel with the rest?

**6.0.2   p. 2, l. 5:**

"(Rahmstorf and Willebrand, 1981)"—As the reference list indicates, this should probably be Rahmstorf and Willebrand (1995).

**6.0.3  p. 2, ll. 5 & 31:**

"the so-called Bjerknes feedback"—Probably this is because I'm not much versed in climate research, but isn't the "so-called Bjerknes feedback" restricted along the equator? A direct overturning circulation occurs connecting cooling in the eastern Pacific, say, and warming in the western Pacific only along the equator, where the Coriolis force vanishes, and the surface windstress associated with this zonal overturning circulation enhances the upwelling of sea water, which further lowers the sea-surface temperature in the eastern Pacific—a positive feedback, which is "the so-called Bjerknes feedback".

The authors cite Rahmstorf and Willebrand (1995) for "the so-called Bjerknes feedback", but Rahmstorf and Willebrand proposed a negative feedback due to heat transport within the atmosphere, I think.

**6.0.4  p. 2, l. 7:**

"tuning a numbrer of free parameters"—They aren't "free". They represent specific physical processes and hence must be ultimately determined by physics, even though it's in practice difficult to derive their values purely from physical principles.

**6.0.5  p. 3, l. 12:**

Why is $F$ prescribed? I would expect it to change according to the state of the climate system. What do IPCC-class coupled GCMs say about the change in $F$ under global warming, for example?

... but, later in the text, the authors say that $F$ is related to the global atmospheric temperature (equation 12). So, it's not prescribed after all.

[Figure]

**6.0.6 Equations (3)–(10):**

What does this "$\delta$" mean? Is it a typo for "$\partial$"?

**6.0.7 Equation (11):**

State whether $z$'s are fixed, and if so, give their values here or refer the reader to a table or something.

**6.0.8 Equation (12):**

$\Delta T_{glob}$ should be defined. (How is it computed from $\boldsymbol{T_a}$?)

**6.0.9 Equation (12):**

Give $F_0$'s their values here or refer the reader to a table or something.

**6.0.10 Equation (13):**

I may be mistaken, but it seems that the $\boldsymbol{T_a}^4$ is the only nonlinear term. Doesn't it make sense if this term is linearized around a mean state?

**6.0.11 Equation (13):**

The solar flux $\boldsymbol{S}$ is a confusing notation. By the authors' own convention, $\boldsymbol{S} = (S_0, S_1, S_2, S_3, S_4)$, which uses the same symbols as salinity.

**6.0.12 Equation (13):**

The solar flux $S$ should be discussed right below equation (13). Does it depend on time? Table 1 suggests that it's constant in time but that should be stated explicitly. So, does the emulator solves only for annual averages?

**6.0.13 Equation (13):**

Define this $\nabla$ precisely. (But I don't recommend this notation because a gradient of an array is a strange mathematical entity.)

**6.0.14 Equation (14):**

What does this "$\delta$" mean? Is it a typo for "$\partial$"?

**6.0.15 Equation (14):**

State that $H_a$ is defined to vanish at the northern and southern ends of the northern-most and southernmost boxes. (I guess they are so defined, right?)

**6.0.16 p. 5, l. 15:**

I guess we need some discussion on other possible sets of tuning parameters. We have a vast range of possibilities. Then, how have we settled on these seven parameters? Have the authors tried other combinations of parameters?

[Figure]

**6.0.17 p. 5, l. 15:**

$F$ and $h$ are related by equation (12), and so cannot be determined independently. Moreover, if you tune $F$, you can forget about equation (12) and don't need to consider $h$.

**6.0.18 Equation (15):**

Why try to optimize $m$ alone? It's conceivable that widely different states have similar $m$ values. Because we have other variables like salinity, we could choose better sets of parameter values, if we include other variables in the cost function, couldn't we?

**6.0.19 Equation (15):**

I may well be mistaken, but it seems that the differential equations are linear in the tuning parameters and if so, the optimization problem on the new cost function

$$C' \equiv \sum \sum \left( \frac{\mathrm{d}m_{emu}}{\mathrm{d}t} - \frac{\mathrm{d}m_{gcm}}{\mathrm{d}t} \right)^2$$

is a quadratic function of the parameters and can be solved analytically, I think.

**6.0.20 Equation (16):**

The notation "$p_{start}(1-z)$" is confusing because it looks as if $p_{start}(z)$ were a function of $z$. Vectors customarily come after scalars, as in "$(1-z)p_{start}$"

**6.0.21 Step 1:**

I don't understand why we have to repeat this step. Why not choose values that are within the ranges in Table 2 in the first place? We can use a random variable whose PDF is uniform over the specified range for each parameter, can't we? I mean, if $(1-z)p_1$ is below the range, we can just use $p_{1min}$ for the lower bound; that is, we can use $U(\max((1-z)p_1, p_{1min}),\ \min((1+z)p_1, p_{1max}))$ without repetition. The same argument holds for the last part of Step 3.

**6.0.22 Equation (17):**

I may be missing something, but shouldn't (16) and (17) be written in parallel forms? If we write $(1-z)\boldsymbol{p}$ for (16), then we should write $(1-\psi_{it})\boldsymbol{p}$ for (17). If we write $\boldsymbol{p}-\psi_{it}\boldsymbol{p}$ for (17), we should write $\boldsymbol{p}-z\boldsymbol{p}$ for (16). For a moment, I was confused with (17).

**6.0.23 p. 7, l. 16:**

I think that efforts should be made to narrow the range of the parameter values. If parameters are widely different even though the cost function is similar, doesn't that suggest that the parameters aren't well tuned?

What about comparing variables other than $m$ between the emulator and the GCM? Wouldn't that tell which parameter values are bad?

It seems that the authors have forgotten that there is only one reality.

[Figure]

**6.0.24  p. 7, l. 22:**

What is "RCP"? (I may have missed its definition given in the text.) Because how $\epsilon$ is determined is important, it may be helpful to give a bit more information here.

**6.0.25  Equation (19):**

Does the model really use the full time-series of $T_{gcm}$? Or is that a long-term mean? State clearly how is $T_{gcm}$ defined.

If the emulator uses the full time-series, it may not be appropriate for other models or for other scenarios.

**6.0.26  p. 8, l. 2:**

"Note that the temperature forcing files need to be interpolated onto the temporal resolution used in the atmospheric component of the AMOC-emulator"—Awkward in several counts.

1. The interpolation draws the attention of the reader as if it were something noteworthy. *Perhaps the results are sensitive to the method of interpolation?* the reader would wonder.

2. Is the fact that the GCM data are saved in files noteworthy? (I mean, why mention the files at all?)

3. Despite this cautious tone, the interval at which the GCM data is saved is not indicated.

If the result is sensitive to the interpolation, give more details. If not, what about just saying, "The GCM variables are saved at an interval of XXX hours and interpolated on to the time steps of the AMOC-emulator", something along the lines.

6.0.27   p. 8, ll. 5–6:

A similar problem. If interpolation is so noteworthy, give more details. If it's not so big a deal, just say, "the GIS melt forcing is interpolated. . . ." instead of "Note that the GIS melt forcing needs to be interpolated. . . ."

---

## Editor Comment (EC1) · A. Yool (Editor) · 13 Jul 2016

Dear authors,

As you can see, three referee reports have been submitted on your paper, together with an additional comment, and the discussion phase is now complete.

While your referees all remarked on the utility and potential of emulators for exploring uncertainly, all have also raised a number of significant issues with the work described in your manuscript. In particular, the relatively poor performance of the emulator has drawn attention, as well as the overly positive description of this in your manuscript. As they are simplified representations of models that are already simplified versions

of reality, it is critical that emulators both perform well and have their shortcomings discussed. Both of which were found lacking here.

Please respond fully to all of the comments and criticisms raised by the referees.

If you have any questions, please do not hesitate to contact me.

With best regards,

Andrew Yool.
* * *

---

## Author Comment (AC1) · 18 Aug 2016

**Response to general comment by P. G. Myers,**

**We thanks P. G. Myers for his constructive comments on the manuscript. Below we will give detailed answers to the individual comments.**

*page 2, Line 18: Would the approach work in a GCM whose AMOC is controlled by the strength of southern hemisphere winds? This is a potential major caveat, that might be worth discussing more at the end of the paper when the authors expand on applications.*
**Thanks for this question. The usage of this AMOC-emulator to mimic the AMOC behavior in more complex GCMs is based on the assumption that meridional density differences are the first order driver of AMOC changes on long time-scales. If in a GCM this is not the case, the AMOC-emulator should not be used. We agree that this is an important point, and have thus included a comment on this topic in the revised discussion section of the manuscript (lines 22-25 page 12) "...many processes that are known to impact the AMOC are not considered in the AMOC-emulator, for instance the impact of winds, gyre circulation, Southern Ocean upwelling or deep water formation outside of the North Atlantic (see Sect. 1). If such processes would prove to dominate the AMOC response to future climate change, a different AMOC box model should be considered that places emphasis on that particular process."**

*page 4, line 12: Is it reasonable to assume the wind-driven oceanic meridional fresh-water transport is static. I might expect it to evolve with an evolving climate, especially as high latitude freshwater sources (e.g. GIS runoff) changes.*
**Thanks for this question. Indeed in the real climate, or in higher complexity climate models for that matter, changes in meridional ocean circulation are most likely not only comprised of changes in the density driven circulation, but also in the for instance the gyre transport. A good example of the complexity of such future North Atlantic circulation changes is given in Swingedouw et al. (2015). By using a box-model that is driven by meridional density differences, such complexities are considered of secondary importance. However, since we tune the box-model to mimic the behavior of fully coupled models that do include such processes, they do influence the sensitivity of the emulator to changes in temperature and GIS melt. To our knowledge, the current understanding of the sensitivity of individual mechanisms that drive Atlantic Ocean circulation changes is not sufficient to be incorporated into a sufficiently simple model to be used in an AMOC-emulator.**

*Page 5, line 30: algorithm misspelled.*
**Thanks, it is corrected.**

*page 6, line 4: Why fixed limits of plus/minus 200%? Why not something based on each variable's variability/range?*
**The plus/minus 200% range is indeed somewhat arbitrary, but this approach is chosen because for most of the variables a reasonable estimate of the variability or range does not exist, mostly because the parameters do not have a 'meaning' in the real world. The updated manuscript includes a line discussing this topic (lines 14-15 page 7) "The appropriateness of this arbitrary range of initial parameter values is later verified by ensuring that all final parameter values are well within the initial range."**

*page 6, line 8 - analogous misspelled.*
**Thanks, it is corrected.**

*Summary: Way too short. I'd like to see more discussion on how such an emulator could be improved, its limitations, questions it might be best for, etc.*
We agree that a more substantial discussion section is called for. We have rewritten it to read (lines 4-25 page 12) "Overall, the predictive power of the AMOC-emulator is reasonable when one considers the simplicity of the AMOC box model, but for forcing scenarios that are increasingly far away from the forcings that are used in tuning the AMOC-emulator, the predictive power decreases. A large advantage of using a physics-based AMOC-emulator that is tuned with large climate forcings, over the use of for instance a statistical AMOC-emulator, is that it projects the point after which the AMOC collapses and switches to an off state, as this is an integral part of the physics of the Stommel model. It is clear that using an AMOC-emulator introduces new uncertainty into AMOC projections, however, for which level of added uncertainty an AMOC-emulator is still useful is a question that is difficult to address. Another important consideration when using the AMOC-emulator is the spread in GCM climate forcing scenarios that is included in the tuning process. When using only a single climate change scenario, a  better match can be obtained between the AMOC evolution given by the GCM and AMOC-emulator, however, in this case the reliability of the AMOC-emulator will quickly decrease for different climate forcings. On the other hand, one could use a large number of climate change projections in the tuning process to obtain a lesser fit for individual scenarios, but an AMOC-emulator that is applicable to a much larger range of climate change scenarios. The best strategy to be followed strongly depends on the research question in mind. The assumptions behind the AMOC-emulator presented here, limit it to projecting AMOC changes on multi-decadal and larger timescales. Therefore, the applied GCM-based climate forcings and AMOC strength time series should best be filtered to exclude high resolution variability. Moreover, an AMOC-emulator that is tuned to specific GIS melt experiments is likely not applicable to experiments in which melt water is applied to a different geographical region or with a different seasonal cycle. This is not to say that the presented AMOC-emulator framework cannot equally be applied to other sources of melt water input. Finally, many processes that are known to impact the AMOC are not considered in the AMOC-emulator, for instance the impact of winds, gyre circulation, Southern Ocean upwelling or deep water formation outside of the North Atlantic (see Sect. 1). If such processes would prove to dominate the AMOC response to future climate change, a different AMOC box model should be considered that places emphasis on that particular process."

*Table 1: For last 3 entries, dependent misspelled*
Thanks, it is corrected.

*Figure 1: Why is the atmospheric part squished so much in the vertical compared to the oceanic component?*
We have adjusted the figure in order for it to have more appropriate scaling.

*Figure 4 caption: relative misspelled.*
Thanks, it is corrected.

*Figure 8 caption: right misspelled.*
Thanks, it is corrected.

Reference:
Didier Swingedouw, Christian B. Rodehacke, Steffen M. Olsen, Matthew Menary, Yongqi Gao, Uwe Mikolajewicz, Juliette Mignot. On the reduced sensitivity of the Atlantic overturning to

**Greenland ice sheet melting in projections: a multi-model assessment. Climate dynamics 44, 3261-3279, 2015.**

---

## Author Comment (AC2) · 18 Aug 2016

**Response to anonymous Referee #1**

**We thanks Reviewer 1 for the interesting and extensive comments on the manuscripts. Below we will provide a detailed response to all individual comments.**

*I don't understand the reasoning for why you generate 100 reasonable fits and then select only the 10 best fits (P7 L15). Firstly, this emulator/box-model should be cheap to run, so why choose such small numbers? Surely ensembles of order 10,000 or 100,000 are more reasonable. Secondly, the choice of 10 best fits seems to narrow the ranges of several parameters (e.g. V4, F1, h1). By doing this you rule out large regions of parameter space that give perfectly reasonable fits, and could behave differently under different forcing scenarios. If the primary aim is to assess uncertainty in AMOC projections I would expect to see a rigorous analysis of the uncertainty. By discarding large areas of parameter space uncertainty will certainly be underestimated.*

**Thanks for raising this valid point. Our aim here is not to provide an uncertainty assessment of AMOC projections, it is to provide a method with which one could do this, as for example done in the manuscript by Bakker et al. under review in GRL, now also pointed out in the last line of the main manuscript (lines 1-2 page 13) "The AMOC-emulator is a valuable tool to study the uncertainty in GCM-based AMOC projections, such as the one recently being performed on the results from the AMOCMIP project (Bakker et al., 2016)." The assessment referred to here is based on multiple GCMs, decreasing the need for a large number of AMOC-emulators for a single GCM.**

**With regard to the point that the AMOC-emulator is cheap and could thus be run for tens or even hundreds of thousands of times. This is very true, however, it takes many time steps and iterations to find a single reasonable fit. We have now included the following description to the manuscript (line 30 page 8 to line 2 page 9) "To provide an idea of the computational expenses of the model we provide a back of the envelope calculation. This shows that a single run over all scenarios takes $10^5$ time steps which are done in about 5 seconds. You need on the order of 400 iterations (in which parameter values are perturbed) to find a single reasonable fit, resulting in approximately half an hour to calculate a single reasonable fit on a normal desktop computer." This shows that by using more powerful computers and or running in parallel the number of reasonable fits could be enhanced, but it shows that 10,000 to 100,000 reasonable fits is ambitious nonetheless.**

*In the four scenarios not seen by the emulator (Fig. 8) the behavior of UVic is clearly not captured in two cases (lower left and upper right). There are no confidence intervals plotted (or computed as far as I can tell), but I believe the GCM would lie well outside 2 standard errors in those two cases. Therefore, the GCM would still need to be run for any untested scenario. I would not trust the emulator in its current form.*

**We don't agree with the general notion given by the reviewer. Firstly, it is important to realize that the values given in figure 8 are anomalies with respect to the time series given in figure 7. Thus even the largest mismatch between GCM and AMOC-emulator (~1-2Sv in lower left panel) is 'only' a mismatch of 10-20%. We have added an objective assessment of the predictive power of the AMOC-emulator by comparing the results with a null-model that assumes that the emulator has no predictive power; it doesn't know if an additional forcing on top of the ones used in the tuning procedure would further increase or decrease the AMOC and would thus result in zero anomalies. This assessment shows that in three out of four cases the AMOC-emulator has substantial predictive power. We discuss this assessment in the manuscript (lines 20-30 page 11) "This is quantified by comparing the AMOC-emulator results with a null-model**

that assumes an AMOC-emulator with zero skill, meaning that it simply reproduces the original calibration data. The results from these experiments are shown as anomalies relative to the original scenario, the original being RCP8.5-GIS for RCP8.5x0.5-GIS, RCP8.5x1.5-GIS and RCP8.5-GISx1.5, and RCP4.5-GIS for RCP4.5-GISRCP8.5x1.5. We find that for large changes in the GHG forcing the Uvic-based AMOC-emulators are well capable of predicting the AMOC evolution of UVic in terms of sign and amplitude and perform better than the null-model (upper panels Fig. 8). For large changes in the applied GIS melt forcing the picture is more complex (lower panels Fig. 8). A strong increase in GIS melt under a low GHG scenario shows an excellent performance of the AMOC-emulator and a RSME that is much lower than for the null-model (RCP4.5-GISRCP-8.5x1.5 in Fig. 8), but for the high GHG scenario, a 50% increase in GIS melt leads to a deterioration of the fit between UVic and AMOC-emulator with consequently a larger RSME than that provided by the null-model (RCP8.5-GISx1.5 in Fig. 8). The latter shows that the UVic-based AMOC-emulators tend to overestimate the impact of GIS melt on the AMOC strength under high-end GHG scenarios. Summarizing, in all four cases the emulator predicts the correct sign of the AMOC response to changes in the forcings, and in three out of four cases the predictive power of the AMOC-emulator is better than of the null-model.". Nonetheless, it is important to acknowledge that using an emulator will introduce a new type of error in any assessment, pointed out by the following text in the manuscript (lines 5-7 page 12) "It is clear that using an AMOC-emulator introduces a new type of uncertainty into AMOC projections, however, for which level of added uncertainty an AMOC-emulator is still useful is a question that is difficult to address."

*On multidecadal timescales the emulator is plagued by sensitivity to surface temperature oscillations. These seem to have arisen from the addition of the atmospheric boxes to the ocean box model published by Zickfeld et al., 2004. Can the authors confirm that this is the case, and if so can they control this sensitivity, e.g. by introducing a damping/mixing term?*
The multidecadal AMOC oscillations result from the UVic-based regional temperature forcings of the AMOC-emulator and thus in turn to internal variability of UVic. Zickfeld et al. (2004) applied highly idealized linear temperature increases of global temperature, thus not including any multi-decadal variability. On the contrary, in our approach we directly use regional GCM-based temperature time series to force the AMOC-emulator. In this way the forcing not only takes into account the GCMs global climate sensitivity, but also mechanisms like polar amplification etc. that cause regional temperature change differences. This method also introduces any multi-decadal internal variability that might exist in a GCM into the AMOC-emulator when expressed in regional temperature time series. We acknowledge this feature, but do not see it as an issue.

*If the authors have good reason to retain this behavior they need to test the sensitivity to the phase of the variability. For all of the scenarios, the chosen start date (2006) appears to be shortly after a peak in the strong multidecadal variability, so the AMOC is preconditioned to decline at this time. Under all scenarios the AMOC in the 'best' emulators appear to decline faster than the UVic model. Consequently, the SA tuning and the cost function used may be adversely affected by this multidecadal variability.*
Indeed, following from the usage of the Stommel model to emulate the AMOC, multi-decadal temperature variability and its phasing impact the projected AMOC changes, in the AMOC-emulator, in UVic and most likely also in reality. Perhaps the AMOC response in the AMOC-emulator to regional temperature changes is too direct (as mentioned in the manuscript) and thus the importance of multi-decadal variability overestimated, but we don't see this as a major issue. It seems to us that the years before 2006 represent in fact a time of relatively weak AMOC, not

**strong, thus preconditioning the AMOC-emulator to a somewhat weaker response to global change. We don't agree with the notion of the reviewer that the decline in the emulators is faster than in UVic, they seem very similar to us. Finally, multi-decadal AMOC variability only impacts the absolute value of the cost function, not the resulting optimal fits.**

*On centennial timescales the emulator (as currently presented) does not capture crucial features of the AMOC response to the forcing (Fig. 7). In particular I would draw attention to the RCP4.5 scenarios, in which the GCM exhibits a strong reduction followed by a steady recovery. The emulator fails to identify either the timing or amplitude of the AMOC minimum and it fails to identify the recovery phase. In addition it appears to show signs of a recovery phase under RCP8.5 when UVic shows none. The authors state (P9 L28) that the fit can be improved, but that this would entail a higher overall cost function for the SA tuning method. Is this indicative of a poor choice of cost function? Does it mean that the box model should be tuned separately for each scenario?*

**The failure of the AMOC emulator to capture the slight recovery of the AMOC under RCP4.5 is indeed an issue and shows the limitations of the simple box model to capture all complex feedbacks in the GCM. Indeed, as mentioned in the manuscript, the AMOC-emulator does allow for an AMOC recovery under RCP4.5, but that would mean a large deterioration of the fit of the AMOC emulator to the AMOC in RCP8.5 and thus it would increase the value of the cost function, for which reason this solution is not found through this approach. It is an interesting point if the AMOC-emulator should be tuned separately for each scenario. We added the following to the manuscript to cover this issue (lines 23-33 page 10) "It is also worth noting that the fit for an individual simulation could be improved, for instance the AMOC-emulator does allow for a partial AMOC recovery as UVic shows for RCP4.5, but such an AMOC-emulator is not found through the SA tuning methodology in this example, because it would degrade the fit for the other scenarios and thus lead to an overall higher cost function." More discussion on this topic follows in Sect. 4 of the manuscript (lines 7-13 page 12) "Another important consideration when using the AMOC-emulator is the spread in GCM climate forcing scenarios that is included in the tuning process. When using only a single climate change scenario, a better match can be obtained between the AMOC evolution given by the GCM and AMOC-emulator, however, the reliability of the AMOC-emulator will quickly decrease for different climate forcings. On the other hand, one could use a large number of climate change projections in the tuning process to obtain a lesser fit for individual scenarios, but an AMOC-emulator that is applicable to a much larger range of climate change scenarios. The best strategy to be follow strongly depends on the research question in mind."**

*A far more substantial summary is required. For example, the emulator's limitations need to be clearly stated (and whether/how the authors think these can be addressed). For what purposes are the emulator suitable in its current form, and for what purposes might it be useful subject to further work? With the current analysis, I disagree with the statement that "the UVic-based AMOC-emulator captures well the overall characteristics of the multi-centennial response of the AMOC".*

**Thanks for this comment. We agree that are more substantial and clear discussion is needed to make clear what the model can and cannot do. We have added the following to the discussion section (lines 1-22 page 12) "Overall, the predictive power of the AMOC-emulator is reasonable when one considers the simplicity of the AMOC box model, but for forcing scenarios that are increasingly far away from the forcings that are used in tuning the AMOC-emulator, the predictive power decreases. A large advantage of using a physics-based AMOC-emulator that is tuned with large climate forcings, over the use of for instance a statistical AMOC-emulator, is that it projects the point after which the AMOC collapses and switches to an off state, as this is an integral part of the physics of the Stommel model. It is clear that using an AMOC-emulator**

introduces new uncertainty into AMOC projections, however, for which level of added uncertainty an AMOC-emulator is still useful is a question that is difficult to address. Another important consideration when using the AMOC-emulator is the spread in GCM climate forcing scenarios that is included in the tuning process. When using only a single climate change scenario, a  better match can be obtained between the AMOC evolution given by the GCM and AMOC-emulator, however, in this case the reliability of the AMOC-emulator will quickly decrease for different climate forcings. On the other hand, one could use a large number of climate change projections in the tuning process to obtain a lesser fit for individual scenarios, but an AMOC-emulator that is applicable to a much larger range of climate change scenarios. The best strategy to be followed strongly depends on the research question in mind. The assumptions behind the AMOC-emulator presented here, limit it to projecting AMOC changes on multi-decadal and larger timescales. Therefore, the applied GCM-based climate forcings and AMOC strength time series should best be filtered to exclude high frequency variability. Moreover, an AMOC-emulator that is tuned to specific GIS melt experiments is likely not applicable to experiments in which melt water is applied to a different geographical region or with a different seasonal cycle. This is not to say that the presented AMOC-emulator framework cannot equally be applied to other sources of melt water input. Finally, many processes that are known to impact the AMOC are not considered in the AMOC-emulator, for instance the impact of winds, gyre circulation, Southern Ocean upwelling or deep water formation outside of the North Atlantic (see Sect. 1). If such processes would prove to dominate the AMOC response to future climate change, a different AMOC box model should be considered that places emphasis on that particular process."

*Minor comments:-*
*Page 3 Line 12: Prescribed FW fluxes: F1 and F2 are tuned parameters. I would have expected these to vary as a function of the forcing/climate. What is the justification for fixing them?*
This part was not sufficiently clear in the manuscript and has now been updated. The total freshwater fluxes F1 and F2 are not part of the tuning procedure, but F01 and F02 (the combined wind-driven oceanic and atmospheric meridional freshwater fluxes for the reference state are). The text should have read (line 18 page 4) "Freshwater fluxes $F_{01}$ , $F_{02}$ and coefficients $h_1$ and $h_2$ are included in the tuning procedure (Tab. 2)"

*Page 5 Line 10: What you also fail to consider are nonlinearities between these parameters. Co-varying the parameters in Tables 1 and 2 could yield very different behaviours.*
The parameter fitting method we employ, simulated annealing, randomly varies the individual parameters, thus considering (although not explicitly) both linear and nonlinear relationships between parameters. Moreover, by including Figure 6 we perform a first order test to see whether relationships exist between parameters, which indeed is the case for several of them.

*Page 5 Line 30: algorith > algorithm*
Thank you, it has been corrected.

*Page 6 Line 4: I find the arbitrary choice of +/- 200% rather strange. What is the justification for this?*
We agree that this choice is arbitrary. Our approach has been to take this arbitrary value, perform the analysis and then to analyze whether or not all parameter values that resulted from the fitting procedure were well within the +-200% range (see also figure 6). From this it was decided to keep the +-200% value. This point is clarified by adding (lines 14-15 page 7) "The appropriateness of this arbitrary range of initial parameter values is later verified by ensuring that all final parameter values are well within the initial range."

*Page 6 Line 8: analogues > analogous*
**Thank you, it has been corrected.**

*Check typesetting in Tables (e.g. Table 1 column 2)*
**Thank you, typesetting is checked.**

*Table 1: (typo) dependend > dependent*
**Thank you, it has been corrected.**

*Check typesetting on Figure 8: it appears corrupted.*
**Thank you, typesetting is checked.**

*Figure 4 caption: (typo) relatvie > relative*
**Thank you, it has been corrected.**

*Figure 8 caption: (typo) calculate > calculated*
**Thank you, it has been corrected.**

*Figure 8 caption: (typo) rigth > right*
**Thank you, it has been corrected.**

---

## Author Comment (AC3) · 18 Aug 2016

**Response to anonymous Referee #2**

**We thanks Reviewer 2 for the interesting and extensive comments on the manuscripts. Below we will provide a detailed response to all individual comments.**

*1 General Comments*
*1.1 Introduction*
*The introduction does not sufficiently reflect the state of the literature on AMOC and in particularly not on conceptual models such as the Stommel-Model to study it.*
*Our understanding of AMOC dynamics has advanced considerably over the last years thanks to ongoing observations e.g. in the Rapid array (see Srokosz & Byrden, Science 2015). In addition a recent studies has suggest that the AMOC might already be in decline (Rahmstorf et al. 2015). While of course not directly relevant for the emulator itself, such observational findings need to be discussed in an approach that emulate AMOC behaviour over the next centuries. This should also include a discussion of atmospheric imprints on the AMOC e.g. such as atmospheric blocking events. It should also allow to assess the performance of GCMs in relation to the observational record.*
**We cannot agree with the points raised above. We are presenting a new modeling framework in a journal for model development. As such, we agree that some insights as to why we think a new modeling framework is needed is called for, but a discussion of observed AMOC changes, AMOC fingerprints or the performance of GCMs in relation to the observational record does not seem appropriate in this journal and is beyond the scope of this paper..**

*Much more important though is the discussion in relation to the emulator approach taken. Stommel type models have been used since quite some time and might be able to capture key dynamics of the AMOC (e.g. bistability). However, they at the same time have faced a lot of criticism and alternative models describing AMOC behaviour exist. This is in particular related to the relevance of Southern Ocean upwelling reflected in a conceptual model by Gnanadesikan (1999) related to changes in the pycnocline depth. A dynamic that is completely missing in the Stommel approach.*
*This has been explored further in conceptual models and attempts exist to unify pycnocline and freshwater-feedback dynamics. In this context, the authors should consider the work of Sijp et al. (2012) that they may find helpful.*
*Another question directly relating to the physical plausibility of the Stommel model relates to the relationship of circulation strength and meridional density gradient in a geostrophic ocean. The authors should consider work by Gregoy & Tailleux (2010) that present a kinetic energy approach essential providing a physical explanation for the (empirically supported) meridional density gradient outlining the relevance of the Western Boundary Current in modelling AMOC dynamics.*
*These comments should not be seen as undermining the Stommel model approach taken here, but they need to be addressed. In short, the authors should show motivate their approach in the light of the most recent literature.*
**Thanks for pointing this out and we agree that a more thorough discussion of the pro's and con's of the used Stommel model is called for. An important caveat of using a Stommel model is that Southern Ocean upwelling, the role of Southern Hemisphere mid-latitude winds and other processes are neglected, a point that we have added to the discussion of this manuscript. Our choice to use the Stommel model was driven by two considerations. Firstly, to our knowledge no unified simple AMOC model exists and as such it is not clear if other models are better or worse than the Stommel model in relating surface temperature and freshwater flux changes to the AMOC strength. Secondly, the Stommel model allows for rather straightforward inclusion of temperature and freshwater forcings based on GCM simulations, while for other models like the ones mentioned above it is not clear to us how this could be done. Finally, it is important to note**

**that we did not set out to construct a new simple model that describes the main dynamics of the AMOC, but rather to use an existing model and build a framework around it that can easily be applied to GCM climate change and AMOC projections.**

**Following the above, we have updated the introduction to read (lines 14-23 page 2) "At the center of our approach is the assumption that changes in AMOC strength are linearly related to changes in the Atlantic meridional density contrast. Since Stommel (1961) a large number of studies have provided evidence for an important role of the Atlantic meridional density contrast in driving AMOC changes (e.g. Rahmstorf, 1996; Gregory and Tailleux, 2011; Butler et al., 2016). Nonetheless, it neglects several important processes, like the role of Southern Ocean upwelling, winds and deep water formation (e.g. Gnanadesikan, 1999; de Boer et al., 2010) and a unified theory describing the fundamental mechanisms driving and sustaining the AMOC lacks to this date (Lozier, 2010). Using a Stommel model to emulate AMOC changes driven by surface temperature and freshwater forcings seems appropriate in the light of present-day knowledge and the apparent leading role of surface buoyancy changes in simulated future AMOC weakening (IPCC Climate Change, 2013). Moreover, the model is easy to use, interpreted and can be forced directly with GCM-based forcing fields. Nonetheless, the processes that have been omitted and the simplicity of the model should be considered when interpreting the results."**
**To the discussion section we have added (lines 20-24 page 12) "...many processes that are known to impact the AMOC are not considered in the AMOC-emulator, for instance the impact of winds, gyre circulation, Southern Ocean upwelling or deep water formation outside of the North Atlantic (see Sect. 1). If such processes would prove to dominate the AMOC response to future climate change, a different AMOC box model should be considered that places emphasis on that particular process."**

*1.2 The Emulator model*
*Here, the work dominantly builds on a previous model by Zickfeld et al. (2004) plus a representation of the Bjerknes feedback. It does however not become sufficiently clear, why this addition will represent a substantial advancement. The authors show the differences in Fig. 9 and describe that this will represent a negative feedback on the AMOC dynamics. But it's not clear, if Figure 9 shows two sets calibrated individually (with and without atmospheric feedback) or just from the optimal parameter set with this feedback switched on and off. Therefore, I cannot judge if the conclusion drawn by the authors on the importance of the effect are due to their specific parameter set or not.*
*It would add merit, if the authors could show that the model including the Bjerknes effect will in the end outperform the no-atmospheric feedback model in the fitting procedure. This would also justify, why there model is actually better than the one presented in Zickfeld (2004).*
**Thanks for providing this comment. Firstly, the main improvement with respect to the Zickfeld et al. (2004) model is that we provide a framework that allows one to use limited GCM AMOC and climate projections to tune an AMOC-emulator in order to perform an uncertainty analysis. The Zickfeld et al. (2004) approach used a full ~20.000yr long hysteresis simulation to tune their emulator, not feasible for most IPCC-type GCMs. Moreover, they did not force their emulator with GCM-based temperature changes or consider inter-GCM differences in regional temperature changes. Those are the features we see as most important changes with respect to earlier work, a view that is now better reflected in the introduction of the manuscript by (line 23 page 1 to line 3 page 2) "To this end we developed an AMOC-emulator framework. It entails a simple box model that uses physical relationships to represent the most important mechanisms and feedbacks that govern the AMOC's response to changes in regional surface temperatures, freshwater fluxes and enhanced melting of the GIS. The AMOC-emulator can be forced by temperature and melt water fluxes from any GCM, and using AMOC time series the free**

**parameters of the box model are tuned to mimic the GCM's AMOC sensitivity to future climate change." and in later on in the introduction it reads (lines 11-13 page 2) "the approach described here is designed specifically to allow future studies in which a limited number of climate projections from multiple GCMs, limited in the simulated forcing scenarios and simulation length, to be combined into a Bayesian framework of century time-scale probabilistic AMOC projections."**

**With respect to the added stabilizing Bjerkness feedback, it indeed appears from Figure 9 that it's impact is limited. Figure 9 shows results for the same parameter sets with this feedback switched on and off, allowing for a direct investigation of its impact. Nonetheless, we deem the model including this feedback more realistic. Moreover, the effect is non-negligible (lines 1-6 page 11) "The impact of including atmospheric meridional heat transport is a small, but non-negligible ~1Sv strengthening of the control state of the AMOC (not shown) and, more importantly, a slightly lower sensitivity to changes in radiative forcing and GIS melt (Fig. 9). This confirms our understanding of atmospheric meridional heat transport acting as a negative feedback to AMOC changes. The simulations with the atmospheric feedback included have on average a stronger AMOC by 8.1±1.9% (μ ± σ; calculated over all 10 best fits and over all five forcing scenarios)."**

*Furthermore, the model includes 5 atmospheric boxes. Why are 5 boxes needed and not 3 to resolve the meridional heat transport? I think that can be easily motivated and maybe I missed it. Maybe it's worth considering to restructure the approach by moving subsection 2.3 further up to discuss the setup of the atmospheric forcings.*
**Including high latitude atmospheric boxes allows us to have a closed energy budget and more realistic meridional atmospheric heat transport.**
**Thanks for the suggestion to rearrange this section. We have accordingly switched sections 2.2 and 2.3.**

*In this context, the authors should also reflect on the limitations of the model to reproduce transient AMOC changes that relate to the assumption of well-mixed density within the boxes. This might be in particularly relevant in relation to the Greenland freshwater input. Clearly, this represents an over-simplification and may substantially limit the capabilities of this approach to emulate transient behaviour (I'll further comment on this below).*
**Thanks for pointing this out. We fully agree that a box model can never resolve the complexities of the interaction between Greenland meltwater and the ocean. We have experimented with an additional tuning parameter to include the GCM dependent 'efficiency' of Greenland meltwater to impact the density of the North Atlantic ocean box, but decided against it since the current 7 tuning parameters already allow for sufficient freedom to tune an AMOC-emulator towards the AMOC sensitivity of a specific GCM.**

*1.3 The tuning to complex model output*
*In the manuscript, the model is tuned to an EMIC model UVIC. I think that's generally no problem, but somehow contradicts the initial claims by the authors that this emulator could now be used to run larger ensembles. What is it exactly that the emulator provides that cannot be done with an EMIC?*
*In general terms, the strength of an emulator is it's capability to include projections from a range of different models. We have AMOC projections for several CMIP5 models, why is it not applied to those?*
*In addition, there are the AMOC sensitivity studies by Gregory et al. (2005) and Stouffer et al. (2006) that would provide enough runs to calibrate the model. Why isn't it applied to those runs?*
*In addition, the authors mention the AMOCMIP project. Can the emulator be applied to the AMOCMIP output?*

**Thanks for these comments. It has become clear from the comments of the different reviewers that the aim of this manuscript is not sufficiently clear and we have changed the abstract, introduction and summary sections to improve on this. In this manuscript we want to describe a modeling framework that allows one to use limited GCM output to tune and force an AMOC box model that can in turn be used to perform uncertainty analysis. It is not the aim of this manuscript to provide future AMOC projections or provide such an uncertainty analysis. See also the responses provided above.**

*I checked the project homepage and understood that the AMOCMIP will explictly resolve different Greenland basins separately. Is that correct? If so, and following recent findings that it actually matters a lot for North Atlantic dynamics where the freshwater is actually applied, will this emulator be the best tool to reproduce these dynamics? Or should it maybe consist of a subpolar (Labrador Sea) and North Atlantic box? And/or should conceptual models of convection in marginal seas e.g. by Spall (2004) and Straneo (2009) be integrated?*

**Thanks for this question. Indeed the aim of the simulations in AMOCMIP is to provide 'realistic' Greenland melt scenarios and to apply those to IPCC-type climate change projections. This includes explicitly resolving spatial and seasonal differences in the meltwater flux. Such details cannot be captured by the AMOC-emulator. However, as described above, by tuning the AMOC-emulator to the forcings and AMOC projections of a specific GCM, we take into account the inter-GCM differences in the sensitivity of the AMOC to changes in temperature and freshwater.**

*1.4 Results*
*I've to admit I'm not impressed by the capabilities of the emulator in reproducing the model outcome. As apparent from Fig. 7, the emulator is systematically underestimating AMOC reduction for RCP4.5 and RCP8.5 no melt, while then over-estimating it for RCP8.5 plus GIS (maybe due to non-linearities kicking in here and timescale issues discussed above?). The authors discussion of this simply stating that "It is, however, to be expected that a box-model does not completely capture the behavior of the AMOC as simulated with a higher order climate model" is clearly insufficient. In particular, as there have been much simpler AMOC emulators around that actually perform much better (also and in particularly an AMOC recovery, e.g. Schleussner et al. 2014).*

**Thanks for pointing this out. We agree that there are limitations to the AMOC-emulator and that because of choices that have been made, it appears that previous emulator perform better. There are, however, a number of important things to take into consideration. Firstly, one could perform the tuning on a single GCM forcing scenario and the result will be a closer fit between GCM and emulator AMOC. However, when choosing that approach, one is limited to applying the emulator to forcing scenarios close to the one used for tuning. By using a larger number of scenario in the tuning process, the emulator can be used to test the AMOC for a much larger range of scenarios, albeit at the cost of having larger discrepancies between GCM and emulator. We have added text along these lines to the manuscript (lines 23-33 page 10) "It is also worth noting that the fit for an individual simulation could be improved, for instance the AMOC-emulator does allow for a partial AMOC recovery as UVic shows for RCP4.5, but such an AMOC-emulator is not found through the SA tuning methodology in this example, because it would degrade the fit for the other scenarios and thus lead to an overall higher cost function." More discussion on this topic follows in Sect. 4 of the manuscript (lines 7-13 page 12) "Another important consideration when using the AMOC-emulator is the spread in GCM climate forcing scenarios that is included in the tuning process. When using only a single climate change scenario, a much better match can be obtained between the AMOC evolution given by the GCM and AMOC-emulator, however, the reliability of the AMOC-emulator will quickly decrease for different climate forcings. On the other hand, one could use a large number of climate change projections in the tuning process to**

obtain a lesser fit for individual scenarios, but an AMOC-emulator that is applicable to a much larger range of climate change scenarios. The best strategy to be follow strongly depends on the research question in mind."

Another issue to consider is the use of physics-based or statistical emulators. With a statistical AMOC emulator one could obtain better agreement between GCM and emulator, however, such a model cannot be used to extrapolate for larger forcings. With a physics-based AMOC-emulator one can have more confidence in the response to large forcings, for instance a complete AMOC shutdown, notwithstanding that also in this approach the uncertainty is likely to increase for forcings further away from those used for tuning. This is discussed in the final section of the updated manuscript (lines 1-8 page 12) "Overall, the predictive power of the AMOC-emulator is reasonable when one considers the simplicity of the AMOC box model, but forcing scenarios that are increasingly far away from the forcings that are used in tuning the AMOC-emulator, the predictive power decreases. A large advantage of using a physics-based AMOC-emulator that is tuned with larger large climate forcings, over the use of for instance a statistical AMOC-emulator, is that it projects the point after which the AMOC collapses and switches to an off state, as this is an integral part of the physics of the Stommel model. It is clear that using an AMOC-emulator introduces a new type of uncertainty into AMOC projections, however, for which level of added uncertainty an AMOC-emulator is still useful is a question that is difficult to address."

*The apparent oscillations in the emulator arising from a "too direct response" of the emulator towards multi-decadal surface temperature oscillations also merits more discussion.*

The origin of the oscillations is already mentioned in the manuscript (lines 17-18 page 9) "The UVic-based surface temperature evolution exhibits multi-decadal to centennial oscillations that result from global climate variability originating from the Southern Ocean" and we do not deem it necessary to discuss the resulting AMOC osculations in much detail as they are a feature of the forcing based on this particular climate model and not a feature of the AMOC-emulator. In the discussion section we have added some words describing the kind of temperature forcings that are appropriate to use (lines 15-17 page 12) "The assumptions behind the AMOC-emulator presented here, limit it to projecting AMOC changes on multi-decadal and larger timescales. Therefore, the applied GCM-based climate forcings and AMOC strength time series should be filtered to exclude high resolution variability."

*It is even worse for the predictions in Fig. 8. First of all, the figure is not well-labelled (no y-axis labeling, panels not clearly distinguishable, and what is given by the numbers 5,1,5?) and that there is no such thing as a top-middle panel for only two boxes.*

The conversion of the figure must have gone wrong at some point because the points raised by the reviewer are difficult to understand looking at the figures we have in the manuscript. We will ensure that the figures are correct in the next version.

*For none of the panels, the model actually captures key features. It fails to capture the bumps in the top-left and bottom right, and for the two other panels, it gets it wrong completely. I cannot agree to the author's conclusions that "Overall, the predictive power of the AMOC-emulator is good for reasonable forcing scenarios when one considers the simplicity of the model."*

We don't agree with the general notion given by the reviewer. Firstly, the AMOC-emulator is not designed to emulator decadal AMOC fluctuations as simulated by the GCM. As mentioned in the manuscript, those results from internal climate variability mostly originating from the Southern Ocean and it is not to be expected that the emulator captures those. Moreover, the focus of the AMOC emulator is on multi-decadal to multi-centennial scales, something that is now specifically

mentioned in the discussion (see reply above).

Furthermore, it is important to realize that the values given in figure 8 are anomalies with respect to the time series given in figure 7. Thus even the largest mismatch between GCM and AMOC-emulator (~1-2Sv in lower left panel) is 'only' an mismatch of 10-20%. We have added an objective assessment of the predictive power of the AMOC-emulator by comparing the results with a null-model that assumes that the emulator has no predictive power; it doesn't know if an additional forcing on top of the ones used in the tuning procedure would further increase or decrease the AMOC and would thus result in zero anomalies. This assessment shows that in three out of four cases the AMOC-emulator has substantial predictive power. We discuss this assessment in the manuscript (lines 20-30 page 11) "This is quantified by comparing the AMOC-emulator results with a null-model that assumes an AMOC-emulator with zero skill, meaning that it simply reproduces the original calibration data. The results from these experiments are shown as anomalies relative to the original scenario, the original being RCP8.5-GIS for RCP8.5x0.5-GIS, RCP8.5x1.5-GIS and RCP8.5-GISx1.5, and RCP4.5-GIS for RCP4.5-GISRCP8.5x1.5. We find that for large changes in the GHG forcing the Uvic-based AMOC-emulators are well capable of predicting the AMOC evolution of UVic in terms of sign and amplitude and perform better than the null-model (upper panels Fig. 8). For large changes in the applied GIS melt forcing the picture is more complex (lower panels Fig. 8). A strong increase in GIS melt under a low GHG scenario shows an excellent performance of the AMOC-emulator and a RSME that is much lower than for the null-model (RCP4.5-GISRCP-8.5x1.5 in Fig. 8), but for the high GHG scenario, a 50% increase in GIS melt leads to a deterioration of the fit between UVic and AMOC-emulator with consequently a larger RSME than that provided by the null-model (RCP8.5-GISx1.5 in Fig. 8). The latter shows that the UVic-based AMOC-emulators tend to overestimate the impact of GIS melt on the AMOC strength under high-end GHG scenarios. Summarizing, in all four cases the emulator predicts the correct sign of the AMOC response to changes in the forcings, and in three out of four cases the predictive power of the AMOC-emulator is better than of the null-model.". Nonetheless, it is important to acknowledge that using an emulator will introduce a new type of error in any assessment, pointed out by the following text in the manuscript (lines 5-7 page 12) "It is clear that using an AMOC-emulator introduces a new type of uncertainty into AMOC projections, however, for which level of added uncertainty an AMOC-emulator is still useful is a question that is difficult to address."

*1.5 Summary*
*Generally, I miss a section that reflects on the limitations and short-comings of the approach taken, given in particular the apparent limitations in reproducing the EMIC results. Furthermore, an outlook of where this can be applied and what it specific strengths are compared to other approaches should be included.*

Thanks for this comment. We agree that are more substantial and clear discussion is needed to make clear what the model can and cannot do. We have added the following to the discussion section (lines 1-22 page 12) "Overall, the predictive power of the AMOC-emulator is reasonable when one considers the simplicity of the AMOC box model, but for forcing scenarios that are increasingly far away from the forcings that are used in tuning the AMOC-emulator, the predictive power decreases. A large advantage of using a physics-based AMOC-emulator that is tuned with large climate forcings, over the use of for instance a statistical AMOC-emulator, is that it projects the point after which the AMOC collapses and switches to an off state, as this is an integral part of the physics of the Stommel model. It is clear that using an AMOC-emulator introduces new uncertainty into AMOC projections, however, for which level of added uncertainty an AMOC-emulator is still useful is a question that is difficult to address. Another important consideration when using the AMOC-emulator is the spread in GCM climate forcing

scenarios that is included in the tuning process. When using only a single climate change scenario, a  better match can be obtained between the AMOC evolution given by the GCM and AMOC-emulator, however, in this case the reliability of the AMOC-emulator will quickly decrease for different climate forcings. On the other hand, one could use a large number of climate change projections in the tuning process to obtain a lesser fit for individual scenarios, but an AMOC-emulator that is applicable to a much larger range of climate change scenarios. The best strategy to be followed strongly depends on the research question in mind. The assumptions behind the AMOC-emulator presented here, limit it to projecting AMOC changes on multi-decadal and larger timescales. Therefore, the applied GCM-based climate forcings and AMOC strength time series should best be filtered to exclude high resolution variability. Moreover, an AMOC-emulator that is tuned to specific GIS melt experiments is likely not applicable to experiments in which melt water is applied to a different geographical region or with a different seasonal cycle. This is not to say that the presented AMOC-emulator framework cannot equally be applied to other sources of melt water input. Finally, many processes that are known to impact the AMOC are not considered in the AMOC-emulator, for instance the impact of winds, gyre circulation, Southern Ocean upwelling or deep water formation outside of the North Atlantic (see Sect. 1). If such processes would prove to dominate the AMOC response to future climate change, a different AMOC box model should be considered that places emphasis on that particular process."

---

## Author Comment (AC4) · 18 Aug 2016

**Response to anonymous Referee #3**

**We thanks Reviewer 3 for the interesting and extensive comments on the manuscripts. Below we will provide a detailed response to all individual comments.**

*2 The model formulation*
*2.1 On using a conceptual model*
*I don't understand why this kind of ad-hoc box model is preferred. One would resort to Stommel's box model if one knows almost nothing about AMOC. Stommel's model was just conceptual, whose sole purpose is to get very rough ideas on how AMOC might work, and is not designed for the kind of quantitative modeling that the present authors pursue.*

*Stommel's box model was presented in 1961, and we know a lot more today. I would think that a simple dynamical model like Gnanadesikan's (1999, Science vol. 283, pp. 2077– ) is much better because even uncertain parameters are based on (that is, constrained by) clearly-identified dynamics. In contrast, some of the present authors' "parameterizations" aren't adequately defendable; they are based on hand-waving arguments. For example, how can one defend the parameterization that the AMOC strength is proportional to the interspheric surface-density difference? We know a lot better than that.*

*Perhaps even better, the models of Schloesser et al. (2014, Prog. Oceanogr. Vol. 120, pp. 154– ) and McCreary et al. (2016, Prog. Oceanogr. vol. 123, pp. 46– ) provide constraints among integral quantities, such as AMOC strength, thermocline depth, and meridional density difference, and hence can be utilized as a "box model". Those constrains are derived as solutions to dynamical equations rather than assumed on the basis of hand-waving arguments.*

*In short, I don't see any advantage today in utilizing an old conceptual model for quantitative prediction.*

*2.2 AMOC proportional to interhemispheric density difference?*
*The authors says that "the assumption that the meridional Atlantic density contrast between the North Atlantic and the South Atlantic is the first order driver of the AMOC" is debatable, but I think that's off the mark. The current wisdom is that the Southern-Ocean winds (and perhaps vertical diffusivity) are the first-order driver. They cite Butler et al. (2016) as the other side of the "debate" but Butler et al. do not argue that the surface meridional density gradient "drives" the AMOC. The just use density integrated twice in the vertical as a "diagnostic" of the AMOC.*
*It is clear from ocean GCM studies that the meridional density gradient is not the first order driver of AMOC. When the sea-surface density is restored toward a prescribed profile in an ocean-only GCM and windstress is changed in the Southern Ocean, the AMOC strength changes roughly linearly to the windstress. See Toggweiler et al. (1995, Dee-Sea Research vol. 42, pp.477– ) and the series of studies that follow. This is evidence enough that the interhemispheric density difference does not drive AMOC. Of course, this evidence is based on ocean-only models, and it is possible that the interhemispheric density difference is correlated with the AMOC strength through atmospheric feedbacks, but to use a one-to-one correspondence like (1) needs justification based on atmosphere-ocean coupled dynamics.*
*By the way, I found that Butler et al. (2016) still use the traditional hand-waving parameterization px=Lx / py=Ly. See Schloesser et al. (2012, Prog. Oceanogr. Vol. 101, pp. 33– ) for a better parameterization based a lot more on dynamics.*

**Thanks for describing in detail your view on conceptual models and the drivers of the AMOC. Indeed there is a wide variety of models of different complexity that describe (some aspect of) the AMOC, some more based on dynamic considerations than others, some including parameters**

that are more easily constrained by observations than others. In this study we have chosen to use one of the simplest and most established of such models around which to build our emulator framework, this is done for various reasons: i) it can easily be forced by temperature and Greenland melt outputs from a GCM; ii) it is easy to implement; iii) very fast to run; iv) easy to understand/diagnose the results; v) has several free parameters that can be tuned towards the behavior of a GCM in terms of the sensitivity of the AMOC to changes in heat and freshwater fluxes. We acknowledge that a different model could have been chosen, perhaps one that would turn out to perform better, however, we are not able try every single one of them so that remains unknown.

We do agree that when using the Stommel model to make the connection between changes in temperature and freshwater and changes in the AMOC strength implies that certain processes are not taken into account, like the role of changes in Southern Ocean winds, upwelling and deep water formation when projecting future changes in the AMOC. This is now more clearly described in the manuscript with the following in the introduction (lines 14-23 page 2) "At the center of our approach is the assumption that changes in AMOC strength are linearly related to changes in the Atlantic meridional density contrast. Since Stommel (1961) a large number of studies have provided evidence for an important role of the Atlantic meridional density contrast in driving AMOC changes (e.g. Rahmstorf, 1996; Gregory and Tailleux, 2011; Butler et al., 2016). Nonetheless, this model neglects several important processes, like the role of Southern Ocean upwelling, winds and deep water formation (e.g. Gnanadesikan, 1999; de Boer et al., 2010) and a unified theory describing the fundamental mechanisms driving and sustaining the AMOC lacks to this date (Lozier, 2010). Using a Stommel model to emulate AMOC changes driven by surface temperature and freshwater forcings seems appropriate in the light of present-day knowledge and the apparent leading role of surface buoyancy changes in simulated future AMOC weakening (IPCC Climate Change, 2013). Moreover, the model is easy to use, interpreted and can be forced directly with GCM-based forcing fields. Nonetheless, the processes that have been omitted and the simplicity of the model should be considered when interpreting the results." and the following in the discussion section (lines 20-24 page 12) "...many processes that are known to impact the AMOC are not considered in the AMOC-emulator, for instance the impact of winds, gyre circulation, Southern Ocean upwelling or deep water formation outside of the North Atlantic (see Sect. 1). If such processes would prove to dominate the AMOC response to future climate change, a different AMOC box model should be considered that places emphasis on that particular process."

However, we do not agree with the reviewers view that it is now well established how the AMOC works, what drives future AMOC changes and that the Stommel model has been shown to be wrong. Although we are not experts in the field of conceptual AMOC models and we do not plan to provide a discussion of all literature on the subject, we are not aware of any such consensus in the field. As becomes clear from the references cited above by the reviewer and by the references in the manuscript, many things are still debated and the Stommel model, that is the relationship between AMOC strength and meridional density differences, is still widely used to discuss the mechanisms and stability of the AMOC in complex GCMs.

*3 Tuning and validation*
*3.1 GCMs for tuning*
*Why aren't multiple coupled GCMs used to tune the parameters? Do the authors recommend that the AMOC-emulator be tuned differently for each model?*
The present manuscript aims to describe a modelling framework that can be used in combination with any GCM. Since the AMOC sensitivity to changes in heat and freshwater fluxes is strongly

**GCM-dependent, we indeed recommend that the AMOC-emulator is tuned separately for every GCM to reflect these differences.**

*The emulator is based on equations that represent physical processes in the real world. Then, if at all possible, the parameters should be tuned on the basis of reality. Granted that there is not enough data for the deep ocean. Then the second best thing is the publicly-available collections of coupled GCM runs. I think that studies have indicated that a multi-model ensemble is usually better than a single model to mimic reality. So, the tuned parameters would be more likely better if they are based on multiple models.*

**We agree that ideally one would tune the free parameters within bounds provided by observational data. However, since it is such a highly simplified and conceptual model, the parameters are not easily obtained from observations even if that data would be abundant. For future applications, we indeed recommend that a large number of AMOC-emulators is used, tuned towards different GCMs, in order to provide a range of parameter values that is hopefully as close to reality as possible.**

*3.2 Variables for tuning*
*The variables (salinity, ocean temperature, etc.) of the emulator should be compared with those from the GCM. It is possible that the state of the emulator is very different from that of the GCM even when the AMOC strength m agrees.*

*On a more basic note, have the authors made sure that all the variables of the emulator take reasonable values? I don't think it would be okay if, say, salinity takes a value of -100 psu even if the value of m is reasonable!*

*If the atmosphere and ocean states aren't realistic, how can we trust the emulator?*
*One approach to cope with this problem would be to include other variables than AMOC strength in the cost function. Another approach would be to compare various variables between the runs of the tuned model and those of the GCMs (part of validation). I think both are necessary.*

**Thanks for providing this interesting idea. First of all, we fully agree that one has to make sure that the values for salinity and temperature in the different parts of the box model are realistic. This has been checked. Indeed one could include the comparison between GCM and AMOC-emulator temperatures and salinity in the tuning procedure. However, there does not seem to be a reason why an AMOC emulator that has temperatures and salinities that are closer to the GCMs, would perform better in terms of the AMOC behavior, in fact the tuning of the parameters would very much be steered by getting the right temperatures and salinities and less so by the AMOC, thus likely deteriorating the capabilities of the AMOC-emulator in terms of mimicking the AMOC in the GCM. Since our focus is purely on providing a computational efficient method to provide uncertainty estimates of GCM AMOC projections, we deem the current approach most suited.**

*3.3 Models for validation*
*Moreover, I think the tuned emulator should be validated against another set of different models. Otherwise the validation isn't robust.*

**As discussed above, the sensitivity of the AMOC differs from one GCM to the next, therefore the AMOC-emulator should be tuned separately for every GCM and thus not be validated with results from a different GCM.**

*I also wonder if an ensemble of runs are necessary for the GCM (UVic) for tuning and validation. For*

*example, there is only one run for each case in Figure 8, but doesn't the AMOC strength differ from realization to realization? I don't know how chaotic the GCM is (because it uses a low-degree-of-freedom atmospheric model), but isn't the reality more or less chaotic?*

**Thanks for this question. This is indeed an interesting point. There are two points to this question. Firstly, like mentioned by the reviewer, Uvic is a low resolution GCM that is know to have less variability than higher resolution and complexity models. However, for high and low resolution models, the forced response of the AMOC to strong changes in temperature and freshwater is much stronger than internal variability, which is on the order of 1Sv on decadal and longer timescales. There are indications from observations that in the real world AMOC variability is substantially larger, however, those time series are currently too short to make any robust statements about this and, furthermore, this does not impact our GCM-based methodology.**

*4 Forcing*

*I may be missing something, but it's not clear to me what forces the emulator. The solar flux S seems constant in time (Table 1), but then how is the increase in green-house gas represented?*

**Thanks for this comment. The section on AMOC-emulator forcings describes the way we use GCM output to force the AMOC-emulator. Indeed the solar flux is constant in time. The GCM regional temperature changes, including the impact of increased green-house gasses and all feedbacks, are included in the AMOC-emulator through the so-called 'total atmosphere effect' parameter. A parameter of the atmosphere, "a temporal and spatial varying parameter that effectively combines atmospheric emissivity, the greenhouse effect and all other processes included in a GCM that cause regional temperatures to differ from global temperature changes". This way it is ensured that the ocean is regionally forced by almost the same temperature changes as in the GCM. Furthermore, the AMOC-emulator is forced by the GCM-based $F_{GIS}$ forcing.**

*If I understand it correctly, forces the model (equation 19) toward one particular GCM solution, but wouldn't it damp the emulator's variability? especially when the emulator is to simulate a state that is very different from the GCM state used for ? Doesn't this amount to building the solution into the simulation?*

**The GCM-based regional temperatures that are used as forcing, are used in the tuning phase. If one would like to use the AMOC-emulator to simulate a temperature forcing that is different (as is done in the section "Predictive power of the UVic-based AMOC-emulator"), the changes in the 'total atmosphere effect' parameter can be changed accordingly.**

*5 Minor points: math notations*

*5.1 Arrays The authors define boldface math symbols to mean "arrays", but I recommend avoiding this unconventional convention. For example, a multiplication of two "arrays" can mean several different things in conventional mathematics. Equation (13) includes the multiplication of the arrays S and p, which is meant to represent (S1p1; S2p2; : : :), which is hardly conventional. K and Ta have the same problem in (14). Also, equation (11) includes 1=z, by which the authors mean (1=z1; 1=z2; : : :), but which is not widely used in math.*

*5.2 Subscripts*

*I recommend using an upright font for multi-character math symbols such as "start" and "gcm"; or avoiding them. In particular, the subscript "it" looks as if it represented two subscripts i and t. I recommend using a single-character subscript, such as " i1" or if you insist on multi-character subscript, you may want " it□1" using an upright font.*

**Thanks for pointing out the issues with math notations and subscripts. Indeed the notations that are used are confusing and in some places wrong. We have updated the manuscript following the**

**recommendations of the reviewer.**

*6 Point by point comments*
*Some of the following comments support my arguments above, some raise other concerns, and others point out minor, mostly editorial, problems. I wrote many of them as I read the manuscript for the first time, and as a result, they include some redundancy. I leave them as they are, because they often reflect difficulties or problems the reader may encounter as she reads the text.*

*6.0.1 p. 1, l. 19:*
*"due to climate sensitivity, polar amplification, GIS melt and model dependent sensitivity*
*of the AMOC . . . "—I'm confused. Doesn't "climate sensitivity" include all the remaining items in the list? Why is it listed in parallel with the rest?*
**Thanks for pointing this out. Indeed taking climate sensitivity as the the global temperature change for a doubling of CO2, this term is mostly taken to include all other processes that are listed. However, in the model world this is not always the case. For instance, ice-sheet-climate interactions are mostly not considered and thus GIS melt not taken into account. Moreover, one can have the same climate sensitivity, but different polar amplification and the latter can result in a different AMOC response because of the sensitivity of the AMOC to latitudinal temperature differences. Why we prefer to list all of them separately in this context, is because GCMs differ in all those terms, and all those uncertainties can be tested individually with the AMOC-emulator.**

*6.0.2 p. 2, l. 5:*
*"(Rahmstorf and Willebrand, 1981)"—As the reference list indicates, this should probably be Rahmstorf and Willebrand (1995).*
**Thanks for pointing this out, it has been corrected.**

*6.0.3 p. 2, ll. 5 & 31:*
*"the so-called Bjerknes feedback"—Probably this is because I'm not much versed in climate research, but isn't the "so-called Bjerknes feedback" restricted along the equator? A direct overturning circulation occurs connecting cooling in the eastern Pacific, say, and warming in the western Pacific only along the equator, where the Coriolis force vanishes, and the surface windstress associated with this zonal overturning circulation enhances the upwelling of sea water, which further lowers the sea-surface temperature in the eastern Pacific—a positive feedback, which is "the so-called Bjerknes feedback".*
*The authors cite Rahmstorf and Willebrand (1995) for "the so-called Bjerknes feedback", but Rahmstorf and Willebrand proposed a negative feedback due to heat transport within the atmosphere, I think.*
**Indeed this topic is somewhat confusing as indeed the Bjerknes feedback often refers to the feedback described by the reviewer, but this term (or Bjerknes compensation) is also used to describe the compensation between meridional heat transport by the ocean and atmosphere as first proposed by Bjerknes in 1964. The latter indeed provides a stabilizing or negative feedback to AMOC changes.**

*6.0.4 p. 2, l. 7:*
*"tuning a numbrer of free parameters"—They aren't "free". They represent specific physical processes and hence must be ultimately determined by physics, even though it's in practice difficult to derive their values purely from physical principles.*
**The term 'free parameter' is used here to make the distinction between parameters that are prescribed and those that are not, or in other words, those that are part of the tuning process and**

those that are not. All of them represent physical processes and should (and often are) determined from observations.

*6.0.5 p. 3, l. 12:*
*Why is F prescribed? I would expect it to change according to the state of the climate system. What do IPCC-class coupled GCMs say about the change in F under global warming, for example? . . . but, later in the text, the authors say that F is related to the global atmospheric temperature (equation 12). So, it's not prescribed after all.*
**Thanks for pointing this out. The manuscript is not sufficiently clear on this topic and changes have been included for clarification. $F_i$ consists of two parts (equation 12), a part that is fixed in time ($F_{0i}$) and a part that is a function of global temperature changes ($h_i \Delta T_{glob}$). Both $F_i$ and $h_i$ c are part of the tuning process.**

*6.0.6 Equations (3)–(10): What does this "" mean? Is it a typo for "@"?*
**Thanks for pointing this out, we have updated the manuscript for clarity.**

*6.0.7 Equation (11):*
*State whether z's are fixed, and if so, give their values here or refer the reader to a table or something.*
**The formulation now includes a clear notation showing that $z$ is a function of $i$ and and a reference to Table 1 is given.**

*6.0.8 Equation (12):*
*Tglob should be defined. (How is it computed from Ta?)*
**This has been rewritten to read "global atmospheric surface temperature anomalies".**

*6.0.9 Equation (12):*
*Give F0's their values here or refer the reader to a table or something.*
**This line now reads "Freshwater fluxes F01 , F02 and coefficients h1 and h2 are included in the tuning procedure (Tab. 2)."**

*6.0.10 Equation (13):*
*I may be mistaken, but it seems that the Ta4 is the only nonlinear term. Doesn't it make sense if this term is linearized around a mean state?*
**Yes it could be, but we prefer to keep the current form.**

*6.0.11 Equation (13):*
*The solar flux S is a confusing notation. By the authors' own convention, S =(S0; S1; S2; S3; S4), which uses the same symbols as salinity.*
**Thanks for pointing this out, this is indeed confusing. The notation has been changed to read $I_i$.**

*6.0.12 Equation (13):*
*The solar flux S should be discussed right below equation (13). Does it depend on time? Table 1 suggests that it's constant in time but that should be stated explicitly. So, does the emulator solves only for annual averages?*
**The first line after equation 13 now reads "where σ is the Stefan Boltzmann constant and $I_i$ and $\alpha_i$ the latitude dependent yearly mean incoming shortwave radiation and planetary albedo, respectively (see Tab. 1 for details)".**

*6.0.13 Equation (13):*

*Define this symbol precisely. (But I don't recommend this notation because a gradient of an array is a strange mathematical entity.)*
**Thanks for pointing this out. It should have been a Delta symbol.**

*6.0.14 Equation (14):*
*What does this "" mean? Is it a typo for "@"?*
**This has been corrected.**

*6.0.15 Equation (14):*
*State that Ha is defined to vanish at the northern and southern ends of the northernmost and southernmost boxes. (I guess they are so defined, right?)*
**The following has been added "Meridional heat fluxes are assumed zero at the northern and southern boundaries of the domain."**

*6.0.16 p. 5, l. 15:*
*I guess we need some discussion on other possible sets of tuning parameters. We have a vast range of possibilities. Then, how have we settled on these seven parameters? Have the authors tried other combinations of parameters?*
**We deem this discussed by the line (lines 25-27 page 6) "This selection of parameters is somewhat subjective, but it proved a good balance between, on the one hand, sufficient degrees of freedom to tune the AMOC emulator's behavior towards that of a specific GCM and, on the other hand, the efficiency to find optimal parameter fits."**

*6.0.17 p. 5, l. 15:*
*F and h are related by equation (12), and so cannot be determined independently. Moreover, if you tune F, you can forget about equation (12) and don't need to consider h.*
**We apologize for the confusion that has arisen because of errors in the notation. This has been corrected in the text and figures. $F_{0i}$ and $h_i$ are the parameters used in the tuning process. The former giving the steady state meridional freshwater transport by both the atmosphere and the wind driven ocean part, while the latter controls the changes in atmospheric transport as a function of global temperature changes.**

*6.0.18 Equation (15):*
*Why try to optimize m alone? It's conceivable that widely different states have similar m values. Because we have other variables like salinity, we could choose better sets of parameter values, if we include other variables in the cost function, couldn't we?*
**As discussed above, we don't think that including more variables in the tuning process would lead to a better behavior of the AMOC-emulator in terms of its capacity to mimic the GCMs AMOC sensitivity to changes in temperature and freshwater.**

*6.0.19 Equation (15):*
*I may well be mistaken, but it seems that the differential equations are linear in the tuning parameters and if so, the optimization problem on the new cost function ...is a quadratic function of the parameters and can be solved analytically, I think.*
**The optimization of the parameters cannot be solved analytically as the system is non-linear and includes 7 parameters that influence each other.**

*6.0.20 Equation (16):*
*The notation "pstart(1□z)" is confusing because it looks as if pstart(z) were a function of z. Vectors*

*customarily come after scalars, as in "(1 ▢ z)pstart"*
**It has been changed.**

*6.0.21 Step 1:*
*I don't understand why we have to repeat this step. Why not choose values that are within the ranges in Table 2 in the first place? We can use a random variable whose PDF is uniform over the specified range for each parameter, can't we? I mean, if (1 ▢ z)p1 is below the range, we can just use p1min for the lower bound; that is, we can use U(max((1 ▢ z)p1; p1min); min((1 + z)p1; p1max)) without repetition. The same argument holds for the last part of Step 3.*
**The solution given by the referee will not always give the same results. More specifically, the p1min and p1max values would be used much more often than other values in case random values from outside of the range are often picked. Another solution would be to split $z$ into $z_{min}$ and $z_{max}$ and adjust those values for every parameter to ensure that the randomly picked values are never outside of the imposed ranges. We don't think it matters which solution is picked.**

*6.0.22 Equation (17):*
*I may be missing something, but shouldn't (16) and (17) be written in parallel forms? If we write (1 ▢ z)p for (16), then we should write (1 ▢ it)p for (17). If we write p ▢ itp for (17), we should write p ▢ zp for (16). For a moment, I was confused with (17).*
**Thanks for pointing this out, using the same notation for both equations indeed improves readability.**

*6.0.23 p. 7, l. 16:*
*I think that efforts should be made to narrow the range of the parameter values. If parameters are widely different even though the cost function is similar, doesn't that suggest that the parameters aren't well tuned?*
*What about comparing variables other than m between the emulator and the GCM? Wouldn't that tell which parameter values are bad?*
*It seems that the authors have forgotten that there is only one reality.*
**Firstly, we assume the reviewer is pointing towards 'GCM reality' in this comment, since we do not aim to work towards a single parameter set that provides the closest resemblance to the real world AMOC, no matter how much we would like to do so. However, also when we are talking about using an emulator to mimic the complex AMOC behavior in a GCM, we do not expect that there is a single parameter set that provides the perfect match between GCM and emulator; because of the highly simplified nature of the emulator, it cannot be determined which parameter set is closest to 'GCM reality'.**

*6.0.24 p. 7, l. 22:*
*What is "RCP"? (I may have missed its definition given in the text.) Because how is determined is important, it may be helpful to give a bit more information here.*
**This line reads "RCP4.5 and RCP8.5 (Representative Concentration Pathways; Meinshausen et al., 2011)", which we deem sufficient information, especially since in the context of testing the AMOC-emulator, the exact imposed climate forcings are only of secondary importance.**

*6.0.25 Equation (19):*
*Does the model really use the full time-series of Tgcm? Or is that a long-term mean? State clearly how is Tgcm defined. If the emulator uses the full time-series, it may not be appropriate for other models or for other scenarios.*
**Thanks for pointing this out. $T_{GCM}$ can in principle be of any temporal resolution. However, the**

**model aims at resolving AMOC changes on decadal and longer timescales and as such, including high frequency variability in the temperature forcing could lead to misinterpretation of the results. We have added a discussion point in the final section of the manuscript to clarify the strengths and weaknesses of the AMOC-emulator (lines 15-17 page 12) "The assumptions behind the AMOC-emulator presented here, limit it to projecting AMOC changes on multi-decadal and larger timescales. Therefore, the applied GCM-based climate forcings and AMOC strength time series should be filtered to exclude high resolution variability. ".**

*6.0.26 p. 8, l. 2:*
*"Note that the temperature forcing files need to be interpolated onto the temporal resolution used in the atmospheric component of the AMOC-emulator"—Awkward in several counts.*
*1. The interpolation draws the attention of the reader as if it were something noteworthy. Perhaps the results are sensitive to the method of interpolation? The reader would wonder.*
*2. Is the fact that the GCM data are saved in files noteworthy? (I mean, why mention the files at all?)*
*3. Despite this cautious tone, the interval at which the GCM data is saved is not indicated.*
*If the result is sensitive to the interpolation, give more details. If not, what about just saying, "The GCM variables are saved at an interval of XXX hours and interpolated on to the time steps of the AMOC-emulator", something along the lines.*
**The line has been removed.**

*6.0.27 p. 8, ll. 5–6:*
*A similar problem. If interpolation is so noteworthy, give more details. If it's not so big a deal, just say, "the GIS melt forcing is interpolated. . . ." instead of "Note that the GIS melt forcing needs to be interpolated. . . ."*
**The line has been removed.**

---

## Editor Comment (EC2) · A. Yool (Editor) · 26 Aug 2016

Dear authors,

Thank you for responding to your referees and for the revised manuscript.

I have now had a chance to review both these responses and uploaded revised manuscript. In general, many of the responses to points raised by the referees are clear to me. However, there are a number where it is unclear what - if anything - has been done in the revised manuscript to address them.

I am attaching your response to referees as a supplement with a few comments added where I am unclear on the form of your response (i.e. has anything within the document

changed). I have also made a few passing suggestions that may (or may not) be helpful.

I would be grateful if you could revise your response, taking my comments into account. This would be extremely helpful when reaching a decision on the revised manuscript.

As ever, if you have any questions, please do not hesitate to contact me.

With best regards,

Andrew.

Please also note the supplement to this comment:
http://www.geosci-model-dev-discuss.net/gmd-2016-79/gmd-2016-79-EC2-supplement.pdf

[Figure]

**Supplement:**

**Response to anonymous Referee #1**

**We thanks Reviewer 1 for the interesting and extensive comments on the manuscripts. Below we will provide a detailed response to all individual comments.**

*I don't understand the reasoning for why you generate 100 reasonable fits and then select only the 10 best fits (P7 L15). Firstly, this emulator/box-model should be cheap to run, so why choose such small numbers? Surely ensembles of order 10,000 or 100,000 are more reasonable. Secondly, the choice of 10 best fits seems to narrow the ranges of several parameters (e.g. V4, F1, h1). By doing this you rule out large regions of parameter space that give perfectly reasonable fits, and could behave differently under different forcing scenarios. If the primary aim is to assess uncertainty in AMOC projections I would expect to see a rigorous analysis of the uncertainty. By discarding large areas of parameter space uncertainty will certainly be underestimated.*

**Thanks for raising this valid point. Our aim here is not to provide an uncertainty assessment of AMOC projections, it is to provide a method with which one could do this, as for example done in the manuscript by Bakker et al. under review in GRL, now also pointed out in the last line of the main manuscript (lines 1-2 page 13) "The AMOC-emulator is a valuable tool to study the uncertainty in GCM-based AMOC projections, such as the one recently being performed on the**
[Figure]
 **results from the AMOCMIP project (Bakker et al., 2016)." The assessment referred to here is based on multiple GCMs, decreasing the need for a large number of AMOC-emulators for a single GCM.**

**With regard to the point that the AMOC-emulator is cheap and could thus be run for tens or even hundreds of thousands of times. This is very true, however, it takes many time steps and iterations to find a single reasonable fit. We have now included the following description to the manuscript (line 30 page 8 to line 2 page 9) "To provide an idea of the computational expenses of the model we provide a back of the envelope calculation. This shows that a single run over all scenarios takes $10^5$ time steps which are done in about 5 seconds. You need on the order of 400 iterations (in which parameter values are perturbed) to find a single reasonable fit, resulting in approximately half an hour to calculate a single reasonable fit on a normal desktop computer." This shows that by using more powerful computers and or running in parallel the number of reasonable fits could be enhanced, but it shows that 10,000 to 100,000 reasonable fits is ambitious nonetheless.**

*In the four scenarios not seen by the emulator (Fig. 8) the behavior of UVic is clearly not captured in two cases (lower left and upper right). There are no confidence intervals plotted (or computed as far as I can tell), but I believe the GCM would lie well outside 2 standard errors in those two cases. Therefore, the GCM would still need to be run for any untested scenario. I would not trust the emulator in its current form.*

**We don't agree with the general notion given by the reviewer. Firstly, it is important to realize that the values given in figure 8 are anomalies with respect to the time series given in figure 7. Thus even the largest mismatch between GCM and AMOC-emulator (~1-2Sv in lower left panel) is 'only' a mismatch of 10-20%. We have added an objective assessment of the predictive power of the AMOC-emulator by comparing the results with a null-model that assumes that the emulator has no predictive power; it doesn't know if an additional forcing on top of the ones used in the tuning procedure would further increase or decrease the AMOC and would thus result in zero anomalies. This assessment shows that in three out of four cases the AMOC-emulator has substantial predictive power. We discuss this assessment in the manuscript (lines 20-30 page 11) "This is quantified by comparing the AMOC-emulator results with a null-model**

that assumes an AMOC-emulator with zero skill, meaning that it simply reproduces the original calibration data. The results from these experiments are shown as anomalies relative to the original scenario, the original being RCP8.5-GIS for RCP8.5x0.5-GIS, RCP8.5x1.5-GIS and RCP8.5-GISx1.5, and RCP4.5-GIS for RCP4.5-GISRCP8.5x1.5. We find that for large changes in the GHG forcing the Uvic-based AMOC-emulators are well capable of predicting the AMOC evolution of UVic in terms of sign and amplitude and perform better than the null-model (upper panels Fig. 8). For large changes in the applied GIS melt forcing the picture is more complex (lower panels Fig. 8). A strong increase in GIS melt under a low GHG scenario shows an excellent performance of the AMOC-emulator and a RSME that is much lower than for the null-model (RCP4.5-GISRCP-8.5x1.5 in Fig. 8), but for the high GHG scenario, a 50% increase in GIS melt leads to a deterioration of the fit between UVic and AMOC-emulator with consequently a larger RSME than that provided by the null-model (RCP8.5-GISx1.5 in Fig. 8). The latter shows that the UVic-based AMOC-emulators tend to overestimate the impact of GIS melt on the AMOC strength under high-end GHG scenarios. Summarizing, in all four cases the emulator predicts the correct sign of the AMOC response to changes in the forcings, and in three out of four cases the predictive power of the AMOC-emulator is better than of the null-model.". Nonetheless, it is important to acknowledge that using an emulator will introduce a new type of error in any assessment, pointed out by the following text in the manuscript (lines 5-7 page 12) "It is clear that using an AMOC-emulator introduces a new type of uncertainty into AMOC projections, however, for which level of added uncertainty an AMOC-emulator is still useful is a question that is difficult to address."

*On multidecadal timescales the emulator is plagued by sensitivity to surface temperature oscillations. These seem to have arisen from the addition of the atmospheric boxes to the ocean box model published by Zickfeld et al., 2004. Can the authors confirm that this is the case, and if so can they control this sensitivity, e.g. by introducing a damping/mixing term?*

[Figure]

The multidecadal AMOC oscillations result from the UVic-based regional temperature forcings of the AMOC-emulator and thus in turn to internal variability of UVic. Zickfeld et al. (2004) applied highly idealized linear temperature increases of global temperature, thus not including any multi-decadal variability. On the contrary, in our approach we directly use regional GCM-based temperature time series to force the AMOC-emulator. In this way the forcing not only takes into account the GCMs global climate sensitivity, but also mechanisms like polar amplification etc. that cause regional temperature change differences. This method also introduces any multi-decadal internal variability that might exist in a GCM into the AMOC-emulator when expressed in regional temperature time series. We acknowledge this feature, but do not see it as an issue.

*If the authors have good reason to retain this behavior they need to test the sensitivity to the phase of the variability. For all of the scenarios, the chosen start date (2006) appears to be shortly after a peak in the strong multidecadal variability, so the AMOC is preconditioned to decline at this time. Under all scenarios the AMOC in the 'best' emulators appear to decline faster than the UVic model. Consequently, the SA tuning and the cost function used may be adversely affected by this multidecadal variability.*

[Figure]
Indeed, following from the usage of the Stommel model to emulate the AMOC, multi-decadal temperature variability and its phasing impact the projected AMOC changes, in the AMOC-emulator, in UVic and most likely also in reality. Perhaps the AMOC response in the AMOC-emulator to regional temperature changes is too direct (as mentioned in the manuscript) and thus
 the importance of multi-decadal variability overestimated, but we don't see this as a major issue. It seems to us that the years before 2006 represent in fact a time of relatively weak AMOC, not

**strong, thus preconditioning the AMOC-emulator to a somewhat weaker response to global change. We don't agree with the notion of the reviewer that the decline in the emulators is faster than in UVic, they seem very similar to us. Finally, multi-decadal AMOC variability only impacts the absolute value of the cost function, not the resulting optimal fits.**

*On centennial timescales the emulator (as currently presented) does not capture crucial features of the AMOC response to the forcing (Fig. 7). In particular I would draw attention to the RCP4.5 scenarios, in which the GCM exhibits a strong reduction followed by a steady recovery. The emulator fails to identify either the timing or amplitude of the AMOC minimum and it fails to identify the recovery phase. In addition it appears to show signs of a recovery phase under RCP8.5 when UVic shows none. The authors state (P9 L28) that the fit can be improved, but that this would entail a higher overall cost function for the SA tuning method. Is this indicative of a poor choice of cost function? Does it mean that the box model should be tuned separately for each scenario?*

**The failure of the AMOC emulator to capture the slight recovery of the AMOC under RCP4.5 is indeed an issue and shows the limitations of the simple box model to capture all complex feedbacks in the GCM. Indeed, as mentioned in the manuscript, the AMOC-emulator does allow for an AMOC recovery under RCP4.5, but that would mean a large deterioration of the fit of the AMOC emulator to the AMOC in RCP8.5 and thus it would increase the value of the cost function, for which reason this solution is not found through this approach. It is an interesting point if the AMOC-emulator should be tuned separately for each scenario. We added the following to the manuscript to cover this issue (lines 23-33 page 10) "It is also worth noting that the fit for an individual simulation could be improved, for instance the AMOC-emulator does allow for a partial AMOC recovery as UVic shows for RCP4.5, but such an AMOC-emulator is not found through the SA tuning methodology in this example, because it would degrade the fit for the other scenarios and thus lead to an overall higher cost function." More discussion on this topic follows in Sect. 4 of the manuscript (lines 7-13 page 12) "Another important consideration when using the AMOC-emulator is the spread in GCM climate forcing scenarios that is included in the tuning process. When using only a single climate change scenario, a better match can be obtained between the AMOC evolution given by the GCM and AMOC-emulator, however, the reliability of the AMOC-emulator will quickly decrease for different climate forcings. On the other hand, one could use a large number of climate change projections in the tuning process to obtain a lesser fit for individual scenarios, but an AMOC-emulator that is applicable to a much larger range of climate change scenarios. The best strategy to be follow strongly depends on the research question in mind."**

*A far more substantial summary is required. For example, the emulator's limitations need to be clearly stated (and whether/how the authors think these can be addressed). For what purposes are the emulator suitable in its current form, and for what purposes might it be useful subject to further work? With the current analysis, I disagree with the statement that "the UVic-based AMOC-emulator captures well the overall characteristics of the multi-centennial response of the AMOC".*

**Thanks for this comment. We agree that are more substantial and clear discussion is needed to make clear what the model can and cannot do. We have added the following to the discussion section (lines 1-22 page 12) "Overall, the predictive power of the AMOC-emulator is reasonable when one considers the simplicity of the AMOC box model, but for forcing scenarios that are increasingly far away from the forcings that are used in tuning the AMOC-emulator, the predictive power decreases. A large advantage of using a physics-based AMOC-emulator that is tuned with large climate forcings, over the use of for instance a statistical AMOC-emulator, is that it projects the point after which the AMOC collapses and switches to an off state, as this is an integral part of the physics of the Stommel model. It is clear that using an AMOC-emulator**

[Figure]

introduces new uncertainty into AMOC projections, however, for which level of added uncertainty an AMOC-emulator is still useful is a question that is difficult to address. Another important consideration when using the AMOC-emulator is the spread in GCM climate forcing scenarios that is included in the tuning process. When using only a single climate change scenario, a better match can be obtained between the AMOC evolution given by the GCM and AMOC-emulator, however, in this case the reliability of the AMOC-emulator will quickly decrease for different climate forcings. On the other hand, one could use a large number of climate change projections in the tuning process to obtain a lesser fit for individual scenarios, but an AMOC-emulator that is applicable to a much larger range of climate change scenarios. The best strategy to be followed strongly depends on the research question in mind. The assumptions behind the AMOC-emulator presented here, limit it to projecting AMOC changes on multi-decadal and larger timescales. Therefore, the applied GCM-based climate forcings and AMOC strength time series should best be filtered to exclude high frequency variability. Moreover, an AMOC-emulator that is tuned to specific GIS melt experiments is likely not applicable to experiments in which melt water is applied to a different geographical region or with a different seasonal cycle. This is not to say that the presented AMOC-emulator framework cannot equally be applied to other sources of melt water input. Finally, many processes that are known to impact the AMOC are not considered in the AMOC-emulator, for instance the impact of winds, gyre circulation, Southern Ocean upwelling or deep water formation outside of the North Atlantic (see Sect. 1). If such processes would prove to dominate the AMOC response to future climate change, a different AMOC box model should be considered that places emphasis on that particular process."

*Minor comments:-*
*Page 3 Line 12: Prescribed FW fluxes: F1 and F2 are tuned parameters. I would have expected these to vary as a function of the forcing/climate. What is the justification for fixing them?*
This part was not sufficiently clear in the manuscript and has now been updated. The total freshwater fluxes F1 and F2 are not part of the tuning procedure, but F01 and F02 (the combined wind-driven oceanic and atmospheric meridional freshwater fluxes for the reference state are). The text should have read (line 18 page 4) "Freshwater fluxes $F_{01}$ , $F_{02}$ and coefficients $h_1$ and $h_2$ are included in the tuning procedure (Tab. 2)"

*Page 5 Line 10: What you also fail to consider are nonlinearities between these parameters. Co-varying the parameters in Tables 1 and 2 could yield very different behaviours.*
The parameter fitting method we employ, simulated annealing, randomly varies the individual parameters, thus considering (although not explicitly) both linear and nonlinear relationships between parameters. Moreover, by including Figure 6 we perform a first order test to see whether relationships exist between parameters, which indeed is the case for several of them.

*Page 5 Line 30: algorith > algorithm*
Thank you, it has been corrected.

*Page 6 Line 4: I find the arbitrary choice of +/- 200% rather strange. What is the justification for this?*
We agree that this choice is arbitrary. Our approach has been to take this arbitrary value, perform the analysis and then to analyze whether or not all parameter values that resulted from the fitting procedure were well within the +-200% range (see also figure 6). From this it was decided to keep the +-200% value. This point is clarified by adding (lines 14-15 page 7) "The appropriateness of this arbitrary range of initial parameter values is later verified by ensuring that all final parameter values are well within the initial range."

*Page 6 Line 8: analogues > analogous*
**Thank you, it has been corrected.**

*Check typesetting in Tables (e.g. Table 1 column 2)*
**Thank you, typesetting is checked.**

*Table 1: (typo) dependend > dependent*
**Thank you, it has been corrected.**

*Check typesetting on Figure 8: it appears corrupted.*
**Thank you, typesetting is checked.**

*Figure 4 caption: (typo) relatvie > relative*
**Thank you, it has been corrected.**

*Figure 8 caption: (typo) calculate > calculated*
**Thank you, it has been corrected.**

*Figure 8 caption: (typo) rigth > right*
**Thank you, it has been corrected.**

**Response to anonymous Referee #2**

**We thanks Reviewer 2 for the interesting and extensive comments on the manuscripts. Below we will provide a detailed response to all individual comments.**

*1 General Comments*
*1.1 Introduction*
*The introduction does not sufficiently reflect the state of the literature on AMOC and in particularly not on conceptual models such as the Stommel-Model to study it.*
*Our understanding of AMOC dynamics has advanced considerably over the last years thanks to ongoing observations e.g. in the Rapid array (see Srokosz & Byrden, Science 2015). In addition a recent studies has suggest that the AMOC might already be in decline (Rahmstorf et al. 2015). While of course not directly relevant for the emulator itself, such observational findings need to be discussed in an approach that emulate AMOC behaviour over the next centuries. This should also include a discussion of atmospheric imprints on the AMOC e.g. such as atmospheric blocking events. It should also allow to assess the performance of GCMs in relation to the observational record.*

**We cannot agree with the points raised above. We are presenting a new modeling framework in a journal for model development. As such, we agree that some insights as to why we think a new modeling framework is needed is called for, but a discussion of observed AMOC changes, AMOC fingerprints or the performance of GCMs in relation to the observational record does not seem appropriate in this journal and is beyond the scope of this paper..**

*Much more important though is the discussion in relation to the emulator approach taken. Stommel type models have been used since quite some time and might be able to capture key dynamics of the AMOC (e.g. bistability). However, they at the same time have faced a lot of criticism and alternative models describing AMOC behaviour exist. This is in particular related to the relevance of Southern Ocean upwelling reflected in a conceptual model by Gnanadesikan (1999) related to changes in the pycnocline depth. A dynamic that is completely missing in the Stommel approach.*
*This has been explored further in conceptual models and attempts exist to unify pycnocline and freshwater-feedback dynamics. In this context, the authors should consider the work of Sijp et al. (2012) that they may find helpful.*
*Another question directly relating to the physical plausibility of the Stommel model relates to the relationship of circulation strength and meridional density gradient in a geostrophic ocean. The authors should consider work by Gregoy & Tailleux (2010) that present a kinetic energy approach essential providing a physical explanation for the (empirically supported) meridional density gradient outlining the relevance of the Western Boundary Current in modelling AMOC dynamics.*
*These comments should not be seen as undermining the Stommel model approach taken here, but they need to be addressed. In short, the authors should show motivate their approach in the light of the most recent literature.*

**Thanks for pointing this out and we agree that a more thorough discussion of the pro's and con's of the used Stommel model is called for. An important caveat of using a Stommel model is that Southern Ocean upwelling, the role of Southern Hemisphere mid-latitude winds and other processes are neglected, a point that we have added to the discussion of this manuscript. Our choice to use the Stommel model was driven by two considerations. Firstly, to our knowledge no unified simple AMOC model exists and as such it is not clear if other models are better or worse than the Stommel model in relating surface temperature and freshwater flux changes to the AMOC strength. Secondly, the Stommel model allows for rather straightforward inclusion of temperature and freshwater forcings based on GCM simulations, while for other models like the ones mentioned above it is not clear to us how this could be done. Finally, it is important to note**

that we did not set out to construct a new simple model that describes the main dynamics of the AMOC, but rather to use an existing model and build a framework around it that can easily be applied to GCM climate change and AMOC projections.

Following the above, we have updated the introduction to read (lines 14-23 page 2) "At the center of our approach is the assumption that changes in AMOC strength are linearly related to changes in the Atlantic meridional density contrast. Since Stommel (1961) a large number of studies have provided evidence for an important role of the Atlantic meridional density contrast in driving AMOC changes (e.g. Rahmstorf, 1996; Gregory and Tailleux, 2011; Butler et al., 2016). Nonetheless, it neglects several important processes, like the role of Southern Ocean upwelling, winds and deep water formation (e.g. Gnanadesikan, 1999; de Boer et al., 2010) and a unified theory describing the fundamental mechanisms driving and sustaining the AMOC lacks to this date (Lozier, 2010). Using a Stommel model to emulate AMOC changes driven by surface temperature and freshwater forcings seems appropriate in the light of present-day knowledge and the apparent leading role of surface buoyancy changes in simulated future AMOC weakening (IPCC Climate Change, 2013). Moreover, the model is easy to use, interpreted and can be forced directly with GCM-based forcing fields. Nonetheless, the processes that have been omitted and the simplicity of the model should be considered when interpreting the results."
To the discussion section we have added (lines 20-24 page 12) "...many processes that are known to impact the AMOC are not considered in the AMOC-emulator, for instance the impact of winds, gyre circulation, Southern Ocean upwelling or deep water formation outside of the North Atlantic (see Sect. 1). If such processes would prove to dominate the AMOC response to future climate change, a different AMOC box model should be considered that places emphasis on that particular process."

*1.2 The Emulator model*
*Here, the work dominantly builds on a previous model by Zickfeld et al. (2004) plus a representation of the Bjerknes feedback. It does however not become sufficiently clear, why this addition will represent a substantial advancement. The authors show the differences in Fig. 9 and describe that this will represent a negative feedback on the AMOC dynamics. But it's not clear, if Figure 9 shows two sets calibrated individually (with and without atmospheric feedback) or just from the optimal parameter set with this feedback switched on and off. Therefore, I cannot judge if the conclusion drawn by the authors on the importance of the effect are due to their specific parameter set or not.*
*It would add merit, if the authors could show that the model including the Bjerknes effect will in the end outperform the no-atmospheric feedback model in the fitting procedure. This would also justify, why there model is actually better than the one presented in Zickfeld (2004).*
**Thanks for providing this comment. Firstly, the main improvement with respect to the Zickfeld et al. (2004) model is that we provide a framework that allows one to use limited GCM AMOC and climate projections to tune an AMOC-emulator in order to perform an uncertainty analysis. The Zickfeld et al. (2004) approach used a full ~20.000yr long hysteresis simulation to tune their emulator, not feasible for most IPCC-type GCMs. Moreover, they did not force their emulator with GCM-based temperature changes or consider inter-GCM differences in regional temperature changes. Those are the features we see as most important changes with respect to earlier work, a view that is now better reflected in the introduction of the manuscript by (line 23 page 1 to line 3 page 2) "To this end we developed an AMOC-emulator framework. It entails a simple box model that uses physical relationships to represent the most important mechanisms and feedbacks that govern the AMOC's response to changes in regional surface temperatures, freshwater fluxes and enhanced melting of the GIS. The AMOC-emulator can be forced by temperature and melt water fluxes from any GCM, and using AMOC time series the free**

**parameters of the box model are tuned to mimic the GCM's AMOC sensitivity to future climate change." and in later on in the introduction it reads (lines 11-13 page 2) "the approach described here is designed specifically to allow future studies in which a limited number of climate projections from multiple GCMs, limited in the simulated forcing scenarios and simulwation length, to be combined into a Bayesian framework of century time-scale probabilistic AMOC projections."**

**With respect to the added stabilizing Bjerkness feedback, it indeed appears from Figure 9 that it's impact is limited. Figure 9 shows results for the same parameter sets with this feedback switched on and off, allowing for a direct investigation of its impact. Nonetheless, we deem the model including this feedback more realistic. Moreover, the effect is non-negligible (lines 1-6 page 11) "The impact of including atmospheric meridional heat transport is a small, but non-negligible ~1Sv strengthening of the control state of the AMOC (not shown) and, more importantly, a slightly lower sensitivity to changes in radiative forcing and GIS melt (Fig. 9). This confirms our understanding of atmospheric meridional heat transport acting as a negative feedback to AMOC changes. The simulations with the atmospheric feedback included have on average a stronger AMOC by 8.1±1.9% (μ ± σ; calculated over all 10 best fits and over all five forcing scenarios)."**

*Furthermore, the model includes 5 atmospheric boxes. Why are 5 boxes needed and not 3 to resolve the meridional heat transport? I think that can be easily motivated and maybe I missed it. Maybe it's worth considering to restructure the approach by moving subsection 2.3 further up to discuss the setup of the atmospheric forcings.*

[Figure]

**Including high latitude atmospheric boxes allows us to have a closed energy budget and more realistic meridional atmospheric heat transport.**

**Thanks for the suggestion to rearrange this section. We have accordingly switched sections 2.2 and 2.3.**

*In this context, the authors should also reflect on the limitations of the model to reproduce transient AMOC changes that relate to the assumption of well-mixed density within the boxes. This might be in particularly relevant in relation to the Greenland freshwater input. Clearly, this represents an over-simplification and may substantially limit the capabilities of this approach to emulate transient behaviour (I'll further comment on this below).*

[Figure]

**Thanks for pointing this out. We fully agree that a box model can never resolve the complexities of the interaction between Greenland meltwater and the ocean. We have experimented with an additional tuning parameter to include the GCM dependent 'efficiency' of Greenland meltwater to impact the density of the North Atlantic ocean box, but decided against it since the current 7 tuning parameters already allow for sufficient freedom to tune an AMOC-emulator towards the AMOC sensitivity of a specific GCM.**

*1.3 The tuning to complex model output*
*In the manuscript, the model is tuned to an EMIC model UVIC. I think that's generally no problem, but somehow contradicts the initial claims by the authors that this emulator could now be used to run larger ensembles. What is it exactly that the emulator provides that cannot be done with an EMIC?*
*In general terms, the strength of an emulator is it's capability to include projections from a range of different models. We have AMOC projections for several CMIP5 models, why is it not applied to those?*
*In addition, there are the AMOC sensitivity studies by Gregory et al. (2005) and Stouffer et al. (2006) that would provide enough runs to calibrate the model. Why isn't it applied to those runs?*
*In addition, the authors mention the AMOCMIP project. Can the emulator be applied to the AMOCMIP output?*

**Thanks for these comments. It has become clear from the comments of the different reviewers that the aim of this manuscript is not sufficiently clear and we have changed the abstract, introduction and summary sections to improve on this. In this manuscript we want to describe a modeling framework that allows one to use limited GCM output to tune and force an AMOC box model that can in turn be used to perform uncertainty analysis. It is not the aim of this manuscript to provide future AMOC projections or provide such an uncertainty analysis. See also the responses provided above.**

*I checked the project homepage and understood that the AMOCMIP will explictly resolve different Greenland basins separately. Is that correct? If so, and following recent findings that it actually matters a lot for North Atlantic dynamics where the freshwater is actually applied, will this emulator be the best tool to reproduce these dynamics? Or should it maybe consist of a subpolar (Labrador Sea) and North Atlantic box? And/or should conceptual models of convection in marginal seas e.g. by Spall (2004) and Straneo (2009) be integrated?*

**Thanks for this question. Indeed the aim of the simulations in AMOCMIP is to provide 'realistic' Greenland melt scenarios and to apply those to IPCC-type climate change projections. This includes explicitly resolving spatial and seasonal differences in the meltwater flux. Such details cannot be captured by the AMOC-emulator. However, as described above, by tuning the AMOC-emulator to the forcings and AMOC projections of a specific GCM, we take into account the inter-GCM differences in the sensitivity of the AMOC to changes in temperature and freshwater.**

*1.4 Results*
*I've to admit I'm not impressed by the capabilities of the emulator in reproducing the model outcome. As apparent from Fig. 7, the emulator is systematically underestimating AMOC reduction for RCP4.5 and RCP8.5 no melt, while then over-estimating it for RCP8.5 plus GIS (maybe due to non-linearities kicking in here and timescale issues discussed above?). The authors discussion of this simply stating that "It is, however, to be expected that a box-model does not completely capture the behavior of the AMOC as simulated with a higher order climate model" is clearly insufficient. In particular, as there have been much simpler AMOC emulators around that actually perform much better (also and in particularly an AMOC recovery, e.g. Schleussner et al. 2014).*

[Figure]
 **Thanks for pointing this out. We agree that there are limitations to the AMOC-emulator and that because of choices that have been made, it appears that previous emulator perform better. There are, however, a number of important things to take into consideration. Firstly, one could perform the tuning on a single GCM forcing scenario and the result will be a closer fit between GCM and emulator AMOC. However, when choosing that approach, one is limited to applying the emulator to forcing scenarios close to the one used for tuning. By using a larger number of scenario in the tuning process, the emulator can be used to test the AMOC for a much larger range of scenarios, albeit at the cost of having larger discrepancies between GCM and emulator. We have added text along these lines to the manuscript (lines 23-33 page 10) "It is also worth noting that the fit for an individual simulation could be improved, for instance the AMOC-emulator does allow for a partial AMOC recovery as UVic shows for RCP4.5, but such an AMOC-emulator is not found through the SA tuning methodology in this example, because it would degrade the fit for the other scenarios and thus lead to an overall higher cost function." More discussion on this topic follows in Sect. 4 of the manuscript (lines 7-13 page 12) "Another important consideration when using the AMOC-emulator is the spread in GCM climate forcing scenarios that is included in the tuning process. When using only a single climate change scenario, a much better match can be obtained between the AMOC evolution given by the GCM and AMOC-emulator, however, the reliability of the AMOC-emulator will quickly decrease for different climate forcings. On the other hand, one could use a large number of climate change projections in the tuning process to**

obtain a lesser fit for individual scenarios, but an AMOC-emulator that is applicable to a much larger range of climate change scenarios. The best strategy to be follow strongly depends on the research question in mind."

Another issue to consider is the use of physics-based or statistical emulators. With a statistical AMOC emulator one could obtain better agreement between GCM and emulator, however, such a model cannot be used to extrapolate for larger forcings. With a physics-based AMOC-emulator one can have more confidence in the response to large forcings, for instance a complete AMOC shutdown, notwithstanding that also in this approach the uncertainty is likely to increase for forcings further away from those used for tuning. This is discussed in the final section of the updated manuscript (lines 1-8 page 12) "Overall, the predictive power of the AMOC-emulator is reasonable when one considers the simplicity of the AMOC box model, but forcing scenarios that are increasingly far away from the forcings that are used in tuning the AMOC-emulator, the predictive power decreases. A large advantage of using a physics-based AMOC-emulator that is tuned with larger large climate forcings, over the use of for instance a statistical AMOC-emulator, is that it projects the point after which the AMOC collapses and switches to an off state, as this is an integral part of the physics of the Stommel model. It is clear that using an AMOC-emulator introduces a new type of uncertainty into AMOC projections, however, for which level of added uncertainty an AMOC-emulator is still useful is a question that is difficult to address."

*The apparent oscillations in the emulator arising from a "too direct response" of the emulator towards multi-decadal surface temperature oscillations also merits more discussion.*

The origin of the oscillations is already mentioned in the manuscript (lines 17-18 page 9) "The UVic-based surface temperature evolution exhibits multi-decadal to centennial oscillations that result from global climate variability originating from the Southern Ocean" and we do not deem it necessary to discuss the resulting AMOC osculations in much detail as they are a feature of the forcing based on this particular climate model and not a feature of the AMOC-emulator. In the discussion section we have added some words describing the kind of temperature forcings that are appropriate to use (lines 15-17 page 12) "The assumptions behind the AMOC-emulator presented here, limit it to projecting AMOC changes on multi-decadal and larger timescales. Therefore, the applied GCM-based climate forcings and AMOC strength time series should be filtered to exclude high resolution variability."

*It is even worse for the predictions in Fig. 8. First of all, the figure is not well-labelled (no y-axis labeling, panels not clearly distinguishable, and what is given by the numbers 5,1,5?) and that there is no such thing as a top-middle panel for only two boxes.*

The conversion of the figure must have gone wrong at some point because the points raised by the reviewer are difficult to understand looking at the figures we have in the manuscript. We will ensure that the figures are correct in the next version.

*For none of the panels, the model actually captures key features. It fails to capture the bumps in the top-left and bottom right, and for the two other panels, it gets it wrong completely. I cannot agree to the author's conclusions that "Overall, the predictive power of the AMOC-emulator is good for reasonable forcing scenarios when one considers the simplicity of the model."*

We don't agree with the general notion given by the reviewer. Firstly, the AMOC-emulator is not designed to emulator decadal AMOC fluctuations as simulated by the GCM. As mentioned in the manuscript, those results from internal climate variability mostly originating from the Southern Ocean and it is not to be expected that the emulator captures those. Moreover, the focus of the AMOC emulator is on multi-decadal to multi-centennial scales, something that is now specifically

mentioned in the discussion (see reply above).

Furthermore, it is important to realize that the values given in figure 8 are anomalies with respect to the time series given in figure 7. Thus even the largest mismatch between GCM and AMOC-emulator (~1-2Sv in lower left panel) is 'only' an mismatch of 10-20%. We have added an objective assessment of the predictive power of the AMOC-emulator by comparing the results with a null-model that assumes that the emulator has no predictive power; it doesn't know if an additional forcing on top of the ones used in the tuning procedure would further increase or decrease the AMOC and would thus result in zero anomalies. This assessment shows that in three out of four cases the AMOC-emulator has substantial predictive power. We discuss this assessment in the manuscript (lines 20-30 page 11) "This is quantified by comparing the AMOC-emulator results with a null-model that assumes an AMOC-emulator with zero skill, meaning that it simply reproduces the original calibration data. The results from these experiments are shown as anomalies relative to the original scenario, the original being RCP8.5-GIS for RCP8.5x0.5-GIS, RCP8.5x1.5-GIS and RCP8.5-GISx1.5, and RCP4.5-GIS for RCP4.5-GISRCP8.5x1.5. We find that for large changes in the GHG forcing the Uvic-based AMOC-emulators are well capable of predicting the AMOC evolution of UVic in terms of sign and amplitude and perform better than the null-model (upper panels Fig. 8). For large changes in the applied GIS melt forcing the picture is more complex (lower panels Fig. 8). A strong increase in GIS melt under a low GHG scenario shows an excellent performance of the AMOC-emulator and a RSME that is much lower than for the null-model (RCP4.5-GISRCP-8.5x1.5 in Fig. 8), but for the high GHG scenario, a 50% increase in GIS melt leads to a deterioration of the fit between UVic and AMOC-emulator with consequently a larger RSME than that provided by the null-model (RCP8.5-GISx1.5 in Fig. 8). The latter shows that the UVic-based AMOC-emulators tend to overestimate the impact of GIS melt on the AMOC strength under high-end GHG scenarios. Summarizing, in all four cases the emulator predicts the correct sign of the AMOC response to changes in the forcings, and in three out of four cases the predictive power of the AMOC-emulator is better than of the null-model.". Nonetheless, it is important to acknowledge that using an emulator will introduce a new type of error in any assessment, pointed out by the following text in the manuscript (lines 5-7 page 12) "It is clear that using an AMOC-emulator introduces a new type of uncertainty into AMOC projections, however, for which level of added uncertainty an AMOC-emulator is still useful is a question that is difficult to address."

*1.5 Summary*
*Generally, I miss a section that reflects on the limitations and short-comings of the approach taken, given in particular the apparent limitations in reproducing the EMIC results. Furthermore, an outlook of where this can be applied and what it specific strengths are compared to other approaches should be included.*

Thanks for this comment. We agree that are more substantial and clear discussion is needed to make clear what the model can and cannot do. We have added the following to the discussion section (lines 1-22 page 12) "Overall, the predictive power of the AMOC-emulator is reasonable when one considers the simplicity of the AMOC box model, but for forcing scenarios that are increasingly far away from the forcings that are used in tuning the AMOC-emulator, the predictive power decreases. A large advantage of using a physics-based AMOC-emulator that is tuned with large climate forcings, over the use of for instance a statistical AMOC-emulator, is that it projects the point after which the AMOC collapses and switches to an off state, as this is an integral part of the physics of the Stommel model. It is clear that using an AMOC-emulator introduces new uncertainty into AMOC projections, however, for which level of added uncertainty an AMOC-emulator is still useful is a question that is difficult to address. Another important consideration when using the AMOC-emulator is the spread in GCM climate forcing

scenarios that is included in the tuning process. When using only a single climate change scenario, a  better match can be obtained between the AMOC evolution given by the GCM and AMOC-emulator, however, in this case the reliability of the AMOC-emulator will quickly decrease for different climate forcings. On the other hand, one could use a large number of climate change projections in the tuning process to obtain a lesser fit for individual scenarios, but an AMOC-emulator that is applicable to a much larger range of climate change scenarios. The best strategy to be followed strongly depends on the research question in mind. The assumptions behind the AMOC-emulator presented here, limit it to projecting AMOC changes on multi-decadal and larger timescales. Therefore, the applied GCM-based climate forcings and AMOC strength time series should best be filtered to exclude high resolution variability. Moreover, an AMOC-emulator that is tuned to specific GIS melt experiments is likely not applicable to experiments in which melt water is applied to a different geographical region or with a different seasonal cycle. This is not to say that the presented AMOC-emulator framework cannot equally be applied to other sources of melt water input. Finally, many processes that are known to impact the AMOC are not considered in the AMOC-emulator, for instance the impact of winds, gyre circulation, Southern Ocean upwelling or deep water formation outside of the North Atlantic (see Sect. 1). If such processes would prove to dominate the AMOC response to future climate change, a different AMOC box model should be considered that places emphasis on that particular process."

**Response to anonymous Referee #3**

**We thanks Reviewer 3 for the interesting and extensive comments on the manuscripts. Below we will provide a detailed response to all individual comments.**

*2 The model formulation*
*2.1 On using a conceptual model*
*I don't understand why this kind of ad-hoc box model is preferred. One would resort to Stommel's box model if one knows almost nothing about AMOC. Stommel's model was just conceptual, whose sole purpose is to get very rough ideas on how AMOC might work, and is not designed for the kind of quantitative modeling that the present authors pursue.*

*Stommel's box model was presented in 1961, and we know a lot more today. I would think that a simple dynamical model like Gnanadesikan's (1999, Science vol. 283, pp. 2077– ) is much better because even uncertain parameters are based on (that is, constrained by) clearly-identified dynamics. In contrast, some of the present authors' "parameterizations" aren't adequately defendable; they are based on hand-waving arguments. For example, how can one defend the parameterization that the AMOC strength is proportional to the interspheric surface-density difference? We know a lot better than that.*

*Perhaps even better, the models of Schloesser et al. (2014, Prog. Oceanogr. Vol. 120, pp. 154– ) and McCreary et al. (2016, Prog. Oceanogr. vol. 123, pp. 46– ) provide constraints among integral quantities, such as AMOC strength, thermocline depth, and meridional density difference, and hence can be utilized as a "box model". Those constrains are derived as solutions to dynamical equations rather than assumed on the basis of hand-waving arguments.*

*In short, I don't see any advantage today in utilizing an old conceptual model for quantitative prediction.*

*2.2 AMOC proportional to interhemispheric density difference?*
*The authors says that "the assumption that the meridional Atlantic density contrast between the North Atlantic and the South Atlantic is the first order driver of the AMOC" is debatable, but I think that's off the mark. The current wisdom is that the Southern-Ocean winds (and perhaps vertical diffusivity) are the first-order driver. They cite Butler et al. (2016) as the other side of the "debate" but Butler et al. do not argue that the surface meridional density gradient "drives" the AMOC. The just use density integrated twice in the vertical as a "diagnostic" of the AMOC.*
*It is clear from ocean GCM studies that the meridional density gradient is not the first order driver of AMOC. When the sea-surface density is restored toward a prescribed profile in an ocean-only GCM and windstress is changed in the Southern Ocean, the AMOC strength changes roughly linearly to the windstress. See Toggweiler et al. (1995, Dee-Sea Research vol. 42, pp.477– ) and the series of studies that follow. This is evidence enough that the interhemispheric density difference does not drive AMOC. Of course, this evidence is based on ocean-only models, and it is possible that the interhemispheric density difference is correlated with the AMOC strength through atmospheric feedbacks, but to use a one-to-one correspondence like (1) needs justification based on atmosphere-ocean coupled dynamics.*
*By the way, I found that Butler et al. (2016) still use the traditional hand-waving parameterization $px=Lx$ / $py=Ly$. See Schloesser et al. (2012, Prog. Oceanogr. Vol. 101, pp. 33– ) for a better parameterization based a lot more on dynamics.*
**Thanks for describing in detail your view on conceptual models and the drivers of the AMOC. Indeed there is a wide variety of models of different complexity that describe (some aspect of) the AMOC, some more based on dynamic considerations than others, some including parameters**

that are more easily constrained by observations than others. In this study we have chosen to use one of the simplest and most established of such models around which to build our emulator framework, this is done for various reasons: i) it can easily be forced by temperature and Greenland melt outputs from a GCM; ii) it is easy to implement; iii) very fast to run; iv) easy to understand/diagnose the results; v) has several free parameters that can be tuned towards the behavior of a GCM in terms of the sensitivity of the AMOC to changes in heat and freshwater fluxes. We acknowledge that a different model could have been chosen, perhaps one that would turn out to perform better, however, we are not able try every single one of them so that remains unknown.

We do agree that when using the Stommel model to make the connection between changes in temperature and freshwater and changes in the AMOC strength implies that certain processes are not taken into account, like the role of changes in Southern Ocean winds, upwelling and deep water formation when projecting future changes in the AMOC. This is now more clearly described in the manuscript with the following in the introduction (lines 14-23 page 2) "At the center of our approach is the assumption that changes in AMOC strength are linearly related to changes in the Atlantic meridional density contrast. Since Stommel (1961) a large number of studies have provided evidence for an important role of the Atlantic meridional density contrast in driving AMOC changes (e.g. Rahmstorf, 1996; Gregory and Tailleux, 2011; Butler et al., 2016). Nonetheless, this model neglects several important processes, like the role of Southern Ocean upwelling, winds and deep water formation (e.g. Gnanadesikan, 1999; de Boer et al., 2010) and a unified theory describing the fundamental mechanisms driving and sustaining the AMOC lacks to this date (Lozier, 2010). Using a Stommel model to emulate AMOC changes driven by surface temperature and freshwater forcings seems appropriate in the light of present-day knowledge and the apparent leading role of surface buoyancy changes in simulated future AMOC weakening (IPCC Climate Change, 2013). Moreover, the model is easy to use, interpreted and can be forced directly with GCM-based forcing fields. Nonetheless, the processes that have been omitted and the simplicity of the model should be considered when interpreting the results." and the following in the discussion section (lines 20-24 page 12) "...many processes that are known to impact the AMOC are not considered in the AMOC-emulator, for instance the impact of winds, gyre circulation, Southern Ocean upwelling or deep water formation outside of the North Atlantic (see Sect. 1). If such processes would prove to dominate the AMOC response to future climate change, a different AMOC box model should be considered that places emphasis on that particular process."

[Figure]

However, we do not agree with the reviewers view that it is now well established how the AMOC works, what drives future AMOC changes and that the Stommel model has been shown to be wrong. Although we are not experts in the field of conceptual AMOC models and we do not plan to provide a discussion of all literature on the subject, we are not aware of any such consensus in the field. As becomes clear from the references cited above by the reviewer and by the references in the manuscript, many things are still debated and the Stommel model, that is the relationship between AMOC strength and meridional density differences, is still widely used to discuss the mechanisms and stability of the AMOC in complex GCMs.

*3 Tuning and validation*
*3.1 GCMs for tuning*
*Why aren't multiple coupled GCMs used to tune the parameters? Do the authors recommend that the AMOC-emulator be tuned differently for each model?*
The present manuscript aims to describe a modelling framework that can be used in combination with any GCM. Since the AMOC sensitivity to changes in heat and freshwater fluxes is strongly

[Figure]

**GCM-dependent, we indeed recommend that the AMOC-emulator is tuned separately for every GCM to reflect these differences.**

*The emulator is based on equations that represent physical processes in the real world. Then, if at all possible, the parameters should be tuned on the basis of reality. Granted that there is not enough data for the deep ocean. Then the second best thing is the publicly-available collections of coupled GCM runs. I think that studies have indicated that a multi-model ensemble is usually better than a single model to mimic reality. So, the tuned parameters would be more likely better if they are based on multiple models.*

**We agree that ideally one would tune the free parameters within bounds provided by observational data. However, since it is such a highly simplified and conceptual model, the parameters are not easily obtained from observations even if that data would be abundant. For future applications, we indeed recommend that a large number of AMOC-emulators is used, tuned towards different GCMs, in order to provide a range of parameter values that is hopefully as close to reality as possible.**

*3.2 Variables for tuning*
*The variables (salinity, ocean temperature, etc.) of the emulator should be compared with those from the GCM. It is possible that the state of the emulator is very different from that of the GCM even when the AMOC strength m agrees.*
*On a more basic note, have the authors made sure that all the variables of the emulator take reasonable values? I don't think it would be okay if, say, salinity takes a value of -100 psu even if the value of m is reasonable!*

[Figure]

*If the atmosphere and ocean states aren't realistic, how can we trust the emulator?*
*One approach to cope with this problem would be to include other variables than AMOC strength in the cost function. Another approach would be to compare various variables between the runs of the tuned model and those of the GCMs (part of validation). I think both are necessary.*

**Thanks for providing this interesting idea. First of all, we fully agree that one has to make sure that the values for salinity and temperature in the different parts of the box model are realistic. This has been checked. Indeed one could include the comparison between GCM and AMOC-emulator temperatures and salinity in the tuning procedure. However, there does not seem to be a reason why an AMOC emulator that has temperatures and salinities that are closer to the GCMs, would perform better in terms of the AMOC behavior, in fact the tuning of the parameters would very much be steered by getting the right temperatures and salinities and less so by the AMOC, thus likely deteriorating the capabilities of the AMOC-emulator in terms of mimicking the AMOC in the GCM. Since our focus is purely on providing a computational efficient method to provide uncertainty estimates of GCM AMOC projections, we deem the current approach most suited.**

*3.3 Models for validation*
*Moreover, I think the tuned emulator should be validated against another set of different models. Otherwise the validation isn't robust.*

[Figure]

**As discussed above, the sensitivity of the AMOC differs from one GCM to the next, therefore the AMOC-emulator should be tuned separately for every GCM and thus not be validated with results from a different GCM.**

*I also wonder if an ensemble of runs are necessary for the GCM (UVic) for tuning and validation. For*

*example, there is only one run for each case in Figure 8, but doesn't the AMOC strength differ from realization to realization? I don't know how chaotic the GCM is (because it uses a low-degree-of-freedom atmospheric model), but isn't the reality more or less chaotic?*

**Thanks for this question. This is indeed an interesting point. There are two points to this question. Firstly, like mentioned by the reviewer, Uvic is a low resolution GCM that is know to have less variability than higher resolution and complexity models. However, for high and low resolution models, the forced response of the AMOC to strong changes in temperature and freshwater is much stronger than internal variability, which is on the order of 1Sv on decadal and longer timescales. There are indications from observations that in the real world AMOC variability is substantially larger, however, those time series are currently too short to make any robust statements about this and, furthermore, this does not impact our GCM-based methodology.**

*4 Forcing*

*I may be missing something, but it's not clear to me what forces the emulator. The solar flux S seems constant in time (Table 1), but then how is the increase in green-house gas represented?*

**Thanks for this comment. The section on AMOC-emulator forcings describes the way we use GCM output to force the AMOC-emulator. Indeed the solar flux is constant in time. The GCM regional temperature changes, including the impact of increased green-house gasses and all feedbacks, are included in the AMOC-emulator through the so-called 'total atmosphere effect' parameter. A parameter of the atmosphere, "a temporal and spatial varying parameter that effectively combines atmospheric emissivity, the greenhouse effect and all other processes included in a GCM that cause regional temperatures to differ from global temperature changes". This way it is ensured that the ocean is regionally forced by almost the same temperature changes as in the GCM. Furthermore, the AMOC-emulator is forced by the GCM-based $F_{GIS}$ forcing.**

*If I understand it correctly, forces the model (equation 19) toward one particular GCM solution, but wouldn't it damp the emulator's variability? especially when the emulator is to simulate a state that is very different from the GCM state used for ? Doesn't this amount to building the solution into the simulation?*

**The GCM-based regional temperatures that are used as forcing, are used in the tuning phase. If one would like to use the AMOC-emulator to simulate a temperature forcing that is different (as is done in the section "Predictive power of the UVic-based AMOC-emulator"), the changes in the 'total atmosphere effect' parameter can be changed accordingly.**

*5 Minor points: math notations*

*5.1 Arrays The authors define boldface math symbols to mean "arrays", but I recommend avoiding this unconventional convention. For example, a multiplication of two "arrays" can mean several different things in conventional mathematics. Equation (13) includes the multiplication of the arrays S and p, which is meant to represent (S1p1; S2p2; : : :), which is hardly conventional. K and Ta have the same problem in (14). Also, equation (11) includes 1=z, by which the authors mean (1=z1; 1=z2; : : :), but which is not widely used in math.*

*5.2 Subscripts*

*I recommend using an upright font for multi-character math symbols such as "start" and "gcm"; or avoiding them. In particular, the subscript "it" looks as if it represented two subscripts i and t. I recommend using a single-character subscript, such as " i1" or if you insist on multi-character subscript, you may want " it□1" using an upright font.*

**Thanks for pointing out the issues with math notations and subscripts. Indeed the notations that are used are confusing and in some places wrong. We have updated the manuscript following the**

**recommendations of the reviewer.**

*6 Point by point comments*
*Some of the following comments support my arguments above, some raise other concerns, and others point out minor, mostly editorial, problems. I wrote many of them as I read the manuscript for the first time, and as a result, they include some redundancy. I leave them as they are, because they often reflect difficulties or problems the reader may encounter as she reads the text.*

*6.0.1 p. 1, l. 19:*
*"due to climate sensitivity, polar amplification, GIS melt and model dependent sensitivity of the AMOC . . . "—I'm confused. Doesn't "climate sensitivity" include all the remaining items in the list? Why is it listed in parallel with the rest?*
**Thanks for pointing this out. Indeed taking climate sensitivity as the the global temperature change for a doubling of CO2, this term is mostly taken to include all other processes that are listed. However, in the model world this is not always the case. For instance, ice-sheet-climate interactions are mostly not considered and thus GIS melt not taken into account. Moreover, one can have the same climate sensitivity, but different polar amplification and the latter can result in a different AMOC response because of the sensitivity of the AMOC to latitudinal temperature differences. Why we prefer to list all of them separately in this context, is because GCMs differ in all those terms, and all those uncertainties can be tested individually with the AMOC-emulator.**

*6.0.2 p. 2, l. 5:*
*"(Rahmstorf and Willebrand, 1981)"—As the reference list indicates, this should probably be Rahmstorf and Willebrand (1995).*
**Thanks for pointing this out, it has been corrected.**

*6.0.3 p. 2, ll. 5 & 31:*
*"the so-called Bjerknes feedback"—Probably this is because I'm not much versed in climate research, but isn't the "so-called Bjerknes feedback" restricted along the equator? A direct overturning circulation occurs connecting cooling in the eastern Pacific, say, and warming in the western Pacific only along the equator, where the Coriolis force vanishes, and the surface windstress associated with this zonal overturning circulation enhances the upwelling of sea water, which further lowers the sea-surface temperature in the eastern Pacific—a positive feedback, which is "the so-called Bjerknes feedback".*
*The authors cite Rahmstorf and Willebrand (1995) for "the so-called Bjerknes feedback", but Rahmstorf and Willebrand proposed a negative feedback due to heat transport within the atmosphere, I think.*
**Indeed this topic is somewhat confusing as indeed the Bjerknes feedback often refers to the feedback described by the reviewer, but this term (or Bjerknes compensation) is also used to describe the compensation between meridional heat transport by the ocean and atmosphere as first proposed by Bjerknes in 1964. The latter indeed provides a stabilizing or negative feedback to AMOC changes.**

*6.0.4 p. 2, l. 7:*
*"tuning a numbrer of free parameters"—They aren't "free". They represent specific physical processes and hence must be ultimately determined by physics, even though it's in practice difficult to derive their values purely from physical principles.*
**The term 'free parameter' is used here to make the distinction between parameters that are prescribed and those that are not, or in other words, those that are part of the tuning process and**

**those that are not. All of them represent physical processes and should (and often are) determined from observations.**

*6.0.5 p. 3, l. 12:*
*Why is F prescribed? I would expect it to change according to the state of the climate system. What do IPCC-class coupled GCMs say about the change in F under global warming, for example? . . . but, later in the text, the authors say that F is related to the global atmospheric temperature (equation 12). So, it's not prescribed after all.*
**Thanks for pointing this out. The manuscript is not sufficiently clear on this topic and changes have been included for clarification. $F_i$ consists of two parts (equation 12), a part that is fixed in time ($F_{0i}$) and a part that is a function of global temperature changes ( $h_i \Delta T_{glob}$). Both $F_i$ and $h_i$ c are part of the tuning process.**

*6.0.6 Equations (3)–(10): What does this "" mean? Is it a typo for "@"?*
**Thanks for pointing this out, we have updated the manuscript for clarity.**

*6.0.7 Equation (11):*
*State whether z's are fixed, and if so, give their values here or refer the reader to a table or something.*
**The formulation now includes a clear notation showing that $z$ is a function of $i$ and and a reference to Table 1 is given.**

*6.0.8 Equation (12):*
*Tglob should be defined. (How is it computed from Ta?)*
**This has been rewritten to read "global atmospheric surface temperature anomalies".**

*6.0.9 Equation (12):*
*Give F0's their values here or refer the reader to a table or something.*
**This line now reads "Freshwater fluxes F01 , F02 and coefficients h1 and h2 are included in the tuning procedure (Tab. 2)."**

*6.0.10 Equation (13):*
*I may be mistaken, but it seems that the Ta4 is the only nonlinear term. Doesn't it make sense if this term is linearized around a mean state?*
**Yes it could be, but we prefer to keep the current form.**

*6.0.11 Equation (13):*
*The solar flux S is a confusing notation. By the authors' own convention, S =(S0; S1; S2; S3; S4), which uses the same symbols as salinity.*
**Thanks for pointing this out, this is indeed confusing. The notation has been changed to read $I_i$.**

*6.0.12 Equation (13):*
*The solar flux S should be discussed right below equation (13). Does it depend on time? Table 1 suggests that it's constant in time but that should be stated explicitly. So, does the emulator solves only for annual averages?*
**The first line after equation 13 now reads "where σ is the Stefan Boltzmann constant and $I_i$ and $\alpha_i$ the latitude dependent yearly mean incoming shortwave radiation and planetary albedo, respectively (see Tab. 1 for details)".**

*6.0.13 Equation (13):*

*Define this symbol precisely. (But I don't recommend this notation because a gradient of an array is a strange mathematical entity.)*
**Thanks for pointing this out. It should have been a Delta symbol.**

*6.0.14 Equation (14):*
*What does this "" mean? Is it a typo for "@"?*
**This has been corrected.**

*6.0.15 Equation (14):*
*State that Ha is defined to vanish at the northern and southern ends of the northernmost and southernmost boxes. (I guess they are so defined, right?)*
**The following has been added "Meridional heat fluxes are assumed zero at the northern and southern boundaries of the domain."**

*6.0.16 p. 5, l. 15:*
*I guess we need some discussion on other possible sets of tuning parameters. We have a vast range of possibilities. Then, how have we settled on these seven parameters? Have the authors tried other combinations of parameters?*
**We deem this discussed by the line (lines 25-27 page 6) "This selection of parameters is somewhat subjective, but it proved a good balance between, on the one hand, sufficient degrees of freedom to tune the AMOC emulator's behavior towards that of a specific GCM and, on the other hand, the efficiency to find optimal parameter fits."**

*6.0.17 p. 5, l. 15:*
*F and h are related by equation (12), and so cannot be determined independently. Moreover, if you tune F, you can forget about equation (12) and don't need to consider h.*
**We apologize for the confusion that has arisen because of errors in the notation. This has been corrected in the text and figures. $F_{0i}$ and $h_i$ are the parameters used in the tuning process. The former giving the steady state meridional freshwater transport by both the atmosphere and the wind driven ocean part, while the latter controls the changes in atmospheric transport as a function of global temperature changes.**

*6.0.18 Equation (15):*
*Why try to optimize m alone? It's conceivable that widely different states have similar m values. Because we have other variables like salinity, we could choose better sets of parameter values, if we include other variables in the cost function, couldn't we?*
**As discussed above, we don't think that including more variables in the tuning process would lead to a better behavior of the AMOC-emulator in terms of its capacity to mimic the GCMs AMOC sensitivity to changes in temperature and freshwater.**

*6.0.19 Equation (15):*
*I may well be mistaken, but it seems that the differential equations are linear in the tuning parameters and if so, the optimization problem on the new cost function ...is a quadratic function of the parameters and can be solved analytically, I think.*
**The optimization of the parameters cannot be solved analytically as the system is non-linear and includes 7 parameters that influence each other.**

*6.0.20 Equation (16):*
*The notation "pstart(1□z)" is confusing because it looks as if pstart(z) were a function of z. Vectors*

*customarily come after scalars, as in "(1 □ z)pstart"*
**It has been changed.**

*6.0.21 Step 1:*
*I don't understand why we have to repeat this step. Why not choose values that are within the ranges in Table 2 in the first place? We can use a random variable whose PDF is uniform over the specified range for each parameter, can't we? I mean, if (1 □ z)p1 is below the range, we can just use p1min for the lower bound; that is, we can use U(max((1 □ z)p1; p1min); min((1 + z)p1; p1max)) without repetition. The same argument holds for the last part of Step 3.*
**The solution given by the referee will not always give the same results. More specifically, the p1min and p1max values would be used much more often than other values in case random values from outside of the range are often picked. Another solution would be to split $z$ into $z_{min}$ and $z_{max}$ and adjust those values for every parameter to ensure that the randomly picked values are never outside of the imposed ranges. We don't think it matters which solution is picked.**

*6.0.22 Equation (17):*
*I may be missing something, but shouldn't (16) and (17) be written in parallel forms? If we write (1 □ z)p for (16), then we should write (1 □ it)p for (17). If we write p □ itp for (17), we should write p □ zp for (16). For a moment, I was confused with (17).*
**Thanks for pointing this out, using the same notation for both equations indeed improves readability.**

*6.0.23 p. 7, l. 16:*
*I think that efforts should be made to narrow the range of the parameter values. If parameters are widely different even though the cost function is similar, doesn't that suggest that the parameters aren't well tuned?*
*What about comparing variables other than m between the emulator and the GCM? Wouldn't that tell which parameter values are bad?*
*It seems that the authors have forgotten that there is only one reality.*
**Firstly, we assume the reviewer is pointing towards 'GCM reality' in this comment, since we do not aim to work towards a single parameter set that provides the closest resemblance to the real world AMOC, no matter how much we would like to do so. However, also when we are talking about using an emulator to mimic the complex AMOC behavior in a GCM, we do not expect that there is a single parameter set that provides the perfect match between GCM and emulator; because of the highly simplified nature of the emulator, it cannot be determined which parameter set is closest to 'GCM reality'.**

*6.0.24 p. 7, l. 22:*
*What is "RCP"? (I may have missed its definition given in the text.) Because how is determined is important, it may be helpful to give a bit more information here.*
**This line reads "RCP4.5 and RCP8.5 (Representative Concentration Pathways; Meinshausen et al., 2011)", which we deem sufficient information, especially since in the context of testing the AMOC-emulator, the exact imposed climate forcings are only of secondary importance.**

*6.0.25 Equation (19):*
*Does the model really use the full time-series of Tgcm? Or is that a long-term mean? State clearly how is Tgcm defined. If the emulator uses the full time-series, it may not be appropriate for other models or for other scenarios.*
**Thanks for pointing this out. $T_{GCM}$ can in principle be of any temporal resolution. However, the**

**model aims at resolving AMOC changes on decadal and longer timescales and as such, including high frequency variability in the temperature forcing could lead to misinterpretation of the results. We have added a discussion point in the final section of the manuscript to clarify the strengths and weaknesses of the AMOC-emulator (lines 15-17 page 12) "The assumptions behind the AMOC-emulator presented here, limit it to projecting AMOC changes on multi-decadal and larger timescales. Therefore, the applied GCM-based climate forcings and AMOC strength time series should be filtered to exclude high resolution variability. ".**

*6.0.26 p. 8, l. 2:*
*"Note that the temperature forcing files need to be interpolated onto the temporal resolution used in the atmospheric component of the AMOC-emulator"—Awkward in several counts.*
*1. The interpolation draws the attention of the reader as if it were something noteworthy. Perhaps the results are sensitive to the method of interpolation? The reader would wonder.*
*2. Is the fact that the GCM data are saved in files noteworthy? (I mean, why mention the files at all?)*
*3. Despite this cautious tone, the interval at which the GCM data is saved is not indicated.*
*If the result is sensitive to the interpolation, give more details. If not, what about just saying, "The GCM variables are saved at an interval of XXX hours and interpolated on to the time steps of the AMOC-emulator", something along the lines.*
**The line has been removed.**

*6.0.27 p. 8, ll. 5–6:*
*A similar problem. If interpolation is so noteworthy, give more details. If it's not so big a deal, just say, "the GIS melt forcing is interpolated. . . ." instead of "Note that the GIS melt forcing needs to be interpolated. . . ."*
**The line has been removed.**

[Figure]

**AMOC-emulator framework M-AMOC1.0 for uncertainty assessment of future projections**

Pepijn Bakker[1,*] and Andreas Schmittner[1]

[1]College of Earth, Ocean and Atmospheric Sciences, Oregon State University, USA
[*]Now at MARUM – Center for Marine Environmental Sciences, University of Bremen, Bremen, Germany
*Correspondence to:* Pepijn Bakker (pbakker@marum.de)

**Abstract.** State-of-the-science global climate models show that global warming is likely to weaken the Atlantic Meridional Overturning Circulation (AMOC). While such models are arguably the best tools to perform AMOC projections, they do not allow a comprehensive uncertainty assessment because of limited computational resources. Here we introduce the AMOC-emulator M-AMOC1.0, a model framework designed for probabilistic projections of multi-centennial time scales. M-AMOC1.0 uses complex climate model results to force and tune a box model of the AMOC. Box model parameters are adjusted using a simulated annealing procedure. We provide a detailed description of the AMOC box model and show how complex climate model output can be used to force and tune the box model. Finally, we provide an example based on simulations of future climate change including increased greenhouse-gas levels and enhanced melting of the Greenland Ice Sheet performed with the UVic climate model of intermediate complexity. Despite its simplicity, we show that this modeling framework can capture the first order response of the AMOC in UVic to climate change and thus provide a method that can in future studies be applied to existing and new climate change simulations to provide thorough uncertainty assessments.

[Figure]

**1 Introduction**

The Atlantic Meridional Overturning Circulation (AMOC) is an important part of the climate system due to its effects on the transports of heat, salt, carbon, nutrients and other tracers. Projections consistently show a reduction of the AMOC due to future global warming (*IPCC Climate Change*, 2013), with the possibility of an irreversible transition to a shutdown state (Stommel, 1961; Stouffer et al., 2006), a prime example of a tipping point in the climate system (Lenton et al., 2008). Large ensemble simulations are necessary for probabilistic projections, policy relevant risk assessment of future emission scenarios and to assess the uncertainties of AMOC projections due to climate sensitivity, polar amplification, melting of the Greenland Ice Sheet (GIS) and model dependent sensitivity of the AMOC to such climatic changes. The high degree of complexity and spatial resolution of GCMs make them too computationally expensive to perform such an analysis and thus a model is needed that is much cheaper to run, but nonetheless captures important characteristics of the GCM's AMOC response to climate change.

To this end we developed an AMOC-emulator framework. It entails a simple box model that uses physical relationships to represent the most important mechanisms and feedbacks that govern the AMOC's response to changes in regional surface

---

## Author Comment (AC5) · 29 Aug 2016

**Response to anonymous Referee #1**

**We thanks Reviewer 1 for the interesting and extensive comments on the manuscripts. Below we will provide a detailed response to all individual comments.**

*I don't understand the reasoning for why you generate 100 reasonable fits and then select only the 10 best fits (P7 L15). Firstly, this emulator/box-model should be cheap to run, so why choose such small numbers? Surely ensembles of order 10,000 or 100,000 are more reasonable. Secondly, the choice of 10 best fits seems to narrow the ranges of several parameters (e.g. V4, F1, h1). By doing this you rule out large regions of parameter space that give perfectly reasonable fits, and could behave differently under different forcing scenarios. If the primary aim is to assess uncertainty in AMOC projections I would expect to see a rigorous analysis of the uncertainty. By discarding large areas of parameter space uncertainty will certainly be underestimated.*

**Thanks for raising this valid point. Our aim here is not to provide an uncertainty assessment of AMOC projections, it is to provide a method with which one could do this, as for example done in the manuscript by Bakker et al. under review in GRL, now also pointed out in the last line of the main manuscript**

**(lines 1-2 page 13) "The AMOC-emulator is a valuable tool to study the uncertainty in GCM-based AMOC projections, such as the one recently being performed on the results from the AMOCMIP project (Bakker et al., 2016)."**

**The assessment referred to here is based on multiple GCMs, decreasing the need for a large number of AMOC-emulators for a single GCM.**

**With regard to the point that the AMOC-emulator is cheap and could thus be run for tens or even hundreds of thousands of times. This is very true, however, it takes many time steps and iterations to find a single reasonable fit. We have now included the following description to the manuscript**

**(line 30 page 8 to line 2 page 9) "To provide an idea of the computational expenses of the model we provide a back of the envelope calculation. This shows that a single run over all scenarios takes $10^5$ time steps which are done in about 5 seconds. You need on the order of 400 iterations (in which parameter values are perturbed) to find a single reasonable fit, resulting in approximately half an hour to calculate a single reasonable fit on a normal desktop computer."**

**This shows that by using more powerful computers and or running in parallel the number of reasonable fits could be enhanced, but it shows that 10,000 to 100,000 reasonable fits is ambitious nonetheless.**

*In the four scenarios not seen by the emulator (Fig. 8) the behavior of UVic is clearly not captured in two cases (lower left and upper right). There are no confidence intervals plotted (or computed as far as I can tell), but I believe the GCM would lie well outside 2 standard errors in those two cases. Therefore, the GCM would still need to be run for any untested scenario. I would not trust the emulator in its current form.*

**We don't agree with the general notion given by the reviewer. Firstly, it is important to realize that the values given in figure 8 are anomalies with respect to the time series given in figure 7. Thus even the largest mismatch between GCM and AMOC-emulator (~1-2Sv in lower left panel) is 'only' a mismatch of 10-20%**

**(lines 27-28 page 10) "Nonetheless, the mismatch between the AMOC-emulator and the UVic-based AMOC evolution is smaller than ~10% of the AMOC strength."**

**We have added an objective assessment of the predictive power of the AMOC-emulator by comparing the results with a null-model that assumes that the emulator has no predictive power;**

it doesn't know if an additional forcing on top of the ones used in the tuning procedure would further increase or decrease the AMOC and would thus result in zero anomalies. This assessment shows that in three out of four cases the AMOC-emulator has substantial predictive power. We discuss this assessment in the manuscript

(lines 20-30 page 11) "This is quantified by comparing the AMOC-emulator results with a null-model that assumes an AMOC-emulator with zero skill, meaning that it simply reproduces the original calibration data. The results from these experiments are shown as anomalies relative to the original scenario, the original being RCP8.5-GIS for RCP8.5x0.5-GIS, RCP8.5x1.5-GIS and RCP8.5-GISx1.5, and RCP4.5-GIS for RCP4.5-GISRCP8.5x1.5. We find that for large changes in the GHG forcing the Uvic-based AMOC-emulators are well capable of predicting the AMOC evolution of UVic in terms of sign and amplitude and perform better than the null-model (upper panels Fig. 8). For large changes in the applied GIS melt forcing the picture is more complex (lower panels Fig. 8). A strong increase in GIS melt under a low GHG scenario shows an excellent performance of the AMOC-emulator and a RSME that is much lower than for the null-model (RCP4.5-GISRCP-8.5x1.5 in Fig. 8), but for the high GHG scenario, a 50% increase in GIS melt leads to a deterioration of the fit between UVic and AMOC-emulator with consequently a larger RSME than that provided by the null-model (RCP8.5-GISx1.5 in Fig. 8). The latter shows that the UVic-based AMOC-emulators tend to overestimate the impact of GIS melt on the AMOC strength under high-end GHG scenarios. Summarizing, in all four cases the emulator predicts the correct sign of the AMOC response to changes in the forcings, and in three out of four cases the predictive power of the AMOC-emulator is better than of the null-model."

Nonetheless, it is important to acknowledge that using an emulator will introduce a new type of error in any assessment, pointed out by the following text in the manuscript

(lines 5-7 page 12) "It is clear that using an AMOC-emulator introduces a new type of uncertainty into AMOC projections, however, for which level of added uncertainty an AMOC-emulator is still useful is a question that is difficult to address."

*On multidecadal timescales the emulator is plagued by sensitivity to surface temperature oscillations. These seem to have arisen from the addition of the atmospheric boxes to the ocean box model published by Zickfeld et al., 2004. Can the authors confirm that this is the case, and if so can they control this sensitivity, e.g. by introducing a damping/mixing term?*

The multidecadal AMOC oscillations result from the UVic-based regional temperature forcings of the AMOC-emulator and thus in turn to internal variability of UVic. Zickfeld et al. (2004) applied highly idealized linear temperature increases of global temperature, thus not including any multi-decadal variability. On the contrary, in our approach we directly use regional GCM-based temperature time series to force the AMOC-emulator. In this way the forcing not only takes into account the GCMs global climate sensitivity, but also mechanisms like polar amplification etc. that cause regional temperature change differences. This method also introduces any multi-decadal internal variability that might exist in a GCM into the AMOC-emulator when expressed in regional temperature time series. We acknowledge this feature, but do not see it as an issue.

*If the authors have good reason to retain this behavior they need to test the sensitivity to the phase of the variability. For all of the scenarios, the chosen start date (2006) appears to be shortly after a peak in the strong multidecadal variability, so the AMOC is preconditioned to decline at this time. Under all scenarios the AMOC in the 'best' emulators appear to decline faster than the UVic model. Consequently, the SA tuning and the cost function used may be adversely affected by this multidecadal variability.*

Indeed, following from the usage of the Stommel model to emulate the AMOC, multi-decadal

temperature variability and its phasing impact the projected AMOC changes, in the AMOC-emulator, in UVic and most likely also in reality. Perhaps the AMOC response in the AMOC-emulator to regional temperature changes is too direct (as mentioned in the manuscript)
(lines 25-26 page 10) "simulates a too direct response of the AMOC strength to multi-decadal surface temperature oscillations."
and thus the importance of multi-decadal variability overestimated, but we don't see this as a major issue. It seems to us that the years before 2006 represent in fact a time of relatively weak AMOC, not strong, thus preconditioning the AMOC-emulator to a somewhat weaker response to global change. However, again, we don't see this as an issue since we focus on projecting AMOC changes on multi-decadal and longer timescales, as is now speciffically mentioned in the manuscript
(lines 21-22 page 12) "The assumptions behind the AMOC-emulator presented here, limit it to projecting AMOC changes on multi-decadal and larger timescales."
We don't agree with the notion of the reviewer that the decline in the emulators is faster than in UVic, they are in fact very similar. Finally, multi-decadal AMOC variability only impacts the absolute value of the cost function, not the resulting optimal fits.

*On centennial timescales the emulator (as currently presented) does not capture crucial features of the AMOC response to the forcing (Fig. 7). In particular I would draw attention to the RCP4.5 scenarios, in which the GCM exhibits a strong reduction followed by a steady recovery. The emulator fails to identify either the timing or amplitude of the AMOC minimum and it fails to identify the recovery phase. In addition it appears to show signs of a recovery phase under RCP8.5 when UVic shows none. The authors state (P9 L28) that the fit can be improved, but that this would entail a higher overall cost function for the SA tuning method. Is this indicative of a poor choice of cost function? Does it mean that the box model should be tuned separately for each scenario?*
The failure of the AMOC emulator to capture the slight recovery of the AMOC under RCP4.5 is indeed an issue and shows the limitations of the simple box model to capture all complex feedbacks in the GCM. Indeed, as mentioned in the manuscript, the AMOC-emulator does allow for an AMOC recovery under RCP4.5, but that would mean a large deterioration of the fit of the AMOC emulator to the AMOC in RCP8.5 and thus it would increase the value of the cost function, for which reason this solution is not found through this approach. It is an interesting point if the AMOC-emulator should be tuned separately for each scenario. We added the following to the manuscript to cover this issue
(lines 23-33 page 10) "It is also worth noting that the fit for an individual simulation could be improved, for instance the AMOC-emulator does allow for a partial AMOC recovery as UVic shows for RCP4.5, but such an AMOC-emulator is not found through the SA tuning methodology in this example, because it would degrade the fit for the other scenarios and thus lead to an overall higher cost function."
More discussion on this topic follows in Sect. 4 of the manuscript
(lines 7-13 page 12) "Another important consideration when using the AMOC-emulator is the spread in GCM climate forcing scenarios that is included in the tuning process. When using only a single climate change scenario, a better match can be obtained between the AMOC evolution given by the GCM and AMOC-emulator, however, the reliability of the AMOC-emulator will quickly decrease for different climate forcings. On the other hand, one could use a large number of climate change projections in the tuning process to obtain a lesser fit for individual scenarios, but an AMOC-emulator that is applicable to a much larger range of climate change scenarios. The best strategy to be follow strongly depends on the research question in mind."

*A far more substantial summary is required. For example, the emulator's limitations need to be clearly*

*stated (and whether/how the authors think these can be addressed). For what purposes are the emulator suitable in its current form, and for what purposes might it be useful subject to further work? With the current analysis, I disagree with the statement that "the UVic-based AMOC-emulator captures well the overall characteristics of the multi-centennial response of the AMOC".*

**Thanks for this comment. We agree that are more substantial and clear discussion is needed to make clear what the model can and cannot do. We have added the following to the discussion section to clearly discuss the applicability of the AMOC-emulator and the limitations that should be considered:**

**(lines 1-22 page 12) "Overall, the predictive power of the AMOC-emulator is reasonable when one considers the simplicity of the AMOC box model, but for forcing scenarios that are increasingly far away from the forcings that are used in tuning the AMOC-emulator, the predictive power decreases. A large advantage of using a physics-based AMOC-emulator that is tuned with large climate forcings, over the use of for instance a statistical AMOC-emulator, is that it projects the point after which the AMOC collapses and switches to an off state, as this is an integral part of the physics of the Stommel model. It is clear that using an AMOC-emulator introduces new uncertainty into AMOC projections, however, for which level of added uncertainty an AMOC-emulator is still useful is a question that is difficult to address. Another important consideration when using the AMOC-emulator is the spread in GCM climate forcing scenarios that is included in the tuning process. When using only a single climate change scenario, a better match can be obtained between the AMOC evolution given by the GCM and AMOC-emulator, however, in this case the reliability of the AMOC-emulator will quickly decrease for different climate forcings. On the other hand, one could use a large number of climate change projections in the tuning process to obtain a lesser fit for individual scenarios, but an AMOC-emulator that is applicable to a much larger range of climate change scenarios. The best strategy to be followed strongly depends on the research question in mind. The assumptions behind the AMOC-emulator presented here, limit it to projecting AMOC changes on multi-decadal and larger timescales. Therefore, the applied GCM-based climate forcings and AMOC strength time series should best be filtered to exclude high frequency variability. Moreover, an AMOC-emulator that is tuned to specific GIS melt experiments is likely not applicable to experiments in which melt water is applied to a different geographical region or with a different seasonal cycle. This is not to say that the presented AMOC-emulator framework cannot equally be applied to other sources of melt water input. Finally, many processes that are known to impact the AMOC are not considered in the AMOC-emulator, for instance the impact of winds, gyre circulation, Southern Ocean upwelling or deep water formation outside of the North Atlantic (see Sect. 1). If such processes would prove to dominate the AMOC response to future climate change, a different AMOC box model should be considered that places emphasis on that particular process."**

*Minor comments:-*
*Page 3 Line 12: Prescribed FW fluxes: F1 and F2 are tuned parameters. I would have expected these to vary as a function of the forcing/climate. What is the justification for fixing them?*

**This part was not sufficiently clear in the manuscript and has now been updated. The total freshwater fluxes F1 and F2 are not part of the tuning procedure, but F01 and F02 (the combined wind-driven oceanic and atmospheric meridional freshwater fluxes for the reference state are). The text now reads**
**(line 18 page 4) "Freshwater fluxes $F_{01}$ , $F_{02}$ and coefficients $h_1$ and $h_2$ are included in the tuning procedure (Tab. 2)"**

*Page 5 Line 10: What you also fail to consider are nonlinearities between these parameters. Co-*

*varying the parameters in Tables 1 and 2 could yield very different behaviours.*
**The parameter fitting method we employ, simulated annealing, randomly varies the individual parameters, thus considering (although not explicitly) both linear and nonlinear relationships between parameters. Moreover, by including Figure 6 we perform a first order test to see whether relationships exist between parameters, which indeed is the case for several of them.**

*Page 5 Line 30: algorith > algorithm*
**Thank you, it has been corrected.**

*Page 6 Line 4: I find the arbitrary choice of +/- 200% rather strange. What is the justification for this?*
**We agree that this choice is arbitrary. Our approach has been to take this arbitrary value, perform the analysis and then to analyze whether or not all parameter values that resulted from the fitting procedure were well within the +-200% range (see also figure 6). From this it was decided to keep the +-200% value. This point is clarified by adding**
**(lines 14-15 page 7) "The appropriateness of this arbitrary range of initial parameter values is later verified by ensuring that all final parameter values are well within the initial range."**

*Page 6 Line 8: analogues > analogous*
**Thank you, it has been corrected.**

*Check typesetting in Tables (e.g. Table 1 column 2)*
**Thank you, typesetting is checked.**

*Table 1: (typo) dependend > dependent*
**Thank you, it has been corrected.**

*Check typesetting on Figure 8: it appears corrupted.*
**Thank you, typesetting is checked.**

*Figure 4 caption: (typo) relatvie > relative*
**Thank you, it has been corrected.**

*Figure 8 caption: (typo) calculate > calculated*
**Thank you, it has been corrected.**

*Figure 8 caption: (typo) rigth > right*
**Thank you, it has been corrected.**

**Response to anonymous Referee #2**

**We thanks Reviewer 2 for the interesting and extensive comments on the manuscripts. Below we will provide a detailed response to all individual comments.**

*1 General Comments*
*1.1 Introduction*
*The introduction does not sufficiently reflect the state of the literature on AMOC and in particularly not on conceptual models such as the Stommel-Model to study it.*
*Our understanding of AMOC dynamics has advanced considerably over the last years thanks to ongoing observations e.g. in the Rapid array (see Srokosz & Byrden, Science 2015). In addition a recent studies has suggest that the AMOC might already be in decline (Rahmstorf et al. 2015). While of course not directly relevant for the emulator itself, such observational findings need to be discussed in an approach that emulate AMOC behaviour over the next centuries. This should also include a discussion of atmospheric imprints on the AMOC e.g. such as atmospheric blocking events. It should also allow to assess the performance of GCMs in relation to the observational record.*
**We cannot agree with the points raised above. We are presenting a new modeling framework in a journal for model development. As such, we agree that some insights as to why we think a new modeling framework is needed is called for, but a discussion of observed AMOC changes, AMOC fingerprints or the performance of GCMs in relation to the observational record does not seem appropriate in this journal and is beyond the scope of this paper..**

*Much more important though is the discussion in relation to the emulator approach taken. Stommel type models have been used since quite some time and might be able to capture key dynamics of the AMOC (e.g. bistability). However, they at the same time have faced a lot of criticism and alternative models describing AMOC behaviour exist. This is in particular related to the relevance of Southern Ocean upwelling reflected in a conceptual model by Gnanadesikan (1999) related to changes in the pycnocline depth. A dynamic that is completely missing in the Stommel approach.*
*This has been explored further in conceptual models and attempts exist to unify pycnocline and freshwater-feedback dynamics. In this context, the authors should consider the work of Sijp et al. (2012) that they may find helpful.*
*Another question directly relating to the physical plausibility of the Stommel model relates to the relationship of circulation strength and meridional density gradient in a geostrophic ocean. The authors should consider work by Gregoy & Tailleux (2010) that present a kinetic energy approach essential providing a physical explanation for the (empirically supported) meridional density gradient outlining the relevance of the Western Boundary Current in modelling AMOC dynamics.*
*These comments should not be seen as undermining the Stommel model approach taken here, but they need to be addressed. In short, the authors should show motivate their approach in the light of the most recent literature.*
**Thanks for pointing this out and we agree that a more thorough discussion of the pro's and con's of the used Stommel model is called for. An important caveat of using a Stommel model is that Southern Ocean upwelling, the role of Southern Hemisphere mid-latitude winds and other processes are neglected, a point that we have added to the discussion of this manuscript. Our choice to use the Stommel model was driven by two considerations. Firstly, to our knowledge no unified simple AMOC model exists and as such it is not clear if other models are better or worse than the Stommel model in relating surface temperature and freshwater flux changes to the AMOC strength. Secondly, the Stommel model allows for rather straightforward inclusion of temperature and freshwater forcings based on GCM simulations, while for other models like the ones mentioned above it is not clear to us how this could be done. Finally, it is important to note**

that we did not set out to construct a new simple model that describes the main dynamics of the AMOC, but rather to use an existing model and build a framework around it that can easily be applied to GCM climate change and AMOC projections.

Following the above, we have updated the introduction to read
(lines 14-23 page 2) "At the center of our approach is the assumption that changes in AMOC strength are linearly related to changes in the Atlantic meridional density contrast. Since Stommel (1961) a large number of studies have provided evidence for an important role of the Atlantic meridional density contrast in driving AMOC changes (e.g. Rahmstorf, 1996; Gregory and Tailleux, 2011; Butler et al., 2016). Nonetheless, it neglects several important processes, like the role of Southern Ocean upwelling, winds and deep water formation (e.g. Gnanadesikan, 1999; de Boer et al., 2010) and a unified theory describing the fundamental mechanisms driving and sustaining the AMOC lacks to this date (Lozier, 2010). Using a Stommel model to emulate AMOC changes driven by surface temperature and freshwater forcings seems appropriate in the light of present-day knowledge and the apparent leading role of surface buoyancy changes in simulated future AMOC weakening (IPCC Climate Change, 2013). Moreover, the model is easy to use, interpreted and can be forced directly with GCM-based forcing fields. Nonetheless, the processes that have been omitted and the simplicity of the model should be considered when interpreting the results."
To the discussion section we have added
(lines 20-24 page 12) "...many processes that are known to impact the AMOC are not considered in the AMOC-emulator, for instance the impact of winds, gyre circulation, Southern Ocean upwelling or deep water formation outside of the North Atlantic (see Sect. 1). If such processes would prove to dominate the AMOC response to future climate change, a different AMOC box model should be considered that places emphasis on that particular process."

*1.2 The Emulator model*
*Here, the work dominantly builds on a previous model by Zickfeld et al. (2004) plus a representation of the Bjerknes feedback. It does however not become sufficiently clear, why this addition will represent a substantial advancement. The authors show the differences in Fig. 9 and describe that this will represent a negative feedback on the AMOC dynamics. But it's not clear, if Figure 9 shows two sets calibrated individually (with and without atmospheric feedback) or just from the optimal parameter set with this feedback switched on and off. Therefore, I cannot judge if the conclusion drawn by the authors on the importance of the effect are due to their specific parameter set or not.*
*It would add merit, if the authors could show that the model including the Bjerknes effect will in the end outperform the no-atmospheric feedback model in the fitting procedure. This would also justify, why there model is actually better than the one presented in Zickfeld (2004).*
Thanks for providing this comment. Firstly, the main improvement with respect to the Zickfeld et al. (2004) model is that we provide a framework that allows one to use limited GCM AMOC and climate projections to tune an AMOC-emulator in order to perform an uncertainty analysis. The Zickfeld et al. (2004) approach used a full ~20.000yr long hysteresis simulation to tune their emulator, not feasible for most IPCC-type GCMs. Moreover, they did not force their emulator with GCM-based temperature changes or consider inter-GCM differences in regional temperature changes. Those are the features we see as most important changes with respect to earlier work, a view that is now better reflected in the introduction of the manuscript by
(line 23 page 1 to line 3 page 2) "To this end we developed an AMOC-emulator framework. It entails a simple box model that uses physical relationships to represent the most important mechanisms and feedbacks that govern the AMOC's response to changes in regional surface temperatures, freshwater fluxes and enhanced melting of the GIS. The AMOC-emulator can be

**forced by temperature and melt water fluxes from any GCM, and using AMOC time series the free parameters of the box model are tuned to mimic the GCM's AMOC sensitivity to future climate change."**
**and in later on in the introduction it reads**
**(lines 11-13 page 2) "the approach described here is designed specifically to allow future studies in which a limited number of climate projections from multiple GCMs, limited in the simulated forcing scenarios and simulation length, to be combined into a Bayesian framework of century time-scale probabilistic AMOC projections."**

**With respect to the added stabilizing Bjerkness feedback, it indeed appears from Figure 9 that it's impact is limited. Figure 9 shows results for the same parameter sets with this feedback switched on and off, allowing for a direct investigation of its impact. Nonetheless, we deem the model including this feedback more realistic. Moreover, the effect is non-negligible**
**(lines 1-6 page 11) "The impact of including atmospheric meridional heat transport is a small, but non-negligible ~1Sv strengthening of the control state of the AMOC (not shown) and, more importantly, a slightly lower sensitivity to changes in radiative forcing and GIS melt (Fig. 9). This confirms our understanding of atmospheric meridional heat transport acting as a negative feedback to AMOC changes. The simulations with the atmospheric feedback included have on average a stronger AMOC by 8.1±1.9% (μ ± σ; calculated over all 10 best fits and over all five forcing scenarios)."**

*Furthermore, the model includes 5 atmospheric boxes. Why are 5 boxes needed and not 3 to resolve the meridional heat transport? I think that can be easily motivated and maybe I missed it. Maybe it's worth considering to restructure the approach by moving subsection 2.3 further up to discuss the setup of the atmospheric forcings.*
**Including high latitude atmospheric boxes allows us to have a closed energy budget and more realistic meridional atmospheric heat transport. This is now described at**
**(lines 14-16 page 5) "Meridional heat fluxes are assumed zero at the northern and southern boundaries of the domain, which for this reason are chosen to be the North and South pole rather than the meridional limits of the ocean component."**
**Thanks for the suggestion to rearrange this section. We have accordingly switched sections 2.2 and 2.3.**

*In this context, the authors should also reflect on the limitations of the model to reproduce transient AMOC changes that relate to the assumption of well-mixed density within the boxes. This might be in particularly relevant in relation to the Greenland freshwater input. Clearly, this represents an over-simplification and may substantially limit the capabilities of this approach to emulate transient behaviour (I'll further comment on this below).*
**Thanks for pointing this out. We fully agree that a box model can never resolve the complexities of the interaction between Greenland meltwater and the ocean. We have experimented with an additional tuning parameter to include the GCM dependent 'efficiency' of Greenland meltwater to impact the density of the North Atlantic ocean box, but decided against it since the current seven tuning parameters already allow for sufficient freedom to tune an AMOC-emulator towards the AMOC sensitivity of a specific GCM. More importantly, the discussion section of the manuscript now describes the limitations of the AMOC-emulator**
**(lines 22-23 page 12) "The assumptions behind the AMOC-emulator presented here, limit it to projecting AMOC changes on multi-decadal and larger timescales."**
**and**
**(lines 24-27 page 12) "...an AMOC-emulator that is tuned to specific GIS melt experiments is**

**likely not applicable to experiments in which melt water is applied to a different geographical region or with a different seasonal cycle. This is not to say that the presented AMOC-emulator framework cannot equally be applied to other sources of melt water input."**

*1.3 The tuning to complex model output*
*In the manuscript, the model is tuned to an EMIC model UVIC. I think that's generally no problem, but somehow contradicts the initial claims by the authors that this emulator could now be used to run larger ensembles. What is it exactly that the emulator provides that cannot be done with an EMIC?*
*In general terms, the strength of an emulator is it's capability to include projections from a range of different models. We have AMOC projections for several CMIP5 models, why is it not applied to those?*
*In addition, there are the AMOC sensitivity studies by Gregory et al. (2005) and Stouffer et al. (2006) that would provide enough runs to calibrate the model. Why isn't it applied to those runs?*
*In addition, the authors mention the AMOCMIP project. Can the emulator be applied to the AMOCMIP output?*
**Thanks for these comments. It has become clear from the comments of the different reviewers that the aim of this manuscript is not sufficiently clear and we have changed the abstract, introduction and summary sections to improve on this. In this manuscript we want to describe a modeling framework that allows one to use limited GCM output to tune and force an AMOC box model that can in turn be used to perform uncertainty analysis. It is not the aim of this manuscript to provide future AMOC projections or provide such an uncertainty analysis. See also the responses provided above.**

*I checked the project homepage and understood that the AMOCMIP will explictly resolve different Greenland basins separately. Is that correct? If so, and following recent findings that it actually matters a lot for North Atlantic dynamics where the freshwater is actually applied, will this emulator be the best tool to reproduce these dynamics? Or should it maybe consist of a subpolar (Labrador Sea) and North Atlantic box? And/or should conceptual models of convection in marginal seas e.g. by Spall (2004) and Straneo (2009) be integrated?*
**Thanks for this question. Indeed the aim of the simulations in AMOCMIP is to provide 'realistic' Greenland melt scenarios and to apply those to IPCC-type climate change projections. This includes explicitly resolving spatial and seasonal differences in the meltwater flux. Such details cannot be captured by the AMOC-emulator. However, as described above, by tuning the AMOC-emulator to the forcings and AMOC projections of a specific GCM, we take into account the inter-GCM differences in the sensitivity of the AMOC to changes in temperature and freshwater.**

*1.4 Results*
*I've to admit I'm not impressed by the capabilities of the emulator in reproducing the model outcome. As apparent from Fig. 7, the emulator is systematically underestimating AMOC reduction for RCP4.5 and RCP8.5 no melt, while then over-estimating it for RCP8.5 plus GIS (maybe due to non-linearities kicking in here and timescale issues discussed above?).*
**Thanks for pointing this out. We agree that there are limitations to the AMOC-emulator and that because of choices that have been made, it appears that previous emulator perform better. There are, however, a number of important things to take into consideration. Firstly, one could perform the tuning on a single GCM forcing scenario and the result will be a closer fit between GCM and emulator AMOC. However, when choosing that approach, one is limited to applying the emulator to forcing scenarios close to the one used for tuning. By using a larger number of scenario in the tuning process, the emulator can be used to test the AMOC for a much larger range of scenarios, albeit at the cost of having larger discrepancies between GCM and emulator. We have added text**

along these lines to the manuscript

(lines 23-33 page 10) "It is also worth noting that the fit for an individual simulation could be improved, for instance the AMOC-emulator does allow for a partial AMOC recovery as UVic shows for RCP4.5, but such an AMOC-emulator is not found through the SA tuning methodology in this example, because it would degrade the fit for the other scenarios and thus lead to an overall higher cost function."

More discussion on this topic follows in Sect. 4 of the manuscript

(lines 7-13  page 12) "Another important consideration when using the AMOC-emulator is the spread in GCM climate forcing scenarios that is included in the tuning process. When using only a single climate change scenario, a much better match can be obtained between the AMOC evolution given by the GCM and AMOC-emulator, however, the reliability of the AMOC-emulator will quickly decrease for different climate forcings. On the other hand, one could use a large number of climate change projections in the tuning process to obtain a lesser fit for individual scenarios, but an AMOC-emulator that is applicable to a much larger range of climate change scenarios. The best strategy to be follow strongly depends on the research question in mind."

*The authors discussion of this simply stating that "It is, however, to be expected that a box-model does not completely capture the behavior of the AMOC as simulated with a higher order climate model" is clearly insufficient. In particular, as there have been much simpler AMOC emulators around that actually perform much better (also and in particularly an AMOC recovery, e.g. Schleussner et al. 2014).*

Another issue to consider is the use of physics-based or statistical emulators. With a statistical AMOC emulator one could obtain better agreement between GCM and emulator, however, such a model cannot be used to extrapolate for larger forcings. With a physics-based AMOC-emulator one can have more confidence in the response to large forcings, for instance a complete AMOC shutdown, notwithstanding that also in this approach the uncertainty is likely to increase for forcings further away from those used for tuning. This is discussed in the final section of the updated manuscript (lines 1-8 page 12)

"Overall, the predictive power of the AMOC-emulator is reasonable when one considers the simplicity of the AMOC box model, but forcing scenarios that are increasingly far away from the forcings that are used in tuning the AMOC-emulator, the predictive power decreases. A large advantage of using a physics-based AMOC-emulator that is tuned with larger large climate forcings, over the use of for instance a statistical AMOC-emulator, is that it projects the point after which the AMOC collapses and switches to an off state, as this is an integral part of the physics of the Stommel model. It is clear that using an AMOC-emulator introduces a new type of uncertainty into AMOC projections, however, for which level of added uncertainty an AMOC-emulator is still useful is a question that is difficult to address."

*The apparent oscillations in the emulator arising from a "too direct response" of the emulator towards multi-decadal surface temperature oscillations also merits more discussion.*
The origin of the oscillations is already mentioned in the manuscript

(lines 17-18 page 9) "The UVic-based surface temperature evolution exhibits multi-decadal to centennial oscillations that result from global climate variability originating from the Southern Ocean"

and we do not deem it necessary to discuss the resulting AMOC osculations in much detail as they are a feature of the forcing based on this particular climate model and not a feature of the AMOC-emulator. In the discussion section we have added some words describing the kind of temperature forcings that are appropriate to use

(lines 15-17 page 12) "The assumptions behind the AMOC-emulator presented here, limit it to

projecting AMOC changes on multi-decadal and larger timescales. Therefore, the applied GCM-based climate forcings and AMOC strength time series should be filtered to exclude high resolution variability."

*It is even worse for the predictions in Fig. 8. First of all, the figure is not well-labelled (no y-axis labeling, panels not clearly distinguishable, and what is given by the numbers 5,1,5?) and that there is no such thing as a top-middle panel for only two boxes.*
**The conversion of the figure must have gone wrong at some point because the points raised by the reviewer are difficult to understand looking at the figures we have in the manuscript. We will ensure that the figures are correct in the next version.**

*For none of the panels, the model actually captures key features. It fails to capture the bumps in the top-left and bottom right, and for the two other panels, it gets it wrong completely. I cannot agree to the author's conclusions that "Overall, the predictive power of the AMOC-emulator is good for reasonable forcing scenarios when one considers the simplicity of the model."*
**We don't agree with the general notion given by the reviewer. Firstly, the AMOC-emulator is not designed to emulator decadal AMOC fluctuations as simulated by the GCM. As mentioned in the manuscript, those results from internal climate variability mostly originating from the Southern Ocean and it is not to be expected that the emulator captures those. Moreover, the focus of the AMOC emulator is on multi-decadal to multi-centennial scales, something that is now specifically mentioned in the discussion (see reply above).**
**Furthermore, it is important to realize that the values given in figure 8 are anomalies with respect to the time series given in figure 7. Thus even the largest mismatch between GCM and AMOC-emulator (~1-2Sv in lower left panel) is 'only' a mismatch of 10-20%**
**(lines 27-28 page 10) "Nonetheless, the mismatch between the AMOC-emulator and the UVic-based AMOC evolution is smaller than ~10% of the AMOC strength."**
**We have added an objective assessment of the predictive power of the AMOC-emulator by comparing the results with a null-model that assumes that the emulator has no predictive power; it doesn't know if an additional forcing on top of the ones used in the tuning procedure would further increase or decrease the AMOC and would thus result in zero anomalies. This assessment shows that in three out of four cases the AMOC-emulator has substantial predictive power. We discuss this assessment in the manuscript**
**(lines 20-30 page 11) "This is quantified by comparing the AMOC-emulator results with a null-model that assumes an AMOC-emulator with zero skill, meaning that it simply reproduces the original calibration data. The results from these experiments are shown as anomalies relative to the original scenario, the original being RCP8.5-GIS for RCP8.5x0.5-GIS, RCP8.5x1.5-GIS and RCP8.5-GISx1.5, and RCP4.5-GIS for RCP4.5-GISRCP8.5x1.5. We find that for large changes in the GHG forcing the Uvic-based AMOC-emulators are well capable of predicting the AMOC evolution of UVic in terms of sign and amplitude and perform better than the null-model (upper panels Fig. 8). For large changes in the applied GIS melt forcing the picture is more complex (lower panels Fig. 8). A strong increase in GIS melt under a low GHG scenario shows an excellent performance of the AMOC-emulator and a RSME that is much lower than for the null-model (RCP4.5-GISRCP-8.5x1.5 in Fig. 8), but for the high GHG scenario, a 50% increase in GIS melt leads to a deterioration of the fit between UVic and AMOC-emulator with consequently a larger RSME than that provided by the null-model (RCP8.5-GISx1.5 in Fig. 8). The latter shows that the UVic-based AMOC-emulators tend to overestimate the impact of GIS melt on the AMOC strength under high-end GHG scenarios. Summarizing, in all four cases the emulator predicts the correct sign of the AMOC response to changes in the forcings, and in three out of four cases the predictive power of the AMOC-emulator is better than of the null-model."**

**Nonetheless, it is important to acknowledge that using an emulator will introduce a new type of error in any assessment, pointed out by the following text in the manuscript**

**(lines 5-7 page 12) "It is clear that using an AMOC-emulator introduces a new type of uncertainty into AMOC projections, however, for which level of added uncertainty an AMOC-emulator is still useful is a question that is difficult to address."**

*1.5 Summary*

*Generally, I miss a section that reflects on the limitations and short-comings of the approach taken, given in particular the apparent limitations in reproducing the EMIC results. Furthermore, an outlook of where this can be applied and what it specific strengths are compared to other approaches should be included.*

**Thanks for this comment. We agree that are more substantial and clear discussion is needed to make clear what the model can and cannot do. We have added the following to the discussion section**

**(lines 1-22 page 12) "Overall, the predictive power of the AMOC-emulator is reasonable when one considers the simplicity of the AMOC box model, but for forcing scenarios that are increasingly far away from the forcings that are used in tuning the AMOC-emulator, the predictive power decreases. A large advantage of using a physics-based AMOC-emulator that is tuned with large climate forcings, over the use of for instance a statistical AMOC-emulator, is that it projects the point after which the AMOC collapses and switches to an off state, as this is an integral part of the physics of the Stommel model. It is clear that using an AMOC-emulator introduces new uncertainty into AMOC projections, however, for which level of added uncertainty an AMOC-emulator is still useful is a question that is difficult to address. Another important consideration when using the AMOC-emulator is the spread in GCM climate forcing scenarios that is included in the tuning process. When using only a single climate change scenario, a better match can be obtained between the AMOC evolution given by the GCM and AMOC-emulator, however, in this case the reliability of the AMOC-emulator will quickly decrease for different climate forcings. On the other hand, one could use a large number of climate change projections in the tuning process to obtain a lesser fit for individual scenarios, but an AMOC-emulator that is applicable to a much larger range of climate change scenarios. The best strategy to be followed strongly depends on the research question in mind. The assumptions behind the AMOC-emulator presented here, limit it to projecting AMOC changes on multi-decadal and larger timescales. Therefore, the applied GCM-based climate forcings and AMOC strength time series should best be filtered to exclude high resolution variability. Moreover, an AMOC-emulator that is tuned to specific GIS melt experiments is likely not applicable to experiments in which melt water is applied to a different geographical region or with a different seasonal cycle. This is not to say that the presented AMOC-emulator framework cannot equally be applied to other sources of melt water input. Finally, many processes that are known to impact the AMOC are not considered in the AMOC-emulator, for instance the impact of winds, gyre circulation, Southern Ocean upwelling or deep water formation outside of the North Atlantic (see Sect. 1). If such processes would prove to dominate the AMOC response to future climate change, a different AMOC box model should be considered that places emphasis on that particular process."**

**Response to anonymous Referee #3**

**We thanks Reviewer 3 for the interesting and extensive comments on the manuscripts. Below we will provide a detailed response to all individual comments.**

*2 The model formulation*
*2.1 On using a conceptual model*
*I don't understand why this kind of ad-hoc box model is preferred. One would resort to Stommel's box model if one knows almost nothing about AMOC. Stommel's model was just conceptual, whose sole purpose is to get very rough ideas on how AMOC might work, and is not designed for the kind of quantitative modeling that the present authors pursue.*

*Stommel's box model was presented in 1961, and we know a lot more today. I would think that a simple dynamical model like Gnanadesikan's (1999, Science vol. 283, pp. 2077– ) is much better because even uncertain parameters are based on (that is, constrained by) clearly-identified dynamics. In contrast, some of the present authors' "parameterizations" aren't adequately defendable; they are based on hand-waving arguments. For example, how can one defend the parameterization that the AMOC strength is proportional to the interspheric surface-density difference? We know a lot better than that.*

*Perhaps even better, the models of Schloesser et al. (2014, Prog. Oceanogr. Vol. 120, pp. 154– ) and McCreary et al. (2016, Prog. Oceanogr. vol. 123, pp. 46– ) provide constraints among integral quantities, such as AMOC strength, thermocline depth, and meridional density difference, and hence can be utilized as a "box model". Those constrains are derived as solutions to dynamical equations rather than assumed on the basis of hand-waving arguments.*

*In short, I don't see any advantage today in utilizing an old conceptual model for quantitative prediction.*

*2.2 AMOC proportional to interhemispheric density difference?*
*The authors says that "the assumption that the meridional Atlantic density contrast between the North Atlantic and the South Atlantic is the first order driver of the AMOC" is debatable, but I think that's off the mark. The current wisdom is that the Southern-Ocean winds (and perhaps vertical diffusivity) are the first-order driver. They cite Butler et al. (2016) as the other side of the "debate" but Butler et al. do not argue that the surface meridional density gradient "drives" the AMOC. The just use density integrated twice in the vertical as a "diagnostic" of the AMOC.*
*It is clear from ocean GCM studies that the meridional density gradient is not the first order driver of AMOC. When the sea-surface density is restored toward a prescribed profile in an ocean-only GCM and windstress is changed in the Southern Ocean, the AMOC strength changes roughly linearly to the windstress. See Toggweiler et al. (1995, Dee-Sea Research vol. 42, pp.477– ) and the series of studies that follow. This is evidence enough that the interhemispheric density difference does not drive AMOC. Of course, this evidence is based on ocean-only models, and it is possible that the interhemispheric density difference is correlated with the AMOC strength through atmospheric feedbacks, but to use a one-to-one correspondence like (1) needs justification based on atmosphere-ocean coupled dynamics.*
*By the way, I found that Butler et al. (2016) still use the traditional hand-waving parameterization px=Lx / py=Ly. See Schloesser et al. (2012, Prog. Oceanogr. Vol. 101, pp. 33– ) for a better parameterization based a lot more on dynamics.*
**Thanks for describing in detail your view on conceptual models and the drivers of the AMOC. Indeed there is a wide variety of models of different complexity that describe (some aspect of) the AMOC, some more based on dynamic considerations than others, some including parameters**

that are more easily constrained by observations than others. In this study we have chosen to use one of the simplest and most established of such models around which to build our emulator framework, this is done for various reasons: i) it can easily be forced by temperature and Greenland melt outputs from a GCM; ii) it is easy to implement; iii) very fast to run; iv) easy to understand/diagnose the results; v) has several free parameters that can be tuned towards the behavior of a GCM in terms of the sensitivity of the AMOC to changes in heat and freshwater fluxes. We acknowledge that a different model could have been chosen, perhaps one that would turn out to perform better, however, we are not able try every single one of them so that remains unknown.

We do agree that when using the Stommel model to make the connection between changes in temperature and freshwater and changes in the AMOC strength implies that certain processes are not taken into account, like the role of changes in Southern Ocean winds, upwelling and deep water formation when projecting future changes in the AMOC. This is now more clearly described in the manuscript with the following in the introduction (lines 14-23 page 2) "At the center of our approach is the assumption that changes in AMOC strength are linearly related to changes in the Atlantic meridional density contrast. Since Stommel (1961) a large number of studies have provided evidence for an important role of the Atlantic meridional density contrast in driving AMOC changes (e.g. Rahmstorf, 1996; Gregory and Tailleux, 2011; Butler et al., 2016). Nonetheless, this model neglects several important processes that have also been shown to be of importance, like the role of Southern Ocean upwelling, winds and deep water formation (e.g. Gnanadesikan, 1999; de Boer et al., 2010) and a unified theory describing the fundamental mechanisms driving and sustaining the AMOC lacks to this date (Lozier, 2010). Using a Stommel model to emulate AMOC changes driven by surface temperature and freshwater forcings seems appropriate in the light of present-day knowledge and the apparent leading role of surface buoyancy changes in simulated future AMOC weakening (IPCC Climate Change, 2013). Moreover, the model is easy to use, interpreted and can be forced directly with GCM-based forcing fields. Nonetheless, the processes that have been omitted and the simplicity of the model should be considered when interpreting the results." and the following in the discussion section (lines 20-24 page 12) "...many processes that are known to impact the AMOC are not considered in the AMOC-emulator, for instance the impact of winds, gyre circulation, Southern Ocean upwelling or deep water formation outside of the North Atlantic (see Sect. 1). If such processes would prove to dominate the AMOC response to future climate change, a different AMOC box model should be considered that places emphasis on that particular process."

The newly added text cited above also points towards the debate that is still ongoing about the actual drivers of the AMOC (see also cited works). However, we do not agree with the reviewers view that it is now well established how the AMOC works, what drives future AMOC changes and that the Stommel model has been shown to be wrong. Although we are not experts in the field of conceptual AMOC models and we do not plan to provide a discussion of all literature on the subject, we are not aware of any such consensus in the field. As becomes clear from the references cited above by the reviewer and by the references in the manuscript, many things are still debated and the Stommel model, that is the relationship between AMOC strength and meridional density differences, is still widely used to discuss the mechanisms and stability of the AMOC in complex GCMs.

*3 Tuning and validation*
*3.1 GCMs for tuning*
*Why aren't multiple coupled GCMs used to tune the parameters? Do the authors recommend that the AMOC-emulator be tuned differently for each model?*

**The present manuscript aims to describe a modelling framework that can be used in combination with any GCM. Since the AMOC sensitivity to changes in heat and freshwater fluxes is strongly GCM-dependent, we indeed recommend that the AMOC-emulator is tuned separately for every GCM to reflect these differences. Text clarifying this can be found at**
**(lines 26-28 page 2) "GCM-based changes in regional temperatures and GIS melt changes andhow the parameters in the AMOCemulator can be tuned towards AMOC projections of an individual higher order climate model using Simulated Annealing (SA)."**
**And**
**(lines 1-6 page 13) "We have presented an AMOC-emulator that can be used to assess the uncertainties in AMOC projections produced by GCMs under future changes in surface temperatures, freshwater fluxes and melting of the GIS. The AMOC-emulator only requires a limited number of GCM projections of future climate change from any single GCM for tuning and as forcing, and thus a large number of GCM-specific AMOC-emulators can easily be found making it suited to be applied to model inter-comparison initiatives."**

*The emulator is based on equations that represent physical processes in the real world. Then, if at all possible, the parameters should be tuned on the basis of reality. Granted that there is not enough data for the deep ocean. Then the second best thing is the publicly-available collections of coupled GCM runs. I think that studies have indicated that a multi-model ensemble is usually better than a single model to mimic reality. So, the tuned parameters would be more likely better if they are based on multiple models.*
**We agree that ideally one would tune the free parameters within bounds provided by observational data. However, since it is such a highly simplified and conceptual model, the parameters are not easily obtained from observations even if that data would be abundant. For future applications, we indeed recommend that a large number of AMOC-emulators is used, tuned towards different GCMs, in order to provide a range of parameter values that is hopefully as close to reality as possible.**

*3.2 Variables for tuning*
*The variables (salinity, ocean temperature, etc.) of the emulator should be compared with those from the GCM. It is possible that the state of the emulator is very different from that of the GCM even when the AMOC strength m agrees.*
*On a more basic note, have the authors made sure that all the variables of the emulator take reasonable values? I don't think it would be okay if, say, salinity takes a value of -100 psu even if the value of m is reasonable!*

*If the atmosphere and ocean states aren't realistic, how can we trust the emulator?*
*One approach to cope with this problem would be to include other variables than AMOC strength in the cost function. Another approach would be to compare various variables between the runs of the tuned model and those of the GCMs (part of validation). I think both are necessary.*
**Thanks for providing this interesting idea. First of all, we fully agree that one has to make sure that the values for salinity and temperature in the different parts of the box model are realistic. This has been checked. Indeed one could include the comparison between GCM and AMOC-emulator temperatures and salinity in the tuning procedure. However, there does not seem to be a reason why an AMOC emulator that has temperatures and salinities that are closer to the GCMs, would perform better in terms of the AMOC behavior, in fact there is the risk that the tuning of the parameters would be steered by temperatures and salinities instead of the AMOC, thus likely deteriorating the capabilities of the AMOC-emulator in terms of mimicking the AMOC in the**

GCM. Another problem with putting to much emphasis on the comparison of GCM and AMOC-emulator temperatures and salinities, lies in the fact that in a GCM there are many other processes, apart from the AMOC, through which changes in atmospheric temperatures and GIS melt impact ocean temperature and salinity. Since our focus is purely on providing a computational efficient method to provide uncertainty estimates of GCM AMOC projections, we deem the current approach most suited.

*3.3 Models for validation*
*Moreover, I think the tuned emulator should be validated against another set of different models. Otherwise the validation isn't robust.*
As discussed above, the sensitivity of the AMOC differs from one GCM to the next, therefore the AMOC-emulator should be tuned separately for every GCM and thus not be validated with results from a different GCM. The use of this AMOC-emulator would be to allow one to perform a large number of sensitivity experiments for a single GCM, thus making it possible to assess the impact of forcing uncertainty in an individual GCM AMOC projections. This is made clear now in the final section of the manuscripts
(lines 1-6 page 13) "We have presented an AMOC-emulator that can be used to assess the uncertainties in AMOC projections produced by GCMs under future changes in surface temperatures, freshwater fluxes and melting of the GIS. The AMOC-emulator only requires a limited number of GCM projections of future climate change from any single GCM for tuning and as forcing, and thus a large number of GCM-specific AMOC-emulators can easily be found making it suited to be applied to model inter-comparison initiatives."

*I also wonder if an ensemble of runs are necessary for the GCM (UVic) for tuning and validation. For example, there is only one run for each case in Figure 8, but doesn't the AMOC strength differ from realization to realization? I don't know how chaotic the GCM is (because it uses a low-degree-of-freedom atmospheric model), but isn't the reality more or less chaotic?*
Thanks for this question. This is indeed an interesting point. There are two points to this question. Firstly, like mentioned by the reviewer, Uvic is a low resolution GCM that is know to have less variability than higher resolution and complexity models. However, for high and low resolution models, the forced response of the AMOC to strong changes in temperature and freshwater is much stronger than internal variability, which is on the order of 1Sv on decadal and longer timescales. There are indications from observations that in the real world AMOC variability is substantially larger, however, those time series are currently too short to make any robust statements about this and, furthermore, this does not impact our GCM-based methodology.

*4 Forcing*
*I may be missing something, but it's not clear to me what forces the emulator. The solar flux S seems constant in time (Table 1), but then how is the increase in green-house gas represented?*
Thanks for this comment. The section on AMOC-emulator forcings describes the way we use GCM output to force the AMOC-emulator. Indeed the solar flux is constant in time. The GCM regional temperature changes, including the impact of increased green-house gasses and all feedbacks, are included in the AMOC-emulator through the so-called 'total atmosphere effect' parameter. A parameter of the atmosphere, "a temporal and spatial varying parameter that effectively combines atmospheric emissivity, the greenhouse effect and all other processes included in a GCM that cause regional temperatures to differ from global temperature changes". This way it is ensured that the ocean is regionally forced by almost the same temperature changes as in the GCM. Furthermore, the AMOC-emulator is forced by the GCM-based $F_{GIS}$ forcing.

*If I understand it correctly, forces the model (equation 19) toward one particular GCM solution, but wouldn't it damp the emulator's variability? especially when the emulator is to simulate a state that is very different from the GCM state used for ? Doesn't this amount to building the solution into the simulation?*

**The GCM-based regional temperatures that are used as forcing, are used in the tuning phase. If one would like to use the AMOC-emulator to simulate a temperature forcing that is different (as is done in the section "Predictive power of the UVic-based AMOC-emulator"), the changes in the 'total atmosphere effect' parameter can be changed accordingly.**

*5 Minor points: math notations*
*5.1 Arrays The authors define boldface math symbols to mean "arrays", but I recommend avoiding this unconventional convention. For example, a multiplication of two "arrays" can mean several different things in conventional mathematics. Equation (13) includes the multiplication of the arrays S and p, which is meant to represent (S1p1; S2p2; : : :), which is hardly conventional. K and Ta have the same problem in (14). Also, equation (11) includes 1=z, by which the authors mean (1=z1; 1=z2; : : :), but which is not widely used in math.*
*5.2 Subscripts*
*I recommend using an upright font for multi-character math symbols such as "start" and "gcm"; or avoiding them. In particular, the subscript "it" looks as if it represented two subscripts i and t. I recommend using a single-character subscript, such as " i1" or if you insist on multi-character subscript, you may want " it□1" using an upright font.*

**Thanks for pointing out the issues with math notations and subscripts. Indeed the notations that are used are confusing and in some places wrong. We have updated the manuscript following the recommendations of the reviewer.**

*6 Point by point comments*
*Some of the following comments support my arguments above, some raise other concerns, and others point out minor, mostly editorial, problems. I wrote many of them as I read the manuscript for the first time, and as a result, they include some redundancy. I leave them as they are, because they often reflect difficulties or problems the reader may encounter as she reads the text.*

*6.0.1 p. 1, l. 19:*
*"due to climate sensitivity, polar amplification, GIS melt and model dependent sensitivity of the AMOC . . . "—I'm confused. Doesn't "climate sensitivity" include all the remaining items in the list? Why is it listed in parallel with the rest?*

**Thanks for pointing this out. Indeed taking climate sensitivity as the the global temperature change for a doubling of $CO_2$, this term is mostly taken to include all other processes that are listed. However, in the model world this is not always the case. For instance, ice-sheet-climate interactions are mostly not considered and thus GIS melt not taken into account. Moreover, one can have the same climate sensitivity, but different polar amplification and the latter can result in a different AMOC response because of the sensitivity of the AMOC to latitudinal temperature differences. Why we prefer to list all of them separately in this context, is because GCMs differ in all those terms, and all those uncertainties can be tested individually with the AMOC-emulator.**

*6.0.2 p. 2, l. 5:*
*"(Rahmstorf and Willebrand, 1981)"—As the reference list indicates, this should probably be Rahmstorf and Willebrand (1995).*
**Thanks for pointing this out, it has been corrected.**

*6.0.3 p. 2, ll. 5 & 31:*
*"the so-called Bjerknes feedback"—Probably this is because I'm not much versed in climate research, but isn't the "so-called Bjerknes feedback" restricted along the equator? A direct overturning circulation occurs connecting cooling in the eastern Pacific, say, and warming in the western Pacific only along the equator, where the Coriolis force vanishes, and the surface windstress associated with this zonal overturning circulation enhances the upwelling of sea water, which further lowers the sea-surface temperature in the eastern Pacific—a positive feedback, which is "the so-called Bjerknes feedback".*
*The authors cite Rahmstorf and Willebrand (1995) for "the so-called Bjerknes feedback", but Rahmstorf and Willebrand proposed a negative feedback due to heat transport within the atmosphere, I think.*
**Indeed this topic is somewhat confusing as indeed the Bjerknes feedback often refers to the feedback described by the reviewer, but this term (or Bjerknes compensation) is also used to describe the compensation between meridional heat transport by the ocean and atmosphere as first proposed by Bjerknes in 1964. The latter indeed provides a stabilizing or negative feedback to AMOC changes.**

*6.0.4 p. 2, l. 7:*
*"tuning a numbrer of free parameters"—They aren't "free". They represent specific physical processes and hence must be ultimately determined by physics, even though it's in practice difficult to derive their values purely from physical principles.*
**The term 'free parameter' is used here to make the distinction between parameters that are prescribed and those that are not, or in other words, those that are part of the tuning process and those that are not. All of them represent physical processes and should (and often are) determined from observations.**

*6.0.5 p. 3, l. 12:*
*Why is F prescribed? I would expect it to change according to the state of the climate system. What do IPCC-class coupled GCMs say about the change in F under global warming, for example? . . . but, later in the text, the authors say that F is related to the global atmospheric temperature (equation 12). So, it's not prescribed after all.*
**Thanks for pointing this out. The manuscript is not sufficiently clear on this topic and changes have been included for clarification. $F_i$ consists of two parts (equation 12), a part that is fixed in time ($F_{0i}$) and a part that is a function of global temperature changes ($h_i \Delta T_{glob}$). Both $F_i$ and $h_i$ c are part of the tuning process.**

*6.0.6 Equations (3)–(10): What does this "" mean? Is it a typo for "@"?*
**Thanks for pointing this out, we have updated the manuscript for clarity.**

*6.0.7 Equation (11):*
*State whether z's are fixed, and if so, give their values here or refer the reader to a table or something.*
**The formulation now includes a clear notation showing that $z$ is a function of $i$ and and a reference to Table 1 is given.**

*6.0.8 Equation (12):*
*Tglob should be defined. (How is it computed from Ta?)*
**This has been rewritten to read "global atmospheric surface temperature anomalies".**

*6.0.9 Equation (12):*
*Give F0's their values here or refer the reader to a table or something.*
**This line now reads "Freshwater fluxes F01 , F02 and coefficients h1 and h2 are included in the tuning procedure (Tab. 2)."**

*6.0.10 Equation (13):*
*I may be mistaken, but it seems that the Ta4 is the only nonlinear term. Doesn't it make sense if this term is linearized around a mean state?*
**Yes it could be, but we prefer to keep the current form.**

*6.0.11 Equation (13):*
*The solar flux S is a confusing notation. By the authors' own convention, S =(S0; S1; S2; S3; S4), which uses the same symbols as salinity.*
**Thanks for pointing this out, this is indeed confusing. The notation has been changed to read $I_i$.**

*6.0.12 Equation (13):*
*The solar flux S should be discussed right below equation (13). Does it depend on time? Table 1 suggests that it's constant in time but that should be stated explicitly. So, does the emulator solves only for annual averages?*
**The first line after equation 13 now reads "where σ is the Stefan Boltzmann constant and $I_i$ and $\alpha_i$ the latitude dependent yearly mean incoming shortwave radiation and planetary albedo, respectively (see Tab. 1 for details)".**

*6.0.13 Equation (13):*
*Define this symbol precisely. (But I don't recommend this notation because a gradient of an array is a strange mathematical entity.)*
**Thanks for pointing this out. It should have been a Delta symbol.**

*6.0.14 Equation (14):*
*What does this "" mean? Is it a typo for "@"?*
**This has been corrected.**

*6.0.15 Equation (14):*
*State that Ha is defined to vanish at the northern and southern ends of the northernmost and southernmost boxes. (I guess they are so defined, right?)*
**The following has been added "Meridional heat fluxes are assumed zero at the northern and southern boundaries of the domain."**

*6.0.16 p. 5, l. 15:*
*I guess we need some discussion on other possible sets of tuning parameters. We have a vast range of possibilities. Then, how have we settled on these seven parameters? Have the authors tried other combinations of parameters?*
**We deem this discussed by the line (lines 25-27 page 6) "This selection of parameters is somewhat subjective, but it proved a good balance between, on the one hand, sufficient degrees of freedom to tune the AMOC emulator's behavior towards that of a specific GCM and, on the other hand, the efficiency to find optimal parameter fits."**

*6.0.17 p. 5, l. 15:*
*F and h are related by equation (12), and so cannot be determined independently. Moreover, if you tune*

*F, you can forget about equation (12) and don't need to consider h.*

**We apologize for the confusion that has arisen because of errors in the notation. This has been corrected in the text and figures. $F_{0i}$ and $h_i$ are the parameters used in the tuning process. The former giving the steady state meridional freshwater transport by both the atmosphere and the wind driven ocean part, while the latter controls the changes in atmospheric transport as a function of global temperature changes.**

*6.0.18 Equation (15):*
*Why try to optimize m alone? It's conceivable that widely different states have similar m values. Because we have other variables like salinity, we could choose better sets of parameter values, if we include other variables in the cost function, couldn't we?*
**As discussed above, we don't think that including more variables in the tuning process would lead to a better behavior of the AMOC-emulator in terms of its capacity to mimic the GCMs AMOC sensitivity to changes in temperature and freshwater.**

*6.0.19 Equation (15):*
*I may well be mistaken, but it seems that the differential equations are linear in the tuning parameters and if so, the optimization problem on the new cost function ...is a quadratic function of the parameters and can be solved analytically, I think.*
**The optimization of the parameters cannot be solved analytically as the system is non-linear and includes 7 parameters that influence each other.**

*6.0.20 Equation (16):*
*The notation "pstart(1□z)" is confusing because it looks as if pstart(z) were a function of z. Vectors customarily come after scalars, as in "(1 □ z)pstart"*
**It has been changed.**

*6.0.21 Step 1:*
*I don't understand why we have to repeat this step. Why not choose values that are within the ranges in Table 2 in the first place? We can use a random variable whose PDF is uniform over the specified range for each parameter, can't we? I mean, if (1 □ z)p1 is below the range, we can just use p1min for the lower bound; that is, we can use U(max((1 □ z)p1; p1min); min((1 + z)p1; p1max)) without repetition. The same argument holds for the last part of Step 3.*
**The solution given by the referee will not always give the same results. More specifically, the p1min and p1max values would be used much more often than other values in case random values from outside of the range are often picked. Another solution would be to split $z$ into $z_{min}$ and $z_{max}$ and adjust those values for every parameter to ensure that the randomly picked values are never outside of the imposed ranges. We don't think it matters which solution is picked.**

*6.0.22 Equation (17):*
*I may be missing something, but shouldn't (16) and (17) be written in parallel forms? If we write (1 □ z)p for (16), then we should write (1 □ it)p for (17). If we write p □ itp for (17), we should write p □ zp for (16). For a moment, I was confused with (17).*
**Thanks for pointing this out, using the same notation for both equations indeed improves readability.**

*6.0.23 p. 7, l. 16:*
*I think that efforts should be made to narrow the range of the parameter values. If parameters are widely different even though the cost function is similar, doesn't that suggest that the parameters aren't*

*well tuned?*

*What about comparing variables other than m between the emulator and the GCM? Wouldn't that tell which parameter values are bad?*

*It seems that the authors have forgotten that there is only one reality.*

**Firstly, we assume the reviewer is pointing towards 'GCM reality' in this comment, since we do not aim to work towards a single parameter set that provides the closest resemblance to the real world AMOC, no matter how much we would like to do so. However, also when we are talking about using an emulator to mimic the complex AMOC behavior in a GCM, we do not expect that there is a single parameter set that provides the perfect match between GCM and emulator; because of the highly simplified nature of the emulator, it cannot be determined which parameter set is closest to 'GCM reality'.**

*6.0.24 p. 7, l. 22:*

*What is "RCP"? (I may have missed its definition given in the text.) Because how is determined is important, it may be helpful to give a bit more information here.*

**This line reads "RCP4.5 and RCP8.5 (Representative Concentration Pathways; Meinshausen et al., 2011)", which we deem sufficient information, especially since in the context of testing the AMOC-emulator, the exact imposed climate forcings are only of secondary importance.**

*6.0.25 Equation (19):*

*Does the model really use the full time-series of Tgcm? Or is that a long-term mean? State clearly how is Tgcm defined. If the emulator uses the full time-series, it may not be appropriate for other models or for other scenarios.*

**Thanks for pointing this out. $T_{GCM}$ can in principle be of any temporal resolution. However, the model aims at resolving AMOC changes on decadal and longer timescales and as such, including high frequency variability in the temperature forcing could lead to misinterpretation of the results. We have added a discussion point in the final section of the manuscript to clarify the strengths and weaknesses of the AMOC-emulator (lines 15-17 page 12) "The assumptions behind the AMOC-emulator presented here, limit it to projecting AMOC changes on multi-decadal and larger timescales. Therefore, the applied GCM-based climate forcings and AMOC strength time series should be filtered to exclude high resolution variability. ".**

*6.0.26 p. 8, l. 2:*

*"Note that the temperature forcing files need to be interpolated onto the temporal resolution used in the atmospheric component of the AMOC-emulator"—Awkward in several counts.*

*1. The interpolation draws the attention of the reader as if it were something noteworthy. Perhaps the results are sensitive to the method of interpolation? The reader would wonder.*

*2. Is the fact that the GCM data are saved in files noteworthy? (I mean, why mention the files at all?)*

*3. Despite this cautious tone, the interval at which the GCM data is saved is not indicated.*

*If the result is sensitive to the interpolation, give more details. If not, what about just saying, "The GCM variables are saved at an interval of XXX hours and interpolated on to the time steps of the AMOC-emulator", something along the lines.*

**The line has been removed.**

*6.0.27 p. 8, ll. 5–6:*

*A similar problem. If interpolation is so noteworthy, give more details. If it's not so big a deal, just say, "the GIS melt forcing is interpolated. . . ." instead of "Note that the GIS melt forcing needs to be interpolated. . . ."*

**The line has been removed.**

**AMOC-emulator framework M-AMOC1.0 for uncertainty assessment of future projections**

Pepijn Bakker[1,*] and Andreas Schmittner[1]

[1]College of Earth, Ocean and Atmospheric Sciences, Oregon State University, USA
[*]Now at MARUM – Center for Marine Environmental Sciences, University of Bremen, Bremen, Germany

*Correspondence to:* Pepijn Bakker (pbakker@marum.de)

**Abstract.** State-of-the-science global climate models show that global warming is likely to weaken the Atlantic Meridional Overturning Circulation (AMOC). While such models are arguably the best tools to perform AMOC projections, they do not allow a comprehensive uncertainty assessment because of limited computational resources. Here we introduce the AMOC-emulator M-AMOC1.0, a model framework designed for probabilistic projections of multi-centennial time scales. M-AMOC1.0 uses complex climate model results to force and tune a box model of the AMOC. Box model parameters are adjusted using a simulated annealing procedure. We provide a detailed description of the AMOC box model and show how complex climate model output can be used to force and tune the box model. Finally, we provide an example based on simulations of future climate change including increased greenhouse-gas levels and enhanced melting of the Greenland Ice Sheet performed with the UVic climate model of intermediate complexity. Despite its simplicity, we show that this modeling framework can capture the first order response of the AMOC in UVic to climate change and thus provide a method that can in future studies be applied to existing and new climate change simulations to provide thorough uncertainty assessments.

**1 Introduction**

The Atlantic Meridional Overturning Circulation (AMOC) is an important part of the climate system due to its effects on the transports of heat, salt, carbon, nutrients and other tracers. Projections consistently show a reduction of the AMOC due to future global warming (*IPCC Climate Change*, 2013), with the possibility of an irreversible transition to a shutdown state (Stommel, 1961; Stouffer et al., 2006), a prime example of a tipping point in the climate system (Lenton et al., 2008). Large ensemble simulations are necessary for probabilistic projections, policy relevant risk assessment of future emission scenarios and to assess the uncertainties of AMOC projections due to climate sensitivity, polar amplification, melting of the Greenland Ice Sheet (GIS) and model dependent sensitivity of the AMOC to such climatic changes. The high degree of complexity and spatial resolution of GCMs make them too computationally expensive to perform such an analysis and thus a model is needed that is much cheaper to run, but nonetheless captures important characteristics of the GCM's AMOC response to climate change.

To this end we developed an AMOC-emulator framework. It entails a simple box model that uses physical relationships to represent the most important mechanisms and feedbacks that govern the AMOC's response to changes in regional surface

temperatures, freshwater fluxes and enhanced melting of the GIS. The AMOC-emulator can be forced by temperature and melt water fluxes from any GCM, and using AMOC time series the free parameters of the box model are tuned to mimic the GCM's AMOC sensitivity to future climate change. The AMOC-emulator presented here uses an adjusted version of the multi-box ocean model of Zickfeld et al. (2004), coupled to a 1D atmospheric energy balance model to include the first-order feedback

5   of atmospheric heat transport to AMOC-induced regional temperature changes (Bjerknes, 1964; Rahmstorf and Willebrand, 1995). It describes the AMOC strength as a linear function of the density contrast between the North Atlantic and the South Atlantic, as first introduced by Stommel (1961). The combined physics-based AMOC model, GCM-based forcings and the statistical tuning method presented in this manuscript is dubbed the M-AMOC1.0 model framework. Previous studies have used simple models to emulate the AMOC behavior of higher complexity climate models, either based on a physical model

10  (Zickfeld et al., 2004; Schleussner et al., 2011) or on a statistical model (Challenor et al., 2006; Schleussner et al., 2014). However, the approach described here is designed specifically to allow future studies in which a limited number of climate projections from multiple GCMs, limited in the simulated forcing scenarios and simulation length, can be combined into a Bayesian framework of century time-scale probabilistic AMOC projections.

   At the center of our approach is the assumption that changes in AMOC strength are linearly related to changes in the Atlantic

15  meridional density contrast. Since Stommel (1961) a large number of studies have provided evidence for an important role of the Atlantic meridional density contrast in driving AMOC changes (e.g. Rahmstorf, 1996; Gregory and Tailleux, 2011; Butler et al., 2016). Nonetheless, this model neglects several important processes, like the role of Southern Ocean upwelling, winds and deep water formation (e.g. Gnanadesikan, 1999; de Boer et al., 2010) and a unified theory describing the fundamental mechanisms driving and sustaining the AMOC lacks to this date (Lozier, 2010). Using a Stommel model to emulate AMOC

20  changes driven by surface temperature and freshwater forcings seems appropriate in the light of present-day knowledge and the apparent leading role of surface buoyancy changes in simulated future AMOC weakening (*IPCC Climate Change*, 2013). Moreover, the model is easy to use, interpreted and can be forced directly with GCM-based forcing fields. Nonetheless, the processes that have been omitted and the simplicity of the model should be considered when interpreting the results.

   In the following the governing equations of the AMOC-emulator are given, a description of how the AMOC-emulator can

25  be forced by GCM-based changes in regional temperatures and GIS melt changes and how the parameters in the AMOC-emulator can be tuned towards AMOC projections of an individual higher order climate model using Simulated Annealing (SA). Afterwards, the methodology will be tested for a series of climate projections with the model of intermediate complexity UVic forced by different scenarios for greenhouse-gas concentration changes and melting of the GIS. Finally, the predictive power of the AMOC-emulator will be evaluated by comparing independent UVic AMOC projections with the outcomes of the

30  UVic-based AMOC-emulator.

**2 Description of the M-AMOC1.0 AMOC-emulator framework**

**2.1 AMOC-emulator box model**

The box model that is at the heart of the AMOC-emulator framework presented here is an extension of the box model described by Zickfeld et al. (2004), that is in turn based on the classical Stommel model (Stommel, 1961). A 1-D atmospheric energy balance model (Sellers, 1969) is added to include the stabilizing feedback between atmospheric meridional heat transport and AMOC induced changes in the meridional temperature gradient (Rahmstorf and Willebrand, 1995), so-called Bjerknes compensation (Bjerknes, 1964). The oceanic part of the AMOC-emulator consists of four well mixed boxes that represent the low-latitude upper Atlantic ocean, North Atlantic, South Atlantic and the deep Atlantic Ocean (Fig. 1). The meridional ocean volume transport of the AMOC $(\mathrm{m}^3\mathrm{s}^{-1})$ is proportional to the density gradient between the North and South Atlantic:

$$m = k\frac{\rho_3 - \rho_1}{\rho_{\mathrm{O}}} \tag{1}$$

with $\rho_{\mathrm{O}}$ the reference density for sea water (Tab. 1) and $k$ a hydraulic constant, a tunable parameter (Tab. 2) that links the meridional density difference to the AMOC strength. Densities are derived from temperature and salinity following the non-linear equation of state of seawater of Millero and Poisson (1981). Temperature and salinity in the different boxes are a function of advection by the AMOC and exchange with the overlying atmosphere. Atmosphere-to-ocean heat exchange $H_{\mathrm{O}i}$ relates to the temperature difference between the ocean $(T_{\mathrm{O}i})$ and the atmospheric surface temperature $(T_{\mathrm{A}i})$ through a thermal coupling constant $\Gamma$, a parameter used in our tuning procedure (Tab. 2):

$$H_{\mathrm{O}i} = \Gamma(T_{\mathrm{A}i} - T_{\mathrm{O}i}), \qquad i = 1,..,3. \tag{2}$$

Meridional freshwater fluxes between the South Atlantic and the low-latitude Atlantic $(F_1)$ and between the low-latitude Atlantic and the North Atlantic $(F_2)$ are formulated to represent both atmospheric water vapor transport as well as wind-driven oceanic freshwater transport. Note that both $F_1$ and $F_2$ are defined positive for northward freshwater transport, therefore, poleward atmospheric water vapor transport is a positive contribution to $F_2$ and a negative contribution to $F_1$. The sign of $F_1$ and $F_2$ depends on the sum of both atmospheric water vapor and wind-driven oceanic freshwater transport. Combining equations (1) and (2), changes in ocean temperature $(T_{\mathrm{O}})$ and salinity $(S)$ as a function of time $(t)$ are given by the following set of ordinary differential equations for the four ocean boxes (see Fig. 1 for the definitions and extents of the different boxes):

$$\frac{dT_{O1}}{dt} = \frac{m}{V_1}(T_{O4} - T_{O1}) + \lambda_1(T_{A1} - T_{O1}) \tag{3}$$

$$\frac{dT_{O2}}{dt} = \frac{m}{V_2}(T_{O1} - T_{O2}) + \lambda_2(T_{A2} - T_{O2}) \tag{4}$$

$$\frac{dT_{O3}}{dt} = \frac{m}{V_3}(T_{O2} - T_{O3}) + \lambda_3(T_{A3} - T_{O3}) \tag{5}$$

$$\frac{dT_{O4}}{dt} = \frac{m}{V_4}(T_{O3} - T_{O4}) \tag{6}$$

$$\frac{dS_1}{dt} = \frac{m}{V_1}(S_4 - S_1) + \frac{S_0 F_1}{V_1} \tag{7}$$

$$\frac{dS_2}{dt} = \frac{m}{V_2}(S_1 - S_2) + \frac{S_0(F_1 - F_2)}{V_2} \tag{8}$$

$$\frac{dS_3}{dt} = \frac{m}{V_3}(S_2 - S_3) + \frac{S_0(F_2 + F_{GIS)}}{V_3} \tag{9}$$

$$\frac{dS_4}{dt} = \frac{m}{V_4}(S_3 - S_4) \tag{10}$$

Here the box volumes are given by $V_i$, $\lambda_i$ are individual coupling constants and $F_{GIS}$ is the imposed melt rate of the GIS. The individual coupling constants in the different boxes are related to the thermal coupling constant $\Gamma$, the box thicknesses $z_i$, $\rho_O$ and the specific heat capacity of seawater $c_O$ (Tab. 1):

$$\lambda_i = \frac{\Gamma}{c_O \rho_O z_i}, \qquad i = 1,..,3. \tag{11}$$

Changes in atmospheric component of the meridional freshwater transport $F_1$ and $F_2$ are parameterized as a function of global atmospheric surface temperature anomalies $\Delta T_{GLOB}$ following Zickfeld et al. (2004):

$$F_i = F_{0i} + h_i \Delta T_{GLOB}, \qquad i = 1,..,2. \tag{12}$$

Here $F_{0i}$ are the combined wind-driven oceanic and atmospheric meridional freshwater fluxes for the reference state and $h_i$ the regional hydrological sensitivities. In this description we make use of the proportionality between global temperature changes and meridional atmospheric water vapour transport found in complex climate models (Manabe and Stouffer, 1981; Ganopolski et al., 2001). Freshwater fluxes $F_{01}$, $F_{02}$ and coefficients $h_1$ and $h_2$ are included in the tuning procedure (Tab. 2). We assume that wind-driven oceanic meridional freshwater transport is static. The freshwater term in the salinity calculation of the North Atlantic box (Eq. 9) allows to include a freshwater forcing originating from the GIS ($F_{GIS}$). Note that this freshwater term is not compensated for elsewhere and therefore leads to global mean salinity changes.

The 1D energy balance model describes the atmospheric surface temperatures $T_{Ai}$ as a function of the top-of-the-atmosphere incoming shortwave radiation (first term on rhs of Eq. 13), outgoing longwave radiation (second term on rhs), meridional

atmospheric heat divergence (third term on rhs) and atmosphere-to-ocean heat exchange (fourth term on rhs; see also Eq. 2) through:

$$C\frac{dT_{\mathrm{A}i}}{dt} = I_i(1-\alpha_i) - \epsilon_i\sigma T_{\mathrm{A}i}^4 - \frac{dH_{\mathrm{A}i}}{dy} - H_{\mathrm{O}i}, \qquad i = 0,..,4.$$  (13)

where $\sigma$ is the Stefan Boltzmann constant and $I_i$ and $\alpha_i$ the latitude dependent yearly mean incoming shortwave radiation and planetary albedo, respectively (see Tab. 1 for details). The factor $\epsilon_i$ in the longwave radiation term of Eq. 13 is traditionally regarded as the atmospheric emissivity. However, as we will see later on, in the context of this AMOC-emulator it is more appropriate to view it as a 'total atmosphere effect', that is a temporal and spatial varying parameter that effectively combines atmospheric emissivity, the greenhouse effect and all other processes included in a GCM that cause regional temperatures to differ from global temperature changes. In the next section it is described how the 'total atmosphere effect' is used to include GCM specific future changes in regional temperatures into the AMOC-emulator. The third term on the rhs of Eq. 13 describes meridional atmospheric heat fluxes, represented here as a diffusive process:

$$H_{\mathrm{A}i} = -CK_i\frac{dT_{\mathrm{A}i}}{dy}, \qquad i = 0,..,4.$$  (14)

with $y$ the meridional distance and $K_i$ the meridional atmospheric eddy diffusivity. The latter includes a simple meridional dependency following a sinusoidal profile based on observations of the meridional heat flux and temperatures (Tab. 1). Meridional heat fluxes are assumed zero at the northern and southern boundaries of the domain. By including meridional atmospheric heat fluxes, we allow for a negative feedback between changes in North Atlantic sea-surface temperatures and atmospheric meridional heat transport. As the AMOC weakens, the North Atlantic cools while the low-latitude Atlantic warms, increasing the meridional temperature contrast that in turn leads to an increase in meridional atmospheric heat transport towards the North Atlantic.

We performed the integration of the set of equations using a simple Euler forward scheme and an asynchronous coupling of the energy balance model and the AMOC box model, with time steps of 7 days and 28 days respectively.

The AMOC-emulator includes a number of constants and parameters. In Tab. 1 all constants and prescribed parameter values are given while the tunable parameters are discussed in Sect. 2.3 and listed in Tab. 2. We acknowledge that some of the prescribed parameter values are uncertain, however, the behavior of the AMOC-emulator is dominated by the tuning parameters.

**2.2 AMOC-emulator forcings**

The AMOC-emulator is designed to be forced with regional temperature changes and/or GIS melt provided by a specific GCM simulation. Regional temperature changes can be used to force the AMOC-emulator through the 'total atmosphere effect'

(see also Sec. 2.1). Using Eq. 13, assuming steady state and no atmosphere-ocean heat exchange we can use the individual GCM-based regional temperature time series ($T_{\mathrm{GCM}i}$) to solve for the time evolution of $\epsilon_i$:

$$\epsilon_i = \frac{I_i(1-\alpha_i) - \frac{d}{dy}\left(-CK_i\frac{dT_{\mathrm{GCM}i}}{dy}\right)}{\sigma T_{\mathrm{GCM}i}^4}, \qquad i = 0,..,4. \tag{15}$$

Through this method we ensure that the regional surface temperature forcings of the AMOC-emulator resembles that of
5   the GCM while still allowing for the AMOC to feedback on surface temperatures by changing the oceanic heat transport and consequently the atmospheric meridional heat transport. We acknowledge that simulated regional surface temperatures of the AMOC-emulator forced by prescribed changes in 'total atmosphere effect', will diverge somewhat from the GCM time series when subsequently atmosphere-ocean heat exchange is included, allowing the AMOC to redistribute heat, but the effect on regional temperatures is small, with a maximum impact of ~0.1K in the North Atlantic in the UVic-based (Sect. 3.1)
10  temperatures used in this study (Fig. 4). Note that this does not imply that the impact of meridional atmospheric heat transport is negligible, because its impact depends strongly on the evolution of the magnitude and spatial distribution of atmospheric and oceanic temperatures. We use regional surface temperatures averaged over all longitudes in a specific latitude band as specified by the latitudinal extents of the different boxes in the atmospheric component of the AMOC-emulator.

Previous AMOC projections did not consider the role of enhanced melting of the GIS in a warming world (*IPCC Climate Change*, 2013). In an ongoing model inter-comparison project (AMOCMIP), a number of state-of-the-science ocean-
15  atmosphere general circulation models (GCMs) are forced with projected changes in greenhouse-gas concentrations and realistic GIS melt to improve existing AMOC projections (Bakker et al., 2016). We therefore explicitly include the possibility to include GIS melt in the AMOC-emulator framework as it is introduced into the GCM ($F_{GIS}$ in Eq. 9), the methodology that we used to project future GIS melt is presented in Sect. 3.1.

**2.3 AMOC-emulator parameter tuning**

A number of parameters determine the AMOC-emulator's sensitivity to changes in regional freshwater fluxes and temperatures, and its response timescales. These parameters allow tuning of the AMOC-emulator to reproduce AMOC evolutions simulated by a specific GCM. We choose seven parameters: $k, \Gamma, F_{01}, F_{02}, h_1, h_2$ and $V_4$. Note that the volume of the deep Atlantic box ($V_4$) is viewed as a tuning parameter because it represents the unknown and model-dependent portion of the deep ocean that
25  is involved in transporting North Atlantic Deep Water southwards. This selection of parameters is somewhat subjective, but it proved a good balance between, on the one hand, sufficient degrees of freedom to tune the AMOC emulator's behavior towards that of a specific GCM and, on the other hand, efficiency in finding optimal parameter fits.
The parameters are tuned by minimizing a total cost function $C$ that describes the misfit between the AMOC evolution simulated by a specific GCM and the AMOC-emulator:

$$C = \sum_{s=1}^{s=end} \sum_{t=1}^{t=end} (m_{\mathrm{EMU}st} - m_{\mathrm{GCM}st})^2 \tag{16}$$

with $m_{\text{EMU}st}$ the AMOC strength simulated by the AMOC-emulator for forcing scenario $s$ and time $t$, and $m_{\text{GCM}st}$ the corresponding AMOC strength simulated by the GCM. This method allows us to include a number of different climate simulations that are driven by different forcings and of different lengths into the tuning of a single AMOC-emulator.

5    The tuning procedure of the AMOC-emulator follows a SA algorithm (Lombardi, 2015), a probabilistic technique to approximate the global optimum of a given function (Fig. 2). In short, it generates a candidate parameter set by applying a random perturbations to the parameters of the previous iteration, and subsequently accepts or declines this new parameter set based on a stochastic mechanism. An example of how the parameters evolve through the SA algorith is given in Figure 3. In detail the method works as follows:

10    1. Start the first iteration by picking random initial values $p_{\text{INIT}i}$ for the different parameters from a uniform distribution $U$:

$$p_{\text{INIT}i} = U((1-z)p_{\text{START}i}, (1+z)p_{\text{START}i}), \qquad i = 1,..,7. \tag{17}$$

with $p_{\text{START}i}$ the start value of tuning parameter $i$ and $z$ a multiplication constant. For $p_{\text{START}i}$ we use the values found by Zickfeld et al. (2004) and $z = 2$, the latter implying that the initial values are randomly selected from the start values $\pm 200\%$. The appropriateness of this arbitrary range of initial parameter values is later verified by ensuring that all final
15    parameter values are well within the initial range. Step 1 is repeated until all parameter values are within the bounds given in Tab. 2.

   2. For this set of initial parameters the AMOC-emulator is used to simulate all scenarios for which GCM output is available and the initial cost function $C_{\text{INIT}}$ (Eq. 16) is calculated.

   3. In the iterative part of the SA algorithm for iteration $it$, candidate parameters $p_{\text{CAND}i}$ are obtained analogous to step 1:

20    $$p_{\text{CAND}i} = U((1-\psi_{it})p^i_{it-1}, (1+\psi_{it})p^i_{it-1}), \qquad i = 1,..,7 \qquad it = 1,...,end \tag{18}$$

Note that for $it - 1 = 0$, the initial values are used instead of the values from the previous iteration step. In contrast to step 1, the multiplication factor $\psi_{it}$ for iteration $it$, the so-called 'SA-temperature', is chosen here to be a function of the magnitude of cost function. This allows for a finer tuning as the global optimum is approached:

$$\psi_{it} = min(\psi_{max}, max(\psi_{min}, a10^b \log_{10} C^c)) \tag{19}$$

25    This formulation enables the algorithm to test candidate parameter values that are far away from the current parameter values during the first stage of the tuning process ($\psi_{max}$=0.2 gives maximum 20% change in parameter values), while

as the global optimum is approached, only local candidate parameters are tested that are close to the current parameter values ($\psi_{min}$=0.01 gives minimum 1% change in parameter values). For constants $a$, $b$ and $c$ we used 2.9, -4 and 3.36 respectively. The efficiency of the tuning procedure is fairly sensitive to the choices of parameters $\psi_{min}$, $\psi_{max}$, $a$, $b$ and $c$. However, the resulting GCM-based AMOC-emulator parameter sets are not. Step 3 is repeated until all values are within the bounds given in Tab. 2.

4. The AMOC-emulator then uses the candidate parameter values $p_{\text{CAND}i}$ to simulate the corresponding AMOC evolution and candidate cost function $C_{\text{CAND}}$.

5. Whether or not the candidate parameter values $p_{\text{CAND}i}$ are kept is determined by a stochastic mechanism that enables the algorithm to 'escape' from a local minimum in the cost function. Figure 3 provides an example of how the cost function increases on several iteration steps in order to escape local minima. The stochastic mechanism works as follows:

if $C_{\text{CAND}} < C_{it-1}$ then $p_{it}^i = p_{\text{CAND}i}$,      i=1,..,7.

if $C_{\text{CAND}} => C_{it-1}$ then:

        if $U(0,1)$ < A then $p_{it}^i = p_{\text{CAND}i}$ else $p_{it}^i = p_{it-1}^i$,      i=1,..,7.

with $U(0,1)$ a random draw from a uniform distribution between 0 and 1. The acceptance criterion $A$ is based on the SA-temperature $\psi_{it}$ and the SA-temperature maximum $\psi_{max}$:

$$A = \psi_{it} A_{INIT} \psi_{max}^{-1}.$$

For the initial acceptance criterion value $A_{\text{INIT}}$ we use 0.6.

6. Steps 3-5 are repeated until a stopping criterion is satisfied. The latter is the case if significant improvements are no longer made. More specifically when the linear trend in $C$ over the last $n$ iterations falls below a critical slope $s_c$. We used $n$=50 and $s_c$=0.05.

Because of non-linearities in the AMOC-emulator, there is more than one parameter set that provides a good fit between the AMOC evolution in the AMOC-emulator and a specific GCM. Moreover, because of the simplicity of the AMOC-emulator, a single best performing parameter set cannot be expected. Therefore, to investigate the uncertainty in parameter space it is appropriate to repeat the SA algorithm until a number of reasonable fits is found and make a selection from this based on the corresponding cost function values. As an example we show here the results for a total number of 100 reasonable fits of which we retain the 10 with the parameter sets with the smallest corresponding cost function. A good maximum cost function for a reasonable fit was found to be $5*10^4 (\text{m}^3\text{s}^{-1})^2$, but we note that this value is dependent on the total number of years used in the fitting procedure.

To provide an idea of the computational expenses of the model we provide a back of the envelope calculation. This shows that a single run over all scenarios takes $\sim 10^5$ time steps which are done in about 5 seconds. You need on the order of 400 iterations (in which parameter values are perturbed) to find a single reasonable fit, resulting in approximately half an hour to calculate a single reasonable fit on a normal desktop computer.

5 ## 3   Testing the AMOC-emulator

**3.1   UVic climate projections**

To investigate the appropriateness of the GCM-based forcings, the effectiveness of the tuning procedure and the resulting predictive power of the resulting AMOC-emulator we performed a number of climate change experiments and sensitivity tests with the University of Victoria (UVic) Earth System Model version 2.9 (Weaver et al., 2001). This model of intermediate 10 complexity includes a three-dimensional dynamical ocean with 19 vertical levels at $3.6° \times 1.8°$ horizontal resolution governed by the primitive equations, coupled to a two-dimensional single-level atmosphere, with moisture and heat balances and fluxes between the two mediums, and a dynamical sea ice model.

Five experiments are used to tune the AMOC-emulator, one following historical GHG changes (1850-2006; referred to as Historical) and four following RCP4.5 and RCP8.5 (Representative Concentration Pathways;  Meinshausen et al., 2011), 15 including or excluding enhanced melting of the GIS (period 2006-2300; experiments are referred to as RCP4.5, RCP4.5-GIS, RCP8.5 and RCP8.5-GIS respectively; see Tab. 3). In these simulations only changes in GHGs and GIS melt are considered, all other boundary conditions are fixed at pre-industrial levels.

The UVic-based regional annual mean atmospheric temperature forcings of the AMOC-emulator are shown in Fig. 4 and available in the supplement. Note that the UVic-based surface temperature evolution exhibits multi-decadal oscillations that 20 result from global climate variability originating from the Southern Ocean, oscillations that the AMOC-emulator translates into AMOC variations as will become clear in Sect. 3.2.

The GIS melt scenarios used in the UVic climate change simulations and in the UVic-based AMOC-emulator simulations, are based on the methodology of Lenaerts et al. (2015). Using a high-resolution regional atmospheric climate model for Greenland (RACMO2), forced by a future projection simulated with the HadGEM2-ES GCM following RCP4.5, they describe 25 a strong correlation between local annual mean runoff and 500hPa summer (JJA) temperature changes over different parts of Greenland. We have used this relationships in combination with CMIP5 multi-model-mean 500hPa summer atmospheric temperature changes to construct GIS melt projections for RCP4.5 and RCP8.5. A fixed seasonal cycle in GIS melt was added following Lenaerts et al. (2015). The resulting GIS melt scenarios show a strong increase in GIS melt with annual mean values of $\sim$0.02Sv and $\sim$0.08Sv ($1\text{Sv} = 10^6 \text{m}^3 \text{s}^{-1}$) in the year 2300 for RCP4.5 and RCP8.5, respectively and with summer 30 maxima almost an order of magnitude larger than this (Fig. 5). Note that the runoff projections of Lenaerts et al. (2015) include simulated changes in evaporation, precipitation, snow melt and ice melt. Therefore Greenland runoff calculated within UVic is fixed at the simulated 1970-2001 average values to avoid double counting the changes in precipitation, evaporation and snow melt. Moreover, this methodology neglects future changes in GIS solid ice discharge. This choice is motivated by the large

uncertainty in the observed and projected sensitivity of GIS solid ice discharge to climate warming, in its magnitude and even in the sign of the future GIS ice discharge contribution to sea-level rise (Nick et al., 2013; Lenaerts et al., 2015; Vizcaino et al., 2015). The GIS melt forcing used in this manuscript is available in the supplement.

**3.2 UVic-based AMOC-emulator**

5  Five UVic simulations are used in the tuning procedure of the AMOC emulator (Tab. 3). An example of how the cost function, SA-temperature and the tuning parameters of the emulator evolve during the SA tuning process is given in Figure 3 and the parameter ranges of the ten best UVic-based AMOC-emulators values are listed in Tab. 2. In this study we use the maximum meridional overturning streamfunction at 26°N below 500m depth as a measure of the AMOC strength, but note that the AMOC-emulator allows one to chose another measure of the AMOC strength.

10  Scatter plots between the individual tuning parameters (Fig. 6) reveal some interesting aspects of the AMOC emulator. There is a strong and nonlinear relationship between the hydraulic constant $k$ and the thermal coupling constant $\Gamma$, with a larger hydraulic constant leading to a smaller thermal coupling constant. Furthermore, in accordance with Equation 12, there is a quasi-linear relation between $F_{01}$ and $h_1$ and also between $F_{02}$ and $h_2$. Finally, it shows that certain parameters are better constrained (i.e. of larger importance), show more clustering of the values with a low cost function (blue colors in Figure 6), for instance $k$, $\Gamma$, $F_{01}$, $V_4$ and $h_1$, while the opposite is true for $F_{02}$ and $h_2$. The smaller importance of $F_{02}$ and $h_2$ also follows from their absence in the steady-state solutions of equations 3–10, as already noted by Zickfeld et al. (2004).

The ten best UVic-based AMOC-emulators are able to reproduce the overall characteristics of the AMOC in UVic in terms of its sensitivity to changes in heat and freshwater forcing (Fig. 7). Both UVic and the UVic-based AMOC-emulators show a decline of the AMOC of ~1Sv over the historical period and a further weakening of ~5Sv and ~9Sv in RCP4.5 and RCP8.5,
20  respectively. The impact of GIS melt appears small for both RCP4.5-GISmelt and RCP8.5-GISmelt (~1Sv). The ten different AMOC emulators are also fairly consistent with each other. There are also differences between the AMOC evolution simulated by UVic and with the UVic-based AMOC-emulators. Most notably is that the UVic-based AMOC-emulator misses the partial AMOC recovery of ~1Sv after 2150 in RCP4.5 and RCP4.5-GISmelt (resulting in root-mean-square-errors, RSME, of $0.88\pm0.08(\text{m}^3\text{s}^{-1})^2$ and $0.78\pm0.07(\text{m}^3\text{s}^{-1})^2$ respectively; $\mu\pm\sigma$), appears slightly too sensitive to the GIS melt forcing
25  in simulation RCP8.5-GIS (resulting in the largest RSME of the four simulations of $0.9\pm0.13(\text{m}^3\text{s}^{-1})^2$) and simulates a too direct response of the AMOC strength to multi-decadal surface temperature oscillations. Nonetheless, the mismatch between the AMOC-emulator and the UVic-based AMOC evolution is smaller than ~10% of the AMOC strength.

Because of the simplicity of the AMOC box model, it is not surprising that it does not fully capture the AMOC behavior of the higher order climate model. Nonetheless, it is important to realize that using any type of emulator will introduce new
30  uncertainty into the analysis. It is also worth noting that the fit for an individual simulation could be improved, for instance the AMOC-emulator does allow for a partial AMOC recovery as UVic shows for RCP4.5, but such an AMOC-emulator is not found through the SA tuning methodology in this example, because it would degrade the fit for the other scenarios and thus lead to an overall higher cost function. More discussion on this topic follows in Sect. 4.

In the presented AMOC-emulator we included a 1D atmospheric energy balance model to allow for a feedback of atmospheric heat transport to AMOC induced surface temperature changes. To investigate the importance of feedback on the AMOC strength evolution, additional experiments have been performed in which this feedback was switched off (Fig. 9). The impact of including atmospheric meridional heat transport is a small, but non-negligible ~1Sv strengthening of the control state of the AMOC (not shown) and, more importantly, a slightly lower sensitivity to changes in radiative forcing and GIS melt (Fig. 9). This confirms our understanding of atmospheric meridional heat transport acting as a negative feedback to AMOC changes. The simulations with the atmospheric feedback included have on average a stronger AMOC by 8.1±1.9% ($\mu \pm \sigma$; calculated over all 10 best fits and over all five forcing scenarios).

**3.3 Predictive power of the UVic-based AMOC-emulator**

The value of constructing an AMOC-emulator is in its capability to perform a large number of simulations that allows quantification of uncertainties for a large range of forcings. This raises the question of the fidelity of the AMOC-emulator and errors induced by its use. Here we will assess the predictive capabilities and errors of the AMOC-emulator by comparing it with results from four additional transient UVic climate change simulations, none of which have been used in the AMOC-emulator tuning procedure.

The four independent transient climate change simulations follow idealized and rather extreme scenarios with changes in either the GHG or GIS melt forcing. In two of the simulations, RCP8.5-GIS is adjusted by multiplying the GHG forcing changes with respect to the 2006 value by a constant factor of 0.5 (RCP8.5x0.5-GIS) or 1.5 (RCP8.5x1.5-GIS). In the third independent simulation the GIS melt forcing of the original RCP8.5-GIS scenario is multiplied by 1.5 (RCP8.5-GISx1.5). The fourth simulation uses the RCP4.5 GHG forcing but combines it with the RCP8.5-GISx1.5 GIS melt forcing (RCP4.5-GISRCP8.5x1.5; see Tab. 3). Some of these idealized scenarios are rather extreme and are unlikely to occur in reality, for instance RCP4.5-GISRCP8.5x1.5, which implies only limited global warming combined with a very large melt rate of the GIS, but they allow us to assess the bounds within which the AMOC-emulator has predictive power, e.g. can or cannot be used.

The predictive power of the AMOC-emulator is quantified by a skill score (SS) following:

$$SS = 1 - \frac{RMSE_{prediction}}{RMSE_{reference}} \tag{20}$$

As reference, or null model, we take an AMOC-emulator with zero prediction skill, meaning that it simply reproduces the original calibration data. We use RCP8.5-GIS as reference for RCP8.5x0.5-GIS, RCP8.5x1.5-GIS and RCP8.5-GISx1.5, and RCP4.5-GIS for RCP4.5-GISRCP8.5x1.5. From Eq. 20 it follows that a skill score between zero and one indicates predictive power of the AMOC-emulator.

We find that for large changes in the GHG forcing, the UVic-based AMOC-emulators are well capable of predicting the AMOC evolution of UVic in terms of sign and amplitude and perform better than the reference model with skill scores of 0.39±0.07 and 0.19±0.06 for RCP8.5x1.5-GIS and RCP8.5x0.5-GIS, respectively (upper panels Fig. 8). For large changes in the applied GIS melt forcing the picture is more complex (lower panels Fig. 8). A strong increase in GIS melt under a low GHG

scenario (RCP4.5-GISRCP-8.5x1.5) shows an excellent performance of the AMOC-emulator with a skill score of 0.76±0.09, but for the high GHG scenario, a 50% increase in GIS melt (RCP8.5-GISx1.5) leads to a deterioration of the fit between UVic and AMOC-emulator with consequently a negative skill score (-1.5±0.56). The latter shows that the UVic-based AMOC-emulators tend to overestimate the impact of GIS melt on the AMOC strength under high-end GHG scenarios. Summarizing, in all four cases the emulator predicts the correct sign of the AMOC response to changes in the forcings, and in three out of four cases the AMOC-emulator is shown to have the predictive power.

**4   Discussion**

Overall, the predictive power of the AMOC-emulator is reasonable when one considers the simplicity of the AMOC box model, but for forcing scenarios that are increasingly far away from the forcings that are used in tuning the AMOC-emulator, the predictive power decreases. A large advantage of using a physics-based AMOC-emulator that is tuned with large climate forcings, over the use of for instance a statistical AMOC-emulator, is that it projects the point after which the AMOC collapses and switches to an off state, as this is an integral part of the physics of the Stommel model. It is clear that using an AMOC-emulator introduces new uncertainty into AMOC projections, however, for which level of added uncertainty an AMOC-emulator is still useful is a question that is difficult to address. Another important consideration when using the AMOC-emulator is the spread in GCM climate forcing scenarios that is included in the tuning process. When using only a single climate change scenario, a better match can be obtained between the AMOC evolution given by the GCM and AMOC-emulator, however, in this case the reliability of the AMOC-emulator will quickly decrease for different climate forcings. On the other hand, one could use a large number of climate change projections in the tuning process to obtain a lesser fit for individual scenarios, but an AMOC-emulator that is applicable to a much larger range of climate change scenarios. The best strategy to be followed strongly depends on the research question in mind.

The assumptions behind the AMOC-emulator presented here, limit it to projecting AMOC changes on multi-decadal and larger timescales. Therefore, the applied GCM-based climate forcings and AMOC strength time series should best be filtered to exclude high frequency variability. Moreover, an AMOC-emulator that is tuned to specific GIS melt experiments is likely not applicable to experiments in which melt water is applied to a different geographical region or with a different seasonal cycle. This is not to say that the presented AMOC-emulator framework cannot equally be applied to other sources of melt water input. Finally, many processes that are known to impact the AMOC are not considered in the AMOC-emulator, for instance the impact of winds, gyre circulation, Southern Ocean upwelling or deep water formation outside of the North Atlantic (see Sect. 1). If such processes would prove to dominate the AMOC response to future climate change, a different AMOC box model should be considered that places emphasis on that particular process.

**5 Summary**

We have presented an AMOC-emulator that can be used to assess the uncertainties in AMOC projections produced by GCMs under future changes in surface temperatures, freshwater fluxes and melting of the GIS. Following a SA algorithm, the AMOC-emulator can be tuned and forced using a limited number of GCM projections of future climate change, making it suited to be applied to model inter-comparison initiatives. We test our methodology by tuning the AMOC-emulator towards the AMOC evolution simulated with the UVic model for the historical period and four future projections up to the year 2300 including increases in GHG concentrations and GIS melt. The results show that the UVic-based AMOC-emulator captures the overall characteristics of the centennial response of the AMOC to changes in regional temperatures and GIS melt. This is confirmed by testing the predictive power of the AMOC emulator for a number of independent UVic simulations that are not used in the tuning procedure. The AMOC-emulator is a valuable tool to study the uncertainty in GCM-based AMOC projections, such as the one recently being performed on the results from the AMOCMIP project (Bakker et al., 2016).

**Code availability**

The MATLAB model code of M-AMOC1.0 and UVic-based example forcing files are available in the supplement.

*Acknowledgements.* This work was supported by a grant from the National Oceanographic and Atmospheric Administration (award number NA15OAR4310239).

[revised manuscript text omitted]